# A Theoretical Study of (Hyper) Self-Attention through the Lens of Interactions: Representation, Training, Generalization

Muhammed Ustaomeroglu [1]   Guannan Qu [1]

## Abstract

Self-attention has emerged as a core component of modern neural architectures, yet its theoretical underpinnings remain elusive. In this paper, we study self-attention through the lens of *interacting entities*, ranging from agents in multi-agent reinforcement learning to alleles in genetic sequences, and show that a single layer linear self-attention can *efficiently* represent, learn, and generalize functions capturing pairwise interactions, including out-of-distribution scenarios. Our analysis reveals that self-attention acts as a *mutual interaction learner* under minimal assumptions on the diversity of interaction patterns observed during training, thereby encompassing a wide variety of real-world domains. In addition, we validate our theoretical insights through experiments demonstrating that self-attention learns interaction functions and generalizes across both population distributions and out-of-distribution scenarios. Building on our theories, we introduce *HyperFeatureAttention*, a novel neural network module designed to learn couplings of different feature-level interactions between entities. Furthermore, we propose *HyperAttention*, a new module that extends beyond pairwise interactions to capture multi-entity dependencies, such as three-way, four-way, or general $n$-way interactions.

## 1. Introduction

Ever since the invention of Transformers (Vaswani et al., 2023), *attention* is the building block for many domains, spanning natural language processing (Brown et al., 2020; Devlin et al., 2019), computer vision (Dosovitskiy et al., 2021), protein structure prediction (Jumper et al., 2021),

reinforcement learning (Chen et al., 2021). Despite the success of attention, our formal understanding of its representation, optimization, and generalization abilities is in its early stages.

Recent theoretical investigations illuminated Transformers' representational abilities/limitations (Liu et al., 2023; Sanford et al., 2023) and training dynamics (Ahn et al., 2023; Jelassi et al., 2022; Gao et al., 2024) from different perspectives (e.g., language modeling, image patch classification, in context learning, etc.). Despite this progress, current theoretical frameworks exhibit critical limitations:

(i) Existing analyses often target isolated problems, lacking a unified perspective for characterizing Transformers' capabilities across diverse domains. In contrast, our theory makes an attempt to provide a unified perspective by assuming that the data comes from a mutual interaction model, which we show captures broad applications.

(ii) Most mathematically rigorous theories overlook test-time generalization, particularly robustness to out-of-distribution (OOD) shifts. Our analysis addresses OOD in terms of length generalization.

(iii) Mathematically rigorous theories typically offer interpretations only for the predetermined parameters. In contrast, our approach explains a broader set of parameters—many of which may initially appear unintuitive.

(iv) Generally, rigorous theories rely on restrictive assumptions about model parameters. In contrast, our framework does not impose such assumptions on the parameters. Our approach only requires mild and possibly inevitable conditions on the data distribution -such as training data versatility.

In this work, we adopt a *interacting entities* viewpoint to study self-attention, where each token represents an interacting entity (e.g., agents in multi-agent reinforcement learning, particles in physical simulations, amino acids in protein sequences, or words in natural language). Specifically, we introduce a function that models interactions among these entities and demonstrate its applicability across diverse domains, including the colliding agents environment, genotype-phenotype mapping task, vision task, and time

---

[1]Department of Electrical and Computer Engineering, Carnegie Mellon University, Pittsburgh, USA. Correspondence to: Muhammed Ustaomeroglu <mustaome@andrew.cmu.edu>.

series prediction. Under this viewpoint, we prove that a single-layer linear self-attention can *efficiently* represent such functions and show that gradient-descent training converges to the parameters realizing these interactions, under mild assumptions on the data.[1][2] In addition, we demonstrate versatility requirements on the train data such that the learned parameters generalize both to the test distribution and to out-of-distribution (length generalization). By neither imposing restrictive constraints on model parameters nor limiting ourselves to particular domains, our framework unifies some diverse application scenarios and offers a novel theoretical lens on how Transformers learn dependencies among multiple interacting entities.

We further validate our theoretical insights on representation, convergence, and generalization through controlled experiments, demonstrating that the learned model parameters closely align with theoretical predictions. Beyond analyzing attention patterns, we highlight how the parameters themselves can be directly interpreted to uncover meaningful interactions among entities.

Building on these insights, investigations confirm that self-attention excels at capturing mutual interactions between entities. Motivated by this, we introduce two novel generalizations named (i) *HyperFeatureAttention*, for capturing couplings of different interactions between features of the entities, and (ii) *HyperAttention*, for capturing higher-order interactions (e.g. three-way or four-way) between entities. Extending our single-layer analysis, we show that HyperFeatureAttention can efficiently represent the couplings of feature interactions. In addition, we show that HyperAttention can represent and learn higher order interactions, with the corresponding theories.

We summarize our main contributions as follows:

- In Section 3, we present a unified perspective where each token (e.g., agent, amino acid, pixel patch) is treated as an *interacting entity*, seamlessly bridging several experimental settings.
- In Section 4, we show that gradient flow, on standard mean squared error loss, converges to a solution that captures how the entities interact. Furthermore, the learned parameters *generalize* to both unseen examples from the task distribution and out of distribution (varying sequence lengths), under suitable versatility conditions in the training data.

---

[1] If linear self-attention requires $\mathcal{O}(k)$ parameters to represent the interaction function, dense layer requires $\mathcal{O}(k \cdot L^2)$ parameters.

[2] We motivate this choice further in subsequent sections, but briefly: linear self-attention preserves essential optimization properties of full Transformers while offering reduced computational complexity and simplified theoretical analysis (Ahn et al., 2024), making it ideal for theoretical investigations. A detailed discussion of linear self-attention appears in Section 2.

- In Section 7, we provide experiments that validate our theoretical predictions with clear interpretation of the learned parameters.
- In Section 5, we introduce *HyperFeatureAttention*, a novel mechanism designed to capture couplings of feature interactions. In Section 6, we present *HyperAttention*, which models *higher-order* dependencies between entities, such as three-way and four-way interactions. Also, we provide accompanying theoretical analyses of these models' capabilities. We also provide some preliminary experiments on these novel models in Section 7.

**Related Works.**

**Transformer Representation Theory.** A large body of work has illuminated the representational abilities of self-attention from various angles (Yao et al., 2023; Bhattamishra et al., 2020a; Wei et al., 2023; Kajitsuka & Sato, 2024; Nath et al., 2024; Luo et al., 2022; Li et al., 2024). For instance, Transformers have been shown to be Turing-complete and capable of simulating intricate sequential computations (Bhattamishra et al., 2020b), act as provably efficient "compilers" for domain-specific languages (Zhai et al., 2024), and approximate key operators such as sparse or local averaging with sublinear complexity (Likhosherstov et al., 2021; Sanford et al., 2023; Edelman et al., 2022). Their abilities and limitations have also been explored in POMDP settings (Lu et al., 2024), automata-theoretic perspectives (Liu et al., 2023), sequence-to-sequence tasks (Yun et al., 2020), and hidden Markov model learning scenarios (Hu et al., 2024).

**Transformer Convergence Analysis.** Parallel to the progress on representation, another line of research has investigated the convergence properties of training Transformers (Ahn et al., 2023; Tarzanagh et al., 2024; Li et al., 2023; Tian et al., 2023; Song et al., 2024; Huang et al., 2024a; Chen et al., 2024). These studies analyze training via gradient flow in simplified yet insightful settings (Yang et al., 2024), establish conditions under which one can efficiently learn multi-head attention layers (Chen & Li, 2024; Deora et al., 2023), or employ mean-field methods to show global convergence in large-scale regimes (Gao et al., 2024). Additional works examine specialized domains such as masked visual pretraining (Huang et al., 2024b) and spatial structure learning (Jelassi et al., 2022), or investigate sparse token selection tasks (Wang et al., 2024).

Despite the valuable insights offered by these studies, the majority have all or some of the limitations, i.e. (i), (ii), (iii) we listed in the second paragraph of introduction.

## 2. Preliminaries

**Self-Attention.** The self-attention mechanism is a core component of Transformers (Vaswani et al., 2023), enabling

models to learn dependencies between input tokens effectively. For a sequence of $L$ input tokens represented as a matrix $\mathbf{X} \in \mathbb{R}^{L \times d}$, the self-attention is defined as:

$$\mathbf{SA}^{\sigma}(\mathbf{X}) = \sigma \left( \frac{\mathbf{X}\mathbf{W}^Q (\mathbf{X}\mathbf{W}^K)^{\top}}{\sqrt{d_k}} \right) \mathbf{X}\mathbf{W}^V,$$

where $\mathbf{W}^Q, \mathbf{W}^K, \mathbf{W}^V \in \mathbb{R}^{d \times d_k}$ are the learnable projection matrices. Defining $\mathbf{C} = \mathbf{W}^Q (\mathbf{W}^K)^{\top} / \sqrt{d_k}$, we can write the same equation as

$$\mathbf{SA}^{\sigma}(\mathbf{X}) = \sigma \left( \mathbf{X}\mathbf{C}\mathbf{X}^{\top} \right) \mathbf{X}\mathbf{W}^V.$$

Since the introduction of *attention* mechanisms (Bahdanau et al., 2016), the function $\sigma : \mathbb{R}^{L \times L} \to \mathbb{R}^{L \times L}$ has been predominantly implemented as a row-wise softmax operation, where the input matrix to $\sigma$, commonly referred to as the *attention scores*, determines the relative importance of different tokens in the sequence.

**Alternative Attention Functions.** Recent advancements have explored alternative $\sigma$ functions beyond the traditional softmax. One prominent direction is using linear self-attention mechanisms, which reduce the computational complexity of self-attention from $\mathcal{O}(L^2)$ to $\mathcal{O}(L)$. Linear attention methods approximate the softmax function while maintaining comparable performance in many tasks (Katharopoulos et al., 2020; Choromanski et al., 2022). For instance, the Performer model introduces kernel-based methods to approximate the softmax operation efficiently, achieving scalability without significant loss in accuracy (Choromanski et al., 2022). Moreover, alternative activation functions, such as ReLU, cosine, polynomial and sigmoid based transformations, have been explored, showing competitive or even superior performance compared to softmax in some tasks (Koohpayegani & Pirsiavash, 2024; Kacham et al., 2024; Ramapuram et al., 2024).

To simplify theoretical analysis while shedding light on a diverse range of self-attention implementations, we adopt a linear variant of self-attention, which preserves the core characteristics of self-attention through its reliance on *attention scores*.

**Linear Self-Attention.** Linear self-attention simplifies the attention by omitting the $\sigma$ operation, resulting in:

$$\mathbf{SA}^{\mathrm{lin}}(\mathbf{X}) = \left( \mathbf{X}\mathbf{C}\mathbf{X}^{\top} \right) \mathbf{X}\mathbf{W}^V. \tag{1}$$

Although omitting the $\sigma$ operation may appear to be oversimplification, extensive theoretical and empirical studies confirm the power of linear self-attention. Notably, layers of linear self-attention can implement gradient descent and preconditioned gradient descent (von Oswald et al., 2023; Ahn et al., 2023). In addition, variants of softmax, ReLU, and linear transformers (including the exact version used here) can perform functional gradient descent to learn nonlinear

functions in context (Cheng et al., 2024). Furthermore, (Ahn et al., 2024) demonstrates that linear self-attention replicates key optimization dynamics of Transformers -such as heavy-tailed gradient noise and ill-conditioned loss landscapes- without softmax or feedforward layers. This simplification retains the computational advantages of linearity while enabling rigorous analysis of phenomena like adaptive optimizer superiority (e.g., Adam over SGD) and gradient convergence. Critically, insights from this abstraction extend to softmax-based Transformers, particularly in understanding optimization stability and generalization under varying data distributions or model depths (Ahn et al., 2024).

In the following sections, we explore the capabilities of linear self-attention across diverse tasks to illustrate its practical and theoretical value. By striking a balance between tractability and expressiveness, linear self-attention offers a powerful framework for investigating and enhancing attention-based architectures.

## 3. Representing Mutual Interactions with Attention

Consider a discrete finite domain (or "vocabulary") $\mathcal{S} = \{\alpha, \beta, \gamma, \omega, \dots\}$ with cardinality $|\mathcal{S}|$. In our setting we have tuples of $L$ elements (or sequences of length $L$) from the domain, denoted by $\mathcal{X}$ and entries of which are uniquely indexed by the integers in $[L]$. Here, $[i]$ denotes the set $\{0, 1, \dots, i-1\}$ for any $i \in \mathbb{Z}^+$. For each index $i$, we denote the corresponding element $\mathcal{X}(i) \in \mathcal{S}$. Thus, we can write $\mathcal{X} = (\mathcal{X}(0), \mathcal{X}(1), \dots, \mathcal{X}(L-1))$, which is distributed according to $\mathcal{X} \sim \mathcal{D}$. We also have tuples $\mathcal{Y}$ with elements from a corresponding relatively small set $\mathcal{S}_{\mathcal{Y}}$. The tuples are jointly distributed according to a task distribution $(\mathcal{X}, \mathcal{Y}) \sim \mathcal{D}_{\mathcal{X} \times \mathcal{Y}}$. In order to train a neural network, we map each element of $\mathcal{S}$ to a $d$-dimensional embedding space via a function $\mathbf{x} : \mathcal{S} \to \mathbb{R}^d$ and each element of $\mathcal{S}_{\mathcal{Y}}$ to a corresponding vector or a scalar depending on the task, via the function $\mathbf{y} : \mathcal{S}_{\mathcal{Y}} \to \mathbb{R}^{d_2}$. Additionally, we stack the embeddings of the elements in $\mathcal{X}$ and $\mathcal{Y}$ as rows of $\mathbf{X} \in \mathbb{R}^{L \times d}$ and $\mathbf{Y} \in \mathbb{R}^{L \times d_2}$ matrices. We denote their distribution as $(\mathbf{X}, \mathbf{Y}) \sim \mathcal{P}_{\mathbf{X} \times \mathbf{Y}}$

Our first result concerns self-attention' the representation for *mutual interactions*, which we define below. We introduce a pairwise effect function. For $\alpha, \beta \in \mathcal{S}$, let $f(\alpha, \beta) \in \mathbb{R}$ measure how strongly entity $\beta$ affects entity $\alpha$, and let $\mathbf{w}_{\beta} \in \mathbb{R}^{d_2}$ represent how that influence is expressed. The *aggregated* effect on the $i$-th entity, from all other entities, is

$$\mathbf{y}_{\mathcal{X}(i)} = \sum_{j \in [L]} f(\mathcal{X}(i), \mathcal{X}(j)) \mathbf{w}_{\mathcal{X}(j)}, \tag{2}$$

capturing mutual interactions: each entity's behavior or state depends on every other entity in the sequence.

**Theorem 3.1** (Representation Ability of Linear Self-Attention). $d = |\mathcal{S}|$ *is sufficient for a single-layer linear self-attention to **exactly** represent any aggregate pairwise interaction functions* $\{\mathbf{y}_{\mathcal{X}(i)}\}_{i=1}^{L}$ *in Eq. 2 for all entities simultaneously. Also,* $d \geq |\mathcal{S}|$ *is necessary for a single layer linear self-attention to **exactly** represent any such functions.*

Consequently, self-attention requires $\Theta(|\mathcal{S}|^2)$ parameters to capture the interactions. The key distinction, however, lies in efficiency. The following theorem demonstrates that self-attention is *efficient* compared to fully connected neural networks. This kind efficiency is one of the reasons we contend that Transformers are mutual interaction learners, while generic fully connected architectures are not.

**Theorem 3.2** (Efficiency of Self-Attention). *A linear fully connected network requires* $\Omega(L^2 \cdot |\mathcal{S}|^2)$ *parameters to represent the aggregate pairwise interaction functions* $\{\mathbf{y}_i\}_{i=1}^{L}$ *in Eq. 2 **exactly**, for all entities simultaneously.*

The proofs of Theorems 3.1 and 3.2 appear in Appendix B. Equation 2 and Theorem 3.1 underpin all later examples: the same simple formula can model multi-agent rewards, pixel patterns, time-series signals, and genotype-phenotype relations, showing that one self-attention layer already captures rich pairwise dependencies.

A more practical interpretation of these results appears in the context of deep neural networks with a **modular perspective**. Learning algorithms, such as gradient descent applied to a loss criterion, often lead to the emergence of task specialization within small subcomponents of a large neural network. Specifically, individual neural network blocks naturally become responsible for particular subtasks -a phenomenon studied in (Elhage et al., 2021; Cammarata et al., 2020). Our representation theorem formally demonstrates that each self-attention block is indeed sufficiently powerful to execute any mutual interaction task potentially allocated to it within a deep neural network, so theoretically supporting the more empirical observations.

We provide the details about this section in Appendix B and provide convergence and generalization analyses in Section 4. Finally, Section 7 presents empirical results that confirm our theoretical findings and provide interpretation of the learned parameters.

**Example 1 (Colliding Agents Environment).** In a multi-agent system with $L$ *identical* agents positioned in a dim-dimensional space, with position vectors $\mathbf{r}_i \in \mathbb{R}^{\dim}$ (or $\mathbf{r}_i \in [N]^{\dim}$ for a discrete setup). We aim to capture how each agent's value function $V_{\mathbf{r}_i}$ depends on other agents' initial states (positions).[3] Seeing that the system is translationally and rotationally invariant, as explained detailly in

---

[3]In a reinforcement learning context, the value function $V_{\mathbf{r}_i}$ for agent $i$ typically denotes the expected return (sum of discounted rewards) from a particular configuration.

Appendix B.1, we can write $j^{th}$ agent's effect on $i^{th}$ agent's value function as, $f(\mathbf{r}_i - \mathbf{r}_j) w_{\mathbf{r}_j}$, where $w_{r_j} \in \mathbb{R}$ is a scalar weight and $f$ depends only on their *relative position*. Then we consider the value function of the $i$-th agent as

$$V_{\mathbf{r}_i} = \sum_{j \neq i}^{L} f(\mathbf{r}_i - \mathbf{r}_j) w_{\mathbf{r}_j}.$$

In Appendix B.1, we illustrate how a reward of $-1$ per distinct collision is captured by the value function above. This function fits directly into (2), allowing a single-layer linear self-attention to represent the multi-agent value function, when discrete positions are encoded orthogonally. In addition in Appendix B.1 we show similar results for non-identical agents setting, too.

**Example 2 (Genotype-Phenotype Mapping Task).** In many genotype–phenotype models, a DNA sequence of length $L$ is composed of alleles which we represent as $\mathcal{X}(i) \in \mathcal{S}$. Some alleles are always active, others' activation level depends on the presence of certain alleles, and some remain inactive regardless of context (Frommlet et al., 2016). Formally, in a simplistic setting,

$$\begin{aligned} P_{\mathcal{X}(i)} = \ & \mathbb{I}\{\mathcal{X}(i) \text{ is always active}\} \\ & + \sum_{j \in [L]} \mathbb{I}\{\mathcal{X}(i) \text{ is activated by } \mathcal{X}(j)\} \ w_{\mathcal{X}(j)}, \end{aligned}$$

where $P_{\mathcal{X}(i)}$ captures activeness of the gene at position $i$. By orthogonally embedding each allele, Theorem 3.1 again ensures a single-layer linear self-attention can replicate these interactions exactly. For more details see the demonstrations at Appendix B.2.

We illustrate the generality of our theory with several additional case studies, time-series prediction (Appendix B.4), a computer-vision task (Appendix B.3), and variants of the colliding-agent environment (Appendix B.1.3). Studying isolated single-layer self-attention models uncovers component-level behaviors that directly inform the design and optimization of deep Transformer architectures. This mirrors the role of circuit theory in electrical engineering: although large-scale designs rely on simulation and proto-typing, foundational insights about transistors, resistors, and capacitors remain indispensable. Likewise, our block-level theory is a step toward a datasheet for self-attention units, helping steer the construction of deep Transformers. Full-network experiments remain vital, but a precise block-level theory focuses the design space and boosts the likelihood of success.

**Why** $d = |\mathcal{S}|$ While setting the embedding dimension $d$ equal to the domain size $|\mathcal{S}|$ may be impractical for large vocabularies, it simplifies our analysis without altering core insights. Our goal is to understand how self attention cap-

tures mutual interactions, and $d = |\mathcal{S}|$ ensures orthogonal domain embeddings, yielding an exact and transparent representation. Compressing to $d < |\mathcal{S}|$ is perpendicular to this focus and can be addressed separately using standard techniques, such as Johnson-Lindenstrauss projections, to approximate high-dimensional orthogonal embeddings. Starting with $d = |\mathcal{S}|$ allows us to establish clean, exact theorems that elucidate how self-attention captures pairwise interactions, while reducing to $d < |\mathcal{S}|$ merely introduces a small approximation gap without altering the core theory. For completeness, we provide an approximate version of Theorem 3.1 in Appendix B (Theorem B.2).

## 4. Training and Generalization

In this section, we analyze how a single layer linear self-attention can achieve zero training error and demonstrate its *generalization* guarantees under mild assumptions. We focus on learning mutual interaction functions of the form (2), which, as ensured by Theorem 3.1, can be represented by linear self-attention.

**Setup and Notation.** Let $\left\{ \left( \mathbf{X}^{(n)}, \mathbf{Y}^{(n)} \right) \right\}_{n=1}^{B}$ be the training data, where $(\mathbf{X}^{(n)}, \mathbf{Y}^{(n)}) \sim \mathcal{P}_{\mathbf{X} \times \mathbf{Y}}^{L^*}$, for a specific $L^*$, so in the training set all tuples have the same length $L^*$. Also, let $\mathcal{P}_{\mathbf{X} \times \mathbf{Y}}^{\forall L}$ be the universal distribution that covers samples of any length. Throughout, we focus on the mean-squared error (MSE) objective

$$L^{\mathrm{MSE}}\left( \mathbf{C}, \mathbf{W}^V \right) = \frac{1}{B} \sum_{n=1}^{B} \left\| \mathbf{SA}_{\mathbf{C}, \mathbf{W}^V}^{\mathrm{lin}}\left( \mathbf{X}^{(n)} \right) - \mathbf{Y}^{(n)} \right\|^2,$$

We address three key questions: **(1) Convergence:** Under what conditions does gradient flow reach zero training error? **(2) Generalization:** When does a perfect fit on the training set imply zero error on *new* data from $\mathcal{P}_{\mathcal{X} \times \mathcal{Y}}^{L^*}$? **(3) OOD Generalization:** Can such a model generalize to *longer* or *shorter* sequences than those seen in training?

**Definition 4.1** (Data Matrix for Element $\mu$). Let $\mathcal{B}_\mu$ as the set of training indices that contain element $\mu \in \mathcal{S}$. Denoting the number of times an element $\mu$ appears in tuple $\mathcal{X}^{(n)}$ as $s_\mu^{(n)}$, we define the data matrix for element $\mu$ as

$$\mathbf{s}^{(n)} = \begin{bmatrix} s_\alpha^{(n)} & s_\beta^{(n)} & \dots \end{bmatrix}^\top, \mathbf{S}_{\mathcal{B}_\mu} = \begin{bmatrix} \dots & \mathbf{s}^{(n)} & \dots \end{bmatrix}_{n \in \mathcal{B}_\mu}^\top.$$

In short, $\left[ \mathbf{S}_{\mathcal{B}_\mu} \right]_{n\nu} = s_\nu^{(n)}$, but for $n$ such that $\mu \in \mathcal{X}^{(n)}$.

**Assumption 4.2** (Training Data Versatility). For all $\mu \in \mathcal{S}$, $\mathbf{S}_{\mathcal{B}_\mu}$ is full column rank.

This assumption is mild in practice. When elements in $\mathcal{X}^{(n)}$ are drawn from a diverse distribution (e.g., uniformly or with non-degenerate correlations), the counts $\{s_\nu^{(n)}\}$ for $\nu \in \mathcal{S}$ naturally vary. This ensures that the columns of $\mathbf{S}_{\mathcal{B}_\mu}$ remain

linearly independent, as redundant patterns (e.g., fixed linear relationships between element counts) are highly unlikely under unstructured or randomized data. Moreover, if the data distribution satisfies even milder assumption that the covariance of $\mathbf{s}^{(n)}$ is positive definite (Assumption E.1), we show in Theorem E.2 that $\mathbb{P}\left( \mathrm{rank}(\mathbf{S}_{\mathcal{B}_\mu}) < |\mathcal{S}| \right) \leq e^{-\gamma |\mathcal{B}_\mu|}$ for some $\gamma \in \mathbb{R}$, meaning Assumption 4.2 is satisfied with high probability.

### 4.1. Convergence

In this section, we show that if the target function is representable by a single-layer linear self-attention model (as guaranteed by Theorem 3.1), then gradient descent on $L^{\mathrm{MSE}}(\mathbf{C}, \mathbf{W}^V)$ converges *zero training error* under mild conditions. It is stated below with proof in Appendix C.

Seeing that our main aim is studying the mutual interaction perspective, so the attention scores are the core component, which is also the core mechanism defining self-attention, we choose $d_2 = 1$ to simplify the convergence analysis. Thus, $\mathbf{W}^V$ is one dimensional which we denote as $\mathbf{w}$ in this subsection. Lastly, we denote $w_\alpha = \mathbf{x}(\alpha)^\top \mathbf{w}$ for any $\alpha \in \mathcal{S}$.

**Assumption 4.3** (Weak Realizability). The task is realizable, i.e, there exist $\mathbf{C}^*$ and $\mathbf{w}^*$ that perfectly fits the training data. That is, $\mathbf{y}^{(n)} = \left( \mathbf{X}^{(n)} \mathbf{C}^* \mathbf{X}^{(n)\top} \right) \mathbf{X}^{(n)} \mathbf{w}^* \ \ \forall n \in \mathcal{B}$.

**Theorem 4.4** (Convergence to Zero Training Error). *Let the dimensions $d = |\mathcal{S}|$ and $d_2 = 1$. Also, let the initial parameters $\mathbf{C}_{(t=0)} = \mathbf{0}$, $\langle \mathbf{x}(\alpha), \mathbf{w}_{(t=0)} \rangle \geq b > 0$, $\forall \alpha \in \mathcal{S}$. Then, under the assumptions 4.2 and 4.3, gradient flow on $L^{\mathrm{MSE}}(\mathbf{C}, \mathbf{w})$ converges to zero training error.*

The realizability assumption decomposes into two parts: (i) the task genuinely involves mutual interactions, and (ii) the data are noise-free. The second condition, analogous to fixing $d = |\mathcal{S}|$, simplifies the presentation without altering the core insight. Extending our results to noisy data is straightforward: in that setting, exact zero-error convergence is replaced by nonzero error bounds and probabilistic guarantees.[4] We leave such generalizations to future work.

### 4.2. Test Generalization

Under training data versatility and strong realizability, achieving zero training error with linear self-attention implies *perfect generalization* to new data from the same distribution. Moreover, under even milder assumptions, zero test error ensures generalization to unseen sequence *lengths*.

**Assumption 4.5** (Strong Realizability). The task is strongly realizable, meaning there exist matrices $\mathbf{C}^\dagger$ and $\mathbf{W}^{V\dagger}$ such that the model perfectly fits the underlying population dis-

---

[4]Condition (ii) mirrors the simplification in Newton's laws of motion, where measurement errors are simply neglected.

tribution at a fixed sequence length $L^*$. Specifically, we assume that $\mathbf{Y} = \left(\mathbf{X}\mathbf{C}^\dagger\mathbf{X}^\top\right)\mathbf{X}\mathbf{W}^{V\dagger}$ holds almost surely for $(\mathbf{X}, \mathbf{Y}) \sim \mathcal{P}_{\mathbf{X}\times\mathbf{Y}}^{L^*}$.

Due to Theorem 3.1 and Appendix B, we can safely assume strong realizability.

**Theorem 4.6** (Generalization). *Suppose Assumptions 4.2 and 4.5 hold, than zero training error forces $(\mathbf{C}, \mathbf{W}^V)$ to agree with $(\mathbf{C}^\dagger, \mathbf{W}^{V\dagger})$ on $(\mathbf{X}, \mathbf{Y}) \sim \mathcal{P}_{\mathbf{X}\times\mathbf{Y}}$ . That is, the solution achieving zero training error also satisfies*

$$\mathbb{E}_{(\mathbf{X}, \mathbf{Y})\sim\mathcal{P}_{\mathbf{X}\times\mathbf{Y}}^{L^*}}\left\| f_{\mathbf{C}, \mathbf{W}^V}(\mathbf{X}) - \mathbf{Y} \right\| = 0.$$

*Hence, it generalizes perfectly to new examples from the same distribution.*

See Appendix D for the proof of Theorem 4.6.

### 4.3. Out of Distribution (Length) Generalization

A unique strength of self-attention is its ability to process sequences of *variable length*. In many tasks, we might train on sequences of length $L^*$ yet hope to predict accurately on sequences of different lengths $L \neq L^*$. To state our result Theorem 4.8 (with proof in Appendix D.2) , we need the following assumption, which holds due to Theorem 3.1.

**Assumption 4.7** (Universal Realizability). The task is universally realizable, meaning there exist matrices $\mathbf{C}^{\forall L}$ and $\mathbf{W}^{V, \forall L}$ such that the model perfectly fits the population distribution for all sequence lengths. Specifically, we assume that $\mathbf{Y} = \left(\mathbf{X}\mathbf{C}^{\forall L}\mathbf{X}^\top\right)\mathbf{X}\mathbf{W}^{V, \forall L}$ holds almost surely for $(\mathbf{X}, \mathbf{Y}) \sim \mathcal{P}_{\mathcal{X}\times\mathcal{Y}}^{\forall L}$.

**Theorem 4.8** (Length Generalization). *Under the Assumptions 4.2 and 4.7, any $\mathbf{C}^\dagger, \mathbf{W}^{V\dagger}$ that generalizes to $\mathcal{P}_{\mathbf{X}\times\mathbf{Y}}^{L^*}$, must generalize to $\mathcal{P}_{\mathbf{X}\times\mathbf{Y}}^{\forall L}$.*

Building on the proof of Theorem 4.8 (in Appendix D.2), we observe a key relationship between the matrices $\mathbf{C}$ and $\mathbf{W}^V$, which underpins the model's ability to generalize. We state this formally in the following corollary:

**Corollary 4.9.** *Two sets of parameters $\left\{\mathbf{C}^1, \mathbf{W}^{1,V}\right\}$ $\left\{\mathbf{C}^2, \mathbf{W}^{2,V}\right\}$ lead to functionally equivalent linear self-attention blocks if and only if they satisfy*

$$\mathcal{T}_{\mu,k}\left(\mathbf{C}^1, \mathbf{W}^{1,V}\right) = \mathcal{T}_{\mu,k}\left(\mathbf{C}^2, \mathbf{W}^{2,V}\right), \forall\mu, k,$$

*where*

$$\mathcal{T}_{\mu,k}(\mathbf{C}, \mathbf{W}) = \sum_{\nu\in\mathcal{S}}\left(\mathbf{x}^\top(\mu)\mathbf{C}\mathbf{x}(\nu)\right)\left(\mathbf{x}^\top(\nu)\mathbf{W}_{:,k}\right).$$

*Consequently, if you apply the transformation $\mathcal{T}_{\mu,k}$ to the parameters, all of the sets of parameters that lead to functionally equivalent linear self-attentions be mapped to a specific matrix (over index $\mu, k$) that depends on the function they represent. Thus, for a specific task all length generalizing sets of parameters will lead to the same matrix under this transformation.*

### 4.4. Discussion

Our theoretical findings in this section show that a single-layer linear self-attention model not only converges to zero training error but also generalizes to both unseen data and unseen sequence lengths. In Section 7, we present controlled experiments, confirming our convergence and generalization guarantees hold.

Building on the **modular perspective** introduced in Section 3, we view a deep Transformer as a stack of self-contained blocks, each capable of shouldering a distinct subtask that emerges during gradient-based optimization. Our analysis shows that when a mutual-interaction subtask is delegated to a single-layer linear self-attention block, that block provably (i) converges to zero error for the subtask and (ii) generalizes to unseen data and longer sequences. Earlier layer-wise results support this modular viewpoint. In deep linear networks, (Shin, 2020) prove that block-coordinate (layer-by-layer) gradient descent reaches the same global optimum as full end-to-end training, provided every hidden layer is at least as wide as the input and output. For nonlinear architectures, (Zeng et al., 2019) show that cyclic block-coordinate descent converges to a stationary point at an $\mathcal{O}(1/k)$ rate, and (Akiyama, 2024) recently extend global-minimum guarantees to networks with strictly monotone activations (and, with skip connections, to modified ReLU nets). Although these results do not yet cover soft-max attention, they suggest that if each Transformer block can provably master its assigned mutual-interaction task—as established here for linear self-attention—then an alternating layer-wise schedule can, under suitable conditions, approach the performance of joint optimization. Our block-level theorem thus provides an "atomic" guarantee that future work can build on to analyze full Transformer training.

A key assumption throughout is *training data versatility* (Assumption 4.2). Intuitively, this requires each element in the domain to appear in sufficiently *diverse* contexts, ensuring the corresponding data matrix $\mathbf{S}_{\mathcal{B}_\mu}$ has full column rank. This richness is crucial for generalization, as it allows the model to learn meaningful interactions that extend beyond the training set. In realistic settings -where domain elements vary meaningfully (e.g., different agent configurations, protein sequences, or natural language tokens)- such rank deficiencies are unlikely. Consequently, data versatility ensures that the model not only fits the training data but also generalizes effectively to new distributions and sequence lengths.

## 5. Extension to HyperFeatureAttention

In the previous sections, we showed how self-attention learns pairwise interactions between entities. However, in practical scenarios, the entities $\mu \in \mathcal{S}$ are not *mono-*

*lithic.* In other words, they are composed of features. For instance, consider $\mu$ to be composed of $M \in \mathbb{Z}^+$ features, $\mu = (\mu_{\phi_1}, \ldots, \mu_{\phi_M})$, where $\mu_{\phi_i} \in \mathcal{S}_{\phi_i}$, so $\mathcal{S} = \mathcal{S}_{\phi_1} \times \ldots \times \mathcal{S}_{\phi_M}$.[5] Those features may be the (red, green, blue) components of an RGB pixel in an image or (height, weight,...) characteristics of a person in a population or something else depending on the contex. To illustrate how the *couplings* of interactions between features are crucial, let us revisit the colliding agents environment of Section 3, with modifications.[6]

**Colliding Agents Revisited.** Assume the same setting as before except now the agents are not identical. We need labels for the agents $\ell_i \in \mathcal{S}_\ell$. Thus, each agent is composed of two features $x_i = (\ell_i, \mathbf{r}_i)$. In Appendix F, we showed that for non-identical agents, it is natural to have the value function of the form

$$V_i = \sum_{j \in [L]} \left( \prod_{a \in \mathcal{A}} f_a \left( \phi_{a,i}, \theta_{a,j} \right) \right) \left( \prod_{a \in \mathcal{A}} w_a \left( \gamma_{a,j} \right) \right), \quad (3)$$

where $\phi_{a,i}$, $\theta_{aj}$, $\gamma_{a,j}$ are the corresponding features (label or position) picked depending on the scenario and $f_a, w_a$ are some functions from features to real numbers.

Here we provide a simplified illustration of how Hyper-FeatureAttention emerges from our theoretical framework; see Appendix F.1 for a detailed motivation. Seeing that $|\mathcal{S}| = |\mathcal{S}_\ell||\mathcal{S}_\mathbf{r}|$, from Theorem 3.1, a linear self-attention requires $d = \Theta(|\mathcal{S}_\ell||\mathcal{S}_\mathbf{r}|)$, so $\Theta(|\mathcal{S}_\ell|^2|\mathcal{S}_\mathbf{r}|^2)$ parameters to represent (3). However, defining new attention matrices, $\mathbf{C}^{(h,a)}$ for each $f_a$, we can represent the corresponding function only with $\Theta(|\mathcal{S}_\ell|^2)$, $\Theta(|\mathcal{S}_\mathbf{r}|^2)$, or $\Theta(|\mathcal{S}_\ell||\mathcal{S}_\ell|)$ parameters depending on which features are used for $f_a$. For example, $f(\ell_i, \ell_j)$ requires $\Theta(|\mathcal{S}_\ell|^2)$ parameters. As a result, if there are $M$ features, self-attention would require embedding dimension and number of parameters in $\Theta(\exp(M))$, while defining attention for individual $f_a$'s only require $\Theta(M)$ (see Appendix F for exact calculation).[7] This brings us to HyperFeatureAttention.

**Definition 5.1** (Linear HyperFeatureAttention of order $A \in \mathbb{Z}^+$).

$$\mathbf{HFA}^{\mathrm{lin}}(\mathbf{X}) = \left( \prod_{a \in [A]}^{\odot} \mathbf{X} \mathbf{C}^{(a)} \mathbf{X}^\top \right) \left( \prod_{a \in [A]}^{\odot} \mathbf{X} \mathbf{W}^{V,(a)} \right),$$

where $\prod^{\odot}$ is Hadamard (element-wise) product between the matrices, and $\mathbf{C}^{(a)}, \mathbf{W}^{V,(a)} \in \mathbb{R}^{d \times d}$.

---

[5]Here, we do not assume knowledge of the features, we just assume each entity have some features.

[6]For an easy transition to the following example we suggest going over Appendix B.1 -without the subsections- and Appendix B.1.3.

[7]For a more practical comparison of parameter counts with approximate represensions please refer to Remark F.1.

From the preceding discussion, Linear HyperFeatureAttention requires only $\Theta(M)$ embedding dimension and parameters to express (3). One may concern that in practice we use layers of multi-head attention, which may possibly express Eq. 3, without exponential embedding dimension. However, we showed in Remark F.6 that even two layer multihead linear self-attention cannot express (3). After all these motivations, we *formally* defined Multihead HyperFeatureAttention in Appendix F.2. Also, a detailed comparison of the new modules' memory and computational requirements versus standard self-attention appears in Appendix H.

In short, similar to how self-attention generalizes dense layers by enabling entity (token)-level interactions, HyperFeatureAttention extends Self-Attention by enabling coupling between the feature interactions. It enhances traditional self-attention by allowing attentions to be coupled, which enables model to capture coupled feature level interactions.

## 6. Extension to HyperAttention

Similar to pairwise interactions, some tasks may involve higher-order interactions, three-way four-way, n-way. Thus, in a tuple $\mathcal{X}$ we may have $\mu \in \mathcal{S}$ that is influenced by a composite function of $\nu$ and $\gamma \in \mathcal{S}$. In this case, we would need our attention scores to be able to capture interaction functions of the form $f(\mu, \nu, \gamma)$. From this need, we develop a novel generalization, named *HyperAttention* for capturing higher-order interactions, alongside (Sanford et al., 2023; Alman & Song, 2023).[8] Here we state third order and linear version, please see Appendix G for the full version.

**Definition 6.1** (Third order Linear HyperAttention).

$$A_{ij_1j_2} = \sum_{\alpha, \zeta_1, \zeta_2 \in [d]} C_{\alpha \zeta_1 \zeta_2} X_{i\alpha} X_{j_1 \zeta_1} X_{j_2 \zeta_2}$$

$$V_{j_1 j_2 \tau} = \sum_{\xi_1, \xi_2 \in [d]} X_{j_1 \xi_1} X_{j_2 \xi_2} W^V_{\xi_1 \xi_2 \tau}$$

$$\mathrm{HA}^{\mathrm{lin}}_{i\tau}(\mathbf{X}) = \sum_{1 \leq j_1 \leq j_2 \leq L} A_{ij_1j_2} V_{j_1 j_2 \tau},$$

where we **denote** $(i, j, k)$-th entry of a tensor $\mathbf{T}$ as $T_{ijk}$ and $\mathbf{C} \in \mathbb{R}^{d \times d \times d}$, $\mathbf{W}^V \in \mathbb{R}^{d \times d \times d_2}$ and Table H.

**Ternary Synergies in Multi-Agent Collaboration** Imagine a multi-agent system in which each agent's payoff depends not just on pairwise interactions with other agents, but on *three-way* synergies. For instance, suppose agent $i$ gets reward only if it forms an alliance with agents $j$ and $k$, but

---

[8]There are small differences between our definition and their ((Sanford et al., 2023; Alman & Song, 2023)) definition because the way we reached to the architecture is different. For instance, their formulation is a low-rank approximation that assumes $V_{j_1 j_2 \tau} = V^1_{j_1 \tau} V^2_{j_2 \tau}$. In contrast, our approach offers more representational capacity while maintaining comparable efficiency.

agents $j$ and $k$ does not form an alliance with each other. Formally, each agent $i$ has a discrete state $\mathcal{X}(i)$ that inclues its strategic type or coalition membership. Whenever $i$, $j$, and $k$ all share a compatible configuration of states, $i$ gains an additional bonus. Concretely, define, $f\big(\mathcal{X}(i), \mathcal{X}(j), \mathcal{X}(k)\big)$, a ternary synergy function which is nonzero only when $i$, $j$, and $k$ are all in an appropriate joint configuration (e.g. $i$ is allied with $j$ and $k$ but $j$ and $k$ are not allied). Let $w_{\mathcal{X}(j),\mathcal{X}(k)}$ be a weight that captures how the pair $(j,k)$ specifically contributes to $i$'s reward under that triple configuration. Then agent $i$'s total payoff takes the form:

$$V_{\mathcal{X}(i)} = \sum_{1 \le j < k \le L} f\big(\mathcal{X}(i), \mathcal{X}(j), \mathcal{X}(k)\big)\, w_{\mathcal{X}(j),\mathcal{X}(k)}.$$

In this setup, standard *pairwise* interactions (as in ordinary self-attention) are insufficient to capture the *triple* compatibility requirement. However, assigning each state $\mathcal{X}(\cdot)$ a suitable embedding enables encoding these ternary synergies into a higher-order extension of (2), allowing the resulting HyperAttention mechanism to learn how three agents jointly influence each other's rewards. We also provably explained how HyperAttention learns those higher order interactions in Appendix G.4.

For an additional example illustrating the representational capabilities of HyperAttention, see Appendix G.2, where we analyze the "skip-trigram bug" (Elhage et al., 2021).

A potential concern is the $\mathcal{O}(L^3)$ computational cost introduced by the final equation in Definition 6.1, which may pose efficiency challenges for long sequences. Leveraging techniques similar to those in (Katharopoulos et al., 2020; Choromanski et al., 2022), we address this issue in Appendix G.3. Also, a detailed comparison of the new modules' memory and computational requirements versus standard self-attention appears in Appendix H.

# 7. Experiments

## 7.1. Experiments for the Theories

We empirically validate our linear-SA theories, i.e. representation (Theorem 3.1), convergence (Theorem 4.4), and generalization (Theorem 4.6, Theorem 4.8) with the colliding agents environment.

**Setup: Colliding Agents on a Cylindrical Grid.** Consider $L$ *identical* agents on a cylindrical grid of size $[N] \times [N]$, with initial position vectors $\mathbf{r}_i = \begin{bmatrix} n_i^x & n_i^y \end{bmatrix} \in [N]^2$, with wrap-around in y (cylinder). Each agent is modeled as a circle of radius $R$ executes a simple "move-right" policy (one step to the right per time step if there is space to move, $n_i^x < N - 1$, otherwise stays where it is). Whenever agent $i$ collides with agent $j$ (distinct), it receives a penalty of $-1$. Our goal is to capture how each agent's final total accumulated reward, that is the value function $V_i$, depends

on the initial states. We leverage Theorem 3.1 and show how this reward structure and the final value function can be exactly represented by a single layer linear self-attention. Under this setting, the final value function for each agent can be expressed as

$$V_{\mathbf{r}_i} = -\sum_{j \neq i}^{L} \mathbb{I}\big\{\min\big(\big|n_i^y - n_j^y\big|, N - \big|n_i^y - n_j^y\big|\big) \le 2R\big\} \tag{4}$$

Seeing that the value function depends only on the $y$-coordinates, we focus our discussion on a one-dimensional case for simplicity. The extension to value functions with higher dimension dependence follows naturally and is illustrated in Appendix B.1. We trained linear self-attention, for different embeddings one-hot and sinusoidal.[9] It has an embedding dimension of $N$ and its parameters are initialized as $\mathbf{C}_{(t=0)} = \mathbf{0}$, $\langle \mathbf{x}(\alpha), \mathbf{w}_{(t=0)}\rangle$, $\forall \alpha \in \mathcal{S}$.

**One-Hot.** We first use a one-hot embedding: each position $n \in [N]$ is represented by the standard basis vector $\mathbf{e}_n \in \mathbb{R}^N$. In Appendix B.1, we demonstrate how a single-layer linear self-attention with $\mathbf{W}^V = -\mathbf{1}_N$, and

$$C_{mn} = \begin{cases} 1, & \min(|m-n|, N-|m-n|) \le 2R, \\ 0, & \text{otherwise}, \end{cases} \tag{5}$$

can *exactly* implement the value function in (4), irrespective of $L$, so all realizability assumptions are satisfied. Under these, Theorem 4.4 predicts that the training mean squared error (MSE) converges to zero, which matches our observations in practice. Furthermore, as the Theorems 4.6 and 4.8 predict, we see generalization results: negligible error ($\Theta(10^{-7})$) on test sets both when $L = 20$ and when $L \in \{2, 5, 10, 30, 40\}$ varies.

**Sinusoidal.** We next consider a sinusoidal embedding, inspired by common positional encodings in attention models (Vaswani et al., 2023; Su et al., 2023). For even $N$, the position $n_i$ of agent $i$ is mapped to

$$\mathbf{p}_i = \Big[\frac{1}{\sqrt{2}}, \ldots, \sin\Big(\frac{2\pi k}{N}n_i\Big), \cos\Big(\frac{2\pi k}{N}n_i\Big), \ldots,$$
$$\frac{1}{\sqrt{2}}\cos\Big(\frac{2\pi}{N}\Big(\frac{N}{2}-1\Big)n_i\Big), \frac{1}{\sqrt{2}}\cos\Big(\frac{2\pi}{N}\frac{N}{2}n_i\Big)\Big]^\top \in \mathbb{R}^N.$$

Note that $\mathbf{p}_i^\top \mathbf{p}_j = \frac{N}{2}\delta_{i,j}$. Due to Lemma B.4 and Threorem B.6, a single layer linear self-attention with $\mathbf{W}^{V, \forall L} = \begin{bmatrix} -\sqrt{2} & 0 & 0 & \ldots \end{bmatrix}^\top \in \mathbb{R}^{N \times 1}$ and diagonal $\mathbf{C}^{\forall L} \in \mathbb{R}^{N \times N}$, such that

$$C_{\mu\mu} = \begin{cases} \frac{4}{N}\big(2R + \frac{1}{2}\big) & \text{if } \mu = 0, \\ \frac{2}{N}\frac{\sin\big[\frac{2\pi}{N}\big(2R+\frac{1}{2}\big)\big(\frac{\mu+1}{2}\big)\big]}{\sin\big[\frac{2\pi}{N}\frac{1}{2}\big(\frac{\mu+1}{2}\big)\big]} & \text{if } \mu \text{ odd}, \\ \frac{2}{N}\frac{\sin\big[\frac{2\pi}{N}\big(2R+\frac{1}{2}\big)\big(\frac{\mu}{2}\big)\big]}{\sin\big[\frac{2\pi}{N}\frac{1}{2}\big(\frac{\mu}{2}\big)\big]} & \text{if } \mu \text{ even}, \end{cases}$$

---

[9] Under $L = 20$, $N = 360$, and $B = 100k$.

| Model | Order | Perplexity $\downarrow$ |
|---|---|---|
| Self-Attention | – | 62.28 |
| HyperFeatureAttention | 4 | 60.22 |
| HyperAttention | 3 | 51.26 |
| HyperAttention (no sharing) | 3 | 48.50 |

*Table 1.* **1-layer, 1-head, 256-token bencmark** All models share GPT3-small hyper-parameters ($d_{\text{model}}$=768, vocab size 50257, and identical optimiser settings). Training: 28 k iterations, batch 0.16M tokens, cosine LR schedule with warm-up ($\eta_{\max}$=6$\times$10$^{-4}$, $\eta_{\min}$=0.1 $\eta_{\max}$). Per-epoch validation perplexities are Gaussian-smoothed; final values are reported. SA and HFA have identical parameter counts and $\Theta(L^2)$ compute; HA retains the same parameter count but, without the low-rank trick, scales as $\Theta(L^3)$. HA (no sharing) has slightly more parameters explained in Appendix G.

| Model | Heads / Order | Perplexity $\downarrow$ |
|---|---|---|
| SA | 3/2 | 28.70 |
| HFA | 3/3 | 27.97 |
| HFA$_{\text{v2}}$ | 4/(2$\times$2, 2$\times$3) | **27.75** |

*Table 2.* **3-layer, 1024-token benchmark.** Hyper-parameters and optimiser settings match Table 1. HFA incurs $<$0.1% extra compute over SA due to attention products. The hybrid HFA$_{\text{v2}}$ (2 SA heads + 2 order-3 HFA heads) attains the best perplexity.

can represent the value functions at Eq. 4 exactly, for any $L$, which is plotted at Figure 1. The entries of **C** are simply Fourier transform of the interaction function, which is an artifact of the sinusoidal embedding. Seeing that the same assumptions are satisfied for sinusoidal embedding, we observe the same convergence and generalization results for sinusoidal embedding.

Although the results match the theorems, the learned parameters do not overlap with the parameters we originally devised, especially for sinusoidal embedding, seen in Figure 2 -these learned parameters lack an intuitive and easy interpretation. However, as discussed in Corollary 4.9, this outcome is a natural consequence of the generalization theories (Theorems 4.6 and 4.8). Therefore, we focus on the matrices we get under the nontrivial transformation explained in 4.9. As shown in Figures 3 and 4, these matrices are indistinguishable from the theoretical counterparts. With only $\Theta\left(10^{-4}\right)$ mean square distance between their entries. Thus, the parameters we get from training are functionally equivalent to the length generalizing parameters we devised.

### 7.2. Experiments for the Novel Modules

To verify that the theoretical benefits of our layers translate into practice, we ran two small-scale next-token-prediction benchmarks on OpenWebText under hyperparameter parity with GPT3-small, see Tables 1 and 2. All models share the same embedding, MLP and optimiser settings (as in GPT3); HFA has the same $\Theta(L^2)$ time/space cost as SA, while HA (evaluated without low-rank tricks) scales as $\Theta(L^3)$ in computation. The tables report the final validation perplexities. The results support our claim that the flexibility with richer interaction blocks improve language-model quality.

## 8. Conclusion

We introduced a unifying theoretical framework that interprets tokens in self-attention as *interacting entities* and showed that a single layer linear self-attention mechanism can capture pairwise interactions across a broad range of domains. Our analysis clarified how self-attention learns and generalizes these interaction functions without relying on domain-specific assumptions. In particular, we proved that (i) such models converge under simple gradient-based training, (ii) they generalize to both unseen data, and variable sequence lengths when the training set is sufficiently diverse

Building on our theories for self-attention, we introduced novel *HyperFeatureAttention* and *HyperAttention* modules that extend self-attention's abilities to capturing couplings of different feature level interactions and higher-order interactions among multiple entities. Beyond the theoretical guarantees, our language-modeling benchmarks show that both HFA and HA consistently achieve lower perplexity than standard attention, confirming that their enhanced interaction capacity translates into tangible performance gains.

Taken together, our theoretical and empirical findings establish self-attention -and its *HyperFeatureAttention* and *HyperAttention* variants- as efficient learners of complex entity interactions. Our language-modeling experiments confirm the feasibility of these modules, showing consistent perplexity reductions under matched compute budgets. A natural next step is to stress-test HFA and HA at larger scales and in applications such as multi-agent control, protein design, and the other scenarios motivated in this paper. We hope this interaction-centric lens spurs the creation of even more adaptable attention architectures and motivates deeper analyses of softmax attention, stacked transformer layers, and emerging attention mechanisms.

## Acknowledgements

This research was supported by NSF Grants 2154171, CA-REER Award 2339112, CMU CyLab Seed Funding.

## Impact Statement

This work advances the theoretical understanding of self-attention and its generalizations -HyperFeatureAttention and HyperAttention- by providing a unifying framework for learning complex interactions. Our insights could benefit diverse applications, including NLP, computer vision, multi-agent systems, and scientific modeling. Improved interpretability and training efficiency may enhance model reliability and user trust.

While theoretical, our findings could indirectly impact real-world risks, such as biased decision-making or misuse in misinformation and surveillance. Mitigating such risks requires policy measures rather than changes to fundamental theory. Additionally, optimizing attention mechanisms may contribute to more efficient models, reducing the environmental footprint of large-scale training. Overall, this work presents no unique societal risks beyond typical concerns in machine learning research.

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

## A. Notation and Definitions

Consider a discrete domain (or "vocabulary") $\mathcal{S} = \{\alpha, \beta, \gamma, \omega, \dots\}$ with cardinality $|\mathcal{S}|$. In our setting we have tuples of $L$ elements (sequences of length $L$) from the domain, denoted by $\mathcal{X}$ and entries of which are uniquely indexed by the integers in $[L]$. Here, $[i]$ denotes the set $\{0, 1, \dots, i-1\}$ for any $i \in \mathbb{Z}^+$, positive integer. We define the function, $\mathcal{X} : [L] \to \mathcal{S}$ assign each index $i$ to the corresponding element $\mathcal{X}(i) \in \mathcal{S}$. Thus, we can write $\mathcal{X} = (\mathcal{X}(0), \mathcal{X}(1), \dots, \mathcal{X}(L-1))$, which is distributed according to $\mathcal{X} \sim \mathcal{D}$. We also have tuples $\mathcal{Y}$, length of which depends on the specific task we have, with elements from a corresponding relatively small set $\mathcal{S}_{\mathcal{Y}}$. The tuples are distributed according to a task distribution $(\mathcal{X}, \mathcal{Y}) \sim \mathcal{D}_{\mathcal{X} \times \mathcal{Y}}$. When needed we use $\mathcal{D}^L$ and $\mathcal{D}_{\mathcal{X} \times \mathcal{Y}}^L$ to denote the same distributions for specifically tuple length of $L$. Similarly, $\mathcal{D}^{\forall L}$ means the distribution covering all possible lengths. In our training dataset, we have $B$ such pairs that $(\mathcal{X}^{(n)}, \mathcal{Y}^{(n)}) \sim \mathcal{D}_{\mathcal{X} \times \mathcal{Y}}$, which are uniquely indexed by the elements in the set $\mathcal{B} = [B]$. We also define, $\mathcal{B}_\mu$ as the set of training indices that contain the element $\mu$. We denote the number of times an element $\mu$ appears in tuple $\mathcal{X}^{(n)}$ as $s_\mu^{(n)}$. We define the corresponding

$$\mathbf{s}^{(n)} = \begin{bmatrix} s_\alpha^{(n)} & s_\beta^{(n)} & \dots \end{bmatrix}^\top, \mathbf{S} = \begin{bmatrix} \mathbf{s}^{(1)} & \mathbf{s}^{(2)} & \dots & \mathbf{s}^{(B)} \end{bmatrix}^\top \in \mathbb{R}^{B \times N} \text{ and } \mathbf{S}_{\mathcal{B}_\mu} = \begin{bmatrix} \dots & \mathbf{s}^{(n)} & \dots \end{bmatrix}_{n \in \mathcal{B}_\mu}^\top \text{ matrices.}$$

In order to train a neural network, we map each element of $\mathcal{S}$ to a $d$-dimensional embedding space via a function $\mathbf{x} : \mathcal{S} \to \mathbb{R}^d$ and each element of $\mathcal{S}_{\mathcal{Y}}$ to a corresponding vector or a scalar depending on the task. We can now define the domain embedding matrix

$$\mathbf{B} = \begin{bmatrix} \mathbf{x}^\top(\alpha) \\ \mathbf{x}^\top(\beta) \\ \vdots \end{bmatrix}. \tag{6}$$

Additionally, we stack the embeddings of the elements in $\mathcal{X}$ and $\mathcal{Y}$ as rows of $\mathbf{X} \in \mathbb{R}^{L \times d}$ and $\mathbf{Y} \in \mathbb{R}^{L \times d_2}$ matrices, which we say are distributed according to $(\mathbf{X}, \mathbf{Y}) \sim \mathcal{P}_{\mathbf{X} \times \mathbf{Y}}^L$. For training sample $n$ it can be written as,

$$\mathbf{X} = \begin{bmatrix} \mathbf{x}^\top(\mathcal{X}(0)) \\ \mathbf{x}^\top(\mathcal{X}(1)) \\ \vdots \\ \mathbf{x}^\top(\mathcal{X}(L-1)) \end{bmatrix} = \begin{bmatrix} \mathbf{x}_0^\top \\ \mathbf{x}_1^\top \\ \vdots \\ \mathbf{x}_{L-1}^\top \end{bmatrix},$$

where in the last equality we denote the embedding of the $i$-th element as $\mathbf{x}_i := \mathbf{x}(\mathcal{X}(i)) \in \mathbb{R}^d$, to reduce the notational cluttering.

We denote matrices and vectors by bold characters. Let $\mathbf{M} \in \mathbb{R}^{d_1 \times d_2}$, $\mathbf{w} \in \mathbb{R}^d$ be any matrix and vector respectively. We denote $k$-th row of a $\mathbf{M}$ as $\mathbf{M}_{k,:}$ or $\mathbf{m}_k$, similarly we denote $k$-th column as $\mathbf{M}_{:,k}$ or $\mathbf{m}^k$. We denote $k$-th entry of $\mathbf{m}$ as $m_k$ and $(k, l)$-th entry of $\mathbf{M}$ as $M_{kl}$. Similarly, if a vector is defined in terms of blocks we explicity state it and denote each the blocks as $\mathbf{w} = \begin{bmatrix} \mathbf{w}_1^\top & \mathbf{w}_2^\top & \dots \end{bmatrix}^\top$. The same notations naturally extend to tensors, e.g., we denote $(i, j, k, \dots)$-th entry of a tensor $\mathbf{T} \in \mathbb{R}^{d \times d \times d \times \dots}$ as $T_{ijk\dots}$.

In addition we denote $i^{th}$ eigenvalue of a matrix $\mathbf{M}$ as $\lambda_i(\mathbf{M})$, where $\lambda_1(\mathbf{M}) \geq \lambda_2(\mathbf{M}) \geq \dots$. Similarly we denote the $i^{th}$ singular value as $\sigma_i(\mathbf{M})$. Lastly we have some operators diagonalization

$$\text{diag}(\mathbf{w}) = \begin{pmatrix} w_1 & & \\ & \ddots & \\ & & w_d \end{pmatrix}$$

and Kronecker product of two matrices $A \in \mathbb{R}^{m \times n}$, $B \in \mathbb{R}^{p \times q}$

$$A \otimes B = \begin{bmatrix} a_{11}B & a_{12}B & \dots & a_{1n}B \\ a_{21}B & a_{22}B & \dots & a_{2n}B \\ \vdots & \vdots & \ddots & \vdots \\ a_{m1}B & a_{m2}B & \dots & a_{mn}B \end{bmatrix},$$

where each entry $a_{ij}$ in $A$ is multiplied by the entire matrix $B$, which results in a matrix of size $(mp) \times (nq)$.

## B. Representation Abilities of Linear Self-Attention

*Proof of Theorem 3.1 (Representation Ability of Linear Self Attention).* We first show **sufficiency** of $d = |\mathcal{S}|$. We can define an *orthonormal* embedding $\mathbf{x} : \mathcal{S} \to \mathbb{R}^N$ such that

$$\mathbf{x}(\alpha)^\top \mathbf{x}(\beta) = \delta_{\alpha,\beta} \quad \forall\, a, b \in \mathcal{S},$$

where $\delta_{a,b}$ is the Kronecker delta. Recall that $(\mathcal{X}, \mathcal{Y}) \sim \mathcal{D}_{\mathcal{X} \times \mathcal{Y}}$ are tuples whose elements are indexed from the set $[L]$. Also, recall that we let $s : [L] \to \mathcal{S}$ map each index $i$ to its corresponding symbol $\mathcal{X}(i) \in \mathcal{S}$. We arrange the embeddings $\mathbf{x}(\mathcal{X}(i))$ row-wise into $\mathbf{X} \in \mathbb{R}^{L \times |\mathcal{S}|}$.

**Goal.** We wish to represent the family of functions

$$\mathbf{y}_{\mathcal{X}(i)} = \sum_{j \in [L]} f\big(\mathcal{X}(i), \mathcal{X}(j)\big) \, \mathbf{w}_{\mathcal{X}(j)}, \quad \text{for each } i \in [L],$$

using a *single-layer linear self-attention* of dimension $d = |\mathcal{S}|$. Here, $f : \mathcal{S} \times \mathcal{S} \to \mathbb{R}$ measures *how strongly* one entity affects another, and $\mathbf{w}_{\mathcal{X}(j)} \in \mathbb{R}^{d_2}$ encodes *how* that influence is expressed (e.g., a direction vector or contribution to a subsequent feature).

Recall that a single-layer linear self-attention can be written as

$$\mathbf{Attn}^{\text{lin}}(\mathbf{X}) = \big(\mathbf{X}\,\mathbf{C}\,\mathbf{X}^\top\big)\,\mathbf{X}\,\mathbf{W}^V,$$

where $\mathbf{C} \in \mathbb{R}^{d \times d}$ captures the *attention scores* (keys $\times$ queries) in a linear setting, $\mathbf{W}^V \in \mathbb{R}^{d \times d_2}$ represents the values transformation.

**Constructing C and V.** We define $\mathbf{C}$ and $\mathbf{W}^V$ to satisfy:

$$\mathbf{x}(\alpha)^\top \mathbf{C}\,\mathbf{x}(\beta) = f(\alpha, \beta), \quad \text{and} \quad \mathbf{x}(a)^\top \mathbf{W}^V = \mathbf{w}_a.$$

Such $\mathbf{C}$ and $\mathbf{V}$ exist because, $\{\mathbf{x}(a)\}_{a \in \mathcal{S}}$ forms an orthonormal basis in $\mathbb{R}^d$, due to $d = |\mathcal{S}|$. Concretely, $\mathbf{C}$ can be chosen so that its bilinear form on basis vectors $\mathbf{x}(\alpha), \mathbf{x}(\beta)$ equals $f(\alpha, \beta)$. Likewise, $\mathbf{W}^V$ can be chosen so that $\mathbf{x}(\alpha)$ maps to $\mathbf{w}_a$.

**Verification.** Given a sequence of length $L$, the matrix $\mathbf{X} \in \mathbb{R}^{L \times |\mathcal{S}|}$ is

$$\mathbf{X} = \begin{bmatrix} \mathbf{x}\big(\mathcal{X}(0)\big)^\top \\ \mathbf{x}\big(\mathcal{X}(1)\big)^\top \\ \vdots \\ \mathbf{x}\big(\mathcal{X}(L-1)\big)^\top \end{bmatrix}.$$

Thus,

$$\mathbf{X}\,\mathbf{C}\,\mathbf{X}^\top = \begin{bmatrix} f\big(\mathcal{X}(0), \mathcal{X}(0)\big) & \cdots & f\big(\mathcal{X}(0), \mathcal{X}(L-1)\big) \\ \vdots & \ddots & \vdots \\ f\big(\mathcal{X}(L-1), \mathcal{X}(0)\big) & \cdots & f\big(\mathcal{X}(L-1), \mathcal{X}(L-1)\big) \end{bmatrix},$$

and

$$\mathbf{X}\,\mathbf{W}^V = \begin{bmatrix} \mathbf{w}_{\mathcal{X}(0)}^\top \\ \mathbf{w}_{\mathcal{X}(1)}^\top \\ \vdots \\ \mathbf{w}_{\mathcal{X}(L-1)^\top} \end{bmatrix}.$$

Multiplying these terms in the linear self-attention expression yields, for each row $i \in [L]$:

$$\big[\big(\mathbf{X}\,\mathbf{C}\,\mathbf{X}^\top\big)\,\mathbf{X}\,\mathbf{W}^V\big]_{i,:} = \sum_{j=0}^{L-1} f\big(\mathcal{X}(i), \mathcal{X}(j)\big)\,\mathbf{w}_{\mathcal{X}(j)}.$$

This matches the desired function $\mathbf{F}_i = \sum_{j \in [L]} f(\mathcal{X}(i), \mathcal{X}(j)) \mathbf{w}_{\mathcal{X}(j)}$. Hence, a single-layer linear self-attention with dimension $|\mathcal{S}|$ can represent *any* pairwise interaction function of this form.

Now, we focus on **necessity** of $d \geq |\mathcal{S}|$. Remember, $f : \mathcal{S} \times \mathcal{S} \to \mathbb{R}$ represent any pairwise interaction function. For fixed embeddings $\{\mathbf{x}(a) \in \mathbb{R}^d\}_{a \in \mathcal{S}}$ and any matrix $\mathbf{C} \in \mathbb{R}^{d \times d}$, the product $\mathbf{X} \mathbf{C} \mathbf{X}^\top$ is at most rank $d$. Formally, if we stack all dictionary embeddings $\mathbf{x}(a)$ into a matrix $\mathbf{X} \in \mathbb{R}^{|\mathcal{S}| \times d}$, then

$$\text{rank}(\mathbf{X} \mathbf{C} \mathbf{X}^\top) \leq d.$$

Thus, any pairwise function $f(a, b) = \mathbf{x}(a)^\top \mathbf{C} \mathbf{x}(b)$ produces an $|\mathcal{S}| \times |\mathcal{S}|$ matrix with rank at most $d$.

Consider the pairwise function $f^*(a, b)$ whose corresponding matrix $F^* \in \mathbb{R}^{|\mathcal{S}| \times |\mathcal{S}|}$ is full-rank, i.e., $\text{rank}(F^*) = |\mathcal{S}|$. For example, take $F^* = \mathbf{I}_{|\mathcal{S}|}$ (the identity matrix), corresponding to $f^*(a, b) = \delta_{a,b}$ (the Kronecker delta).

If $d < |\mathcal{S}|$, then any matrix $\mathbf{X} \mathbf{C} \mathbf{X}^\top$ has rank $\leq d$ and hence cannot equal $F^*$, which has $\text{rank}(F^*) = |\mathcal{S}| > d$. Therefore, the self-attention mechanism cannot represent $f^*(a, b)$ exactly when $d < |\mathcal{S}|$.

Since a single-layer linear self-attention mechanism of dimension $d < |\mathcal{S}|$ cannot represent the full-rank function $f^*$, it follows that $d \geq |\mathcal{S}|$ is *necessary* to exactly represent *all* pairwise functions $f(a, b)$. $\qquad \square$

As a consequence of the above Theorem, it requires $\mathcal{O}(|\mathcal{S}|^2)$ parameters to exactly represent any such sequence to sequence interaction mapping seen in (2). Further, the output dimension, $d_2$, plays a largely peripheral role in the self-attention mechanism's ability to capture pairwise interactions. Its primary function is to align with the desired output representation—whether it be the dimensionality of subsequent features or the final embedding size —without restricting the expressiveness of the attention scores. The latter is dictated by setting $d = |\mathcal{S}|$, ensuring that all necessary pairwise interactions $f(\alpha, \beta)$ can be effectively captured. Thus, $d_2$ should not be viewed as a bottleneck; it is simply a design choice for how the represented interactions are ultimately projected or stored.

**Corollary B.1** (Orthogonal Transforms Preserve Representational Capacity). *Under $d = |\mathcal{S}|$, let $\mathbf{x}(\alpha)$ be the orthonormal embedding constructed therein. For any orthogonal matrix $\mathbf{Q} \in \mathbb{R}^{d \times d}$, define the new embedding $\mathbf{x}'(\alpha) = \mathbf{Q} \mathbf{x}(\alpha)$. Then the same pairwise function $f(\alpha, \beta)$ can be represented exactly by redefining*

$$\mathbf{C}' = \mathbf{Q} \mathbf{C} \mathbf{Q}^\top \quad \text{and} \quad \mathbf{W}'^V = \mathbf{Q} \mathbf{W}^V.$$

*Hence, any orthonormal transformation of $\mathbf{x}(\alpha)$ leaves the representational capacity of the single-layer linear self-attention unchanged. Since the construction in the Proof of Threorem 3.1 (in Appendix B) relies only on the orthogonality of $\mathbf{x}(\alpha)$, the same argument applies to any embedding $\mathbf{x}'(\alpha) = \mathbf{Q} \mathbf{x}(\alpha)$ with $\mathbf{Q} \in \mathbb{R}^{d \times d}$ orthogonal. Thus, representational capacity is invariant to the choice of orthonormal basis.*

*Proof of Theorem 3.2 (Efficiency of Linear Self Attention).* We count the minimum number of parameters needed for a fully connected network.

**Step 1:** Because $f(\alpha, \beta)$ can be chosen arbitrarily for each of the $|\mathcal{S}|^2$ ordered pairs $(\alpha, \beta)$, the model must be able to realize at least $|\mathcal{S}|^2$ independent scalar parameters to represent $f$ itself.

**Step 2:** For a fixed position $i$, the quantity

$$\mathbf{y}_{\mathcal{X}(i)} = \sum_{j=1}^{L} f(\mathcal{X}(i), \mathcal{X}(j)) \mathbf{w}_{\mathcal{X}(j)}$$

sums over $j = 1, \ldots, L$. Although there may be some degeneracies as $f(\mathcal{X}(i), \mathcal{X}(j)) \mathbf{w}_{\mathcal{X}(j)} = f(\mathcal{X}(i), \mathcal{X}(k)) \mathbf{w}_{\mathcal{X}(k)}$ for some $j, k$, or $f(\mathcal{X}(i), \mathcal{X}(j)) \mathbf{w}_{\mathcal{X}(j)} + f(\mathcal{X}(i), \mathcal{X}(k)) \mathbf{w}_{\mathcal{X}(k)} = (f(\mathcal{X}(i), \mathcal{X}(m)) \mathbf{w}_{\mathcal{X}(m)} + f(\mathcal{X}(i), \mathcal{X}(l)) \mathbf{w}_{\mathcal{X}(l)}$, that reduces the total number of parameters, we analyze the most general case. In the most general case, each $f(\mathcal{X}(i), \mathcal{X}(j)) \mathbf{w}_{\mathcal{X}(j)}$ in the summation should be calculated separately. Seeing that there are $L$ such terms in the summation, a fully connected network with no build-in parameter sharing cannot collapse these $L$ terms at no cost. Each summand could be different and must be learned with its own parameters. Hence, even for representing one output $\mathbf{y}_{\mathcal{X}(i)}$, the network needs $\Omega(L \cdot |\mathcal{S}|^2)$.

**Step 3:** The network must simultaneously produce $\mathbf{y}_{\mathcal{X}(1)}, \ldots, \mathbf{y}_{\mathcal{X}(l)}$. Without additional structure or shared weights across output positions, a fully connected network pays a separate parameter cost for each output $\mathbf{y}_{\mathcal{X}(i)}$.

Therefore, any linear fully connected network that realizes all such pairwise interaction mappings with zero error must have at least $\Omega(L^2 |\mathcal{S}|^2)$ parameters. $\qquad\square$

**Theorem B.2** (Approximate Representation Abilities of Single-Layer Linear Self-Attention). *Recall $\mathcal{S}$ is a finite set and $f : \mathcal{S} \times \mathcal{S} \to \mathbb{R}$ is any pairwise interaction function. For each symbol $\mu \in \mathcal{S}$, recall that $\mathbf{w}_\mu \in \mathbb{R}^{d_2}$ is (value) representation. Suppose the embedding dimension is less than the domain size, $d \leq |\mathcal{S}|$. Then, there exist an embedding $\mathbf{x} : \mathcal{S} \to \mathbb{R}^d$, and parameters $\mathbf{C} \in \mathbb{R}^{d \times d}$ and $\mathbf{W}^V \in \mathbb{R}^{d \times d_2}$ for a single-layer linear self-attention mechanism such that, for* any *sequence $s : [L] \to \mathcal{S}$, the output of the linear self-attention block*

$$\mathbf{SA}^{\text{lin}}(\mathbf{X}) = \left(\mathbf{X}\mathbf{C}\mathbf{X}^\top\right)\mathbf{X}\mathbf{W}^V, \tag{7}$$

*approximates the "pairwise-sum" mapping*

$$\mathbf{y}_{\mathcal{X}(i)} = \sum_{j=0}^{L-1} f\left(\mathcal{X}(i), \mathcal{X}(j)\right) \mathbf{w}_{\mathcal{X}(j)}, \tag{8}$$

*such that for all, $i = 0, 1, \ldots, L-1$,*

$$\|\mathbf{SA}^{lin}_{i,:}(\mathbf{X}) - \mathbf{y}_{\mathcal{X}(i)}\|_2 \leq \begin{cases} \zeta_1 L \sigma_{d+1}(\mathbf{F}), & d_2 \leq d < |\mathcal{S}| \\ \zeta_1 L \sigma_{d+1}(\mathbf{F}) + \zeta_2 L \sqrt{\sum_{i=d+1}^{d_2} \sigma_i^2(\mathbf{W})}, & d < d_2 < |\mathcal{S}| \\ \zeta_1 L \sigma_{d+1}(\mathbf{F}) + \zeta_2 L \sqrt{\sum_{i=d+1}^{|\mathcal{S}|} \sigma_i^2(\mathbf{W})}. & d < |\mathcal{S}| \leq d_2, \end{cases}$$

*where $\zeta_1 = \max_{\mu \in \mathcal{S}} \|\mathbf{w}_\mu\|_2$, $\zeta_2 = \max_{\mu, \nu \in \mathcal{S}} \left|\tilde{C}_{\mu\nu}\right|$ and $\sigma_i(\mathbf{F})$ is $i$-th singular value of $\mathbf{F}$ such that $\sigma_1(\mathbf{F}) \geq \sigma_2(\mathbf{F}) \geq \cdots \geq \sigma_{|\mathcal{S}|}(\mathbf{F})$.*

*Proof.* Recall $\mathbf{B}$ is the embedding matrix seen in (6). Form an $|\mathcal{S}| \times |\mathcal{S}|$ matrix $\mathbf{F}$ such that $F_{\mu\nu} = f(\mu, \nu) \quad \forall \mu, \nu \in \mathcal{S}$. We have $\mathbf{C} \in \mathbb{R}^{d \times d}$, so

$$\tilde{\mathbf{C}} := \mathbf{B}\mathbf{C}\mathbf{B}^T \in \mathbb{R}^{|\mathcal{S}| \times |\mathcal{S}|}$$

is an arbitrary matrix of rank at most $d$. Similarly, let $\mathbf{W} \in \mathbb{R}^{|\mathcal{S}| \times d_2}$ such that $\mathbf{W}_{\mu,:} = \mathbf{w}_\mu$, and $\tilde{\mathbf{W}}^V := \mathbf{B}\mathbf{W}^V \in \mathbb{R}^{|\mathcal{S}| \times d_2}$. Thus, for each $i \in [L]$, we can write,

$$\mathbf{y}_{\mathcal{X}(i)} = \sum_{j \in [L]} f\left(\mathcal{X}(i), \mathcal{X}(j)\right) \mathbf{w}_{\mathcal{X}(j)} = \sum_{j \in [L]} F_{\mathcal{X}(i), \mathcal{X}(j)} \mathbf{W}_{\mathcal{X}(j),:} \tag{9}$$

As $d \leq |\mathcal{S}|$ and we can freely choose the embedding, we can choose the embedding such that the embedding base is orthonormal, i.e. $\mathbf{B}^\top \mathbf{B} = \mathbf{I}$, we can write

$$\mathbf{SA}^{\text{lin}}_{i,:} = \mathbf{X}_{i,:}\mathbf{C}\mathbf{X}^\top\mathbf{X}\mathbf{W}^V = \mathbf{X}_{i,:}\mathbf{B}^\top\mathbf{B}\mathbf{C}\mathbf{B}^\top\mathbf{B}\mathbf{X}^\top\mathbf{X}\mathbf{B}^\top\mathbf{B}\mathbf{W}^V = \sum_{j=0}^{L-1} \tilde{C}_{\mathcal{X}(i),\mathcal{X}(j)} \tilde{\mathbf{W}}^V_{\mathcal{X}(j),:} \tag{10}$$

We start the approximation by writing the singular value decomposition of $\mathbf{F}$ as

$$\mathbf{F} = \sum_{i=1}^{|\mathcal{S}|} \sigma_i(\mathbf{F}) u_i v_i^\top,$$

where $\sigma_i(\mathbf{F})$ is $i$-th singular value of $\mathbf{F}$ such that $\sigma_1(\mathbf{F}) \geq \sigma_2(\mathbf{F}) \geq \cdots \geq \sigma_{|\mathcal{S}|}(\mathbf{F})$. We can set $\tilde{\mathbf{C}}$ to

$$\tilde{\mathbf{C}} = \sum_{i=1}^{d} \sigma_i(\mathbf{F}) u_i v_i^\top,$$

which satisfies the only constraint we have on $\tilde{\mathbf{C}}$, that it is of rank $d$. Consequently,

$$\mathbf{F} - \tilde{\mathbf{C}} = \sum_{i=d+1}^{|\mathcal{S}|} \sigma_i(\mathbf{F}) u_i v_i^\top$$

From classical linear algebra we get

$$\max_{ij} |F_{i,j} - \tilde{C}_{i,j}| \le ||\mathbf{F} - \tilde{\mathbf{C}}||_2 = \sigma_{d+1}(\mathbf{F}). \tag{11}$$

We can choose $\tilde{\mathbf{W}}^V$ to minimize $\|\tilde{\mathbf{W}}^V - \mathbf{W}\|_2$. Note that the only constraint on $\tilde{\mathbf{W}}^V$ is it of rank at most $\min(d, d_2)$, because $\tilde{\mathbf{W}}^V = \mathbf{B}\mathbf{W}^V$ where $\mathbf{W}^V \in \mathbb{R}^{d \times d_2}$ can be freely chosen and $\mathbf{B} \in \mathbb{R}^{|\mathcal{S}| \times d}$ is full column rank. Therefore, there exists a $\tilde{\mathbf{W}}^V$ such that $\|\tilde{\mathbf{W}}^V - \mathbf{W}\|_2$ is upper bounded by the $\min(d, d_2) + 1$'st largest singular value of $\mathbf{W}$. Then, using the identity $\|\mathbf{A}\|_2 \le \|\mathbf{A}\|_2$ for any $\mathbf{A}$ matrix,

$$\sqrt{\max_i \sum_j (\tilde{W}_{ij} - W_{ij})^2} \le \|\tilde{\mathbf{W}}^V - \mathbf{W}\|_2,$$

we have,

$$\sqrt{\max_i \sum_j (\tilde{W}_{ij}^V - W_{ij})^2} \le \begin{cases} 0, & d_2 \le d < |\mathcal{S}| \\ \sqrt{\sum_{i=d+1}^{d_2} \sigma_i^2(\mathbf{W})}, & d < d_2 < |\mathcal{S}| \\ \sqrt{\sum_{i=d+1}^{|\mathcal{S}|} \sigma_i^2(\mathbf{W})}. & d < |\mathcal{S}| \le d_2 \end{cases} \tag{12}$$

Returning to Equations 9 and 10, the error can be written as

$$\mathbf{SA}_{i,:}^{\mathrm{lin}} - \mathbf{y}_{\mathcal{X}(i)} = \sum_{j=0}^{L-1} \left\{ \tilde{C}_{\mathcal{X}(i),\mathcal{X}(j)} \tilde{\mathbf{W}}_{\mathcal{X}(j),:}^V - F_{\mathcal{X}(i),\mathcal{X}(j)} \, \mathbf{W}_{\mathcal{X}(j),:} \right\}$$

$$= \sum_{j=0}^{L-1} \left\{ \left( \tilde{C}_{\mathcal{X}(i),\mathcal{X}(j)} - F_{\mathcal{X}(i),\mathcal{X}(j)} \right) \mathbf{W}_{\mathcal{X}(j),:} + \tilde{C}_{\mathcal{X}(i),\mathcal{X}(j)} \left( \tilde{\mathbf{W}}_{\mathcal{X}(j),:}^V - \mathbf{W}_{\mathcal{X}(j),:} \right) \right\}$$

$$\|\mathbf{SA}_{i,:}^{\mathrm{lin}} - \mathbf{y}_{\mathcal{X}(i)}\|_2 \le \sum_{j=0}^{L-1} \left\{ \left| \tilde{C}_{\mathcal{X}(i),\mathcal{X}(j)} - F_{\mathcal{X}(i),\mathcal{X}(j)} \right| \|\mathbf{W}_{\mathcal{X}(j),:}\|_2 + \left| \tilde{C}_{\mathcal{X}(i),\mathcal{X}(j)} \right| \left\| \tilde{\mathbf{W}}_{\mathcal{X}(j),:}^V - \mathbf{W}_{\mathcal{X}(j),:} \right\|_2 \right\}$$

Letting $\zeta_1 = \max_{\mu \in \mathcal{S}} \|\mathbf{w}_\mu\|_2$, $\zeta_2 = \max_{\mu, \nu \in \mathcal{S}} \left| \tilde{C}_{\mu\nu} \right|$ substituting Equations 11 and 12,

$$\|\mathbf{SA}_{i,:}^{\mathrm{lin}} - \mathbf{y}_{\mathcal{X}(i)}\|_2 \le \sum_{j=0}^{L-1} \sigma_{d+1}(\mathbf{F}) \|\mathbf{W}_{\mathcal{X}(j),:}\|_2 + \left| \tilde{C}_{\mathcal{X}(i),\mathcal{X}(j)} \right| \left\| \tilde{\mathbf{W}}_{\mathcal{X}(j),:}^V - \mathbf{W}_{\mathcal{X}(j),:} \right\|_2$$

$$\le \begin{cases} \zeta_1 L \sigma_{d+1}(\mathbf{F}), & d_2 \le d < |\mathcal{S}| \\ \zeta_1 L \sigma_{d+1}(\mathbf{F}) + \zeta_2 L \sqrt{\sum_{i=d+1}^{d_2} \sigma_i(\mathbf{W})}, & d < d_2 < |\mathcal{S}| \\ \zeta_1 L \sigma_{d+1}(\mathbf{F}) + \zeta_2 L \sqrt{\sum_{i=d+1}^{|\mathcal{S}|} \sigma_i(\mathbf{W})}, & d < |\mathcal{S}| \le d_2 \end{cases}$$

$\square$

**Justification for the $d = |\mathcal{S}|$ Assumption.** A common concern may arise from our theoretical results, which assume the embedding dimension $d$ equals the domain size $|\mathcal{S}|$. While this may seem restrictive for large vocabularies, this assumption is well-motivated for the following reasons:

- **Clarifying the Core Mechanism and Simplifying Analysis.** Our primary goal is to study *how self-attention models interactions between entities*, independent of embedding dimension constraints. Setting $d = |\mathcal{S}|$ ensures an

exact orthonormal representation, making both representational power and training dynamics fully transparent. This choice highlights the fundamental ability of self-attention to encode all pairwise interactions while simplifying the theoretical analysis without losing generality. By removing extraneous complexities, we preserve the core insights into representation, convergence, and generalization.

- **Establishing a Natural Theoretical Baseline.** The assumption $d = |\mathcal{S}|$ serves as an idealized yet expressive starting point in theoretical analysis, providing a *one-to-one mapping from domain elements to embeddings*. This eliminates unnecessary confounding factors, allowing a clean study of self-attention's structural properties. Future work can systematically extend these results to cases where $d < |\mathcal{S}|$, exploring compressed or approximate embeddings.

- **Scalability to Lower Dimensions.** Although we analyze $d = |\mathcal{S}|$, our insights extend to $d < |\mathcal{S}|$ through approximate orthonormal embeddings. Techniques such as *random projections* (e.g., via the Johnson–Lindenstrauss lemma) allow for efficient lower-dimensional embeddings while preserving essential properties. Thus, our results remain applicable in practical settings with small approximation errors.

In summary, the $d = |\mathcal{S}|$ assumption is a deliberate choice to make our analysis transparent and tractable, enabling exact theorems that clarify the *core design principles* of self-attention. While not always practical, the insights gained from this assumption shed light on why self-attention mechanisms are so effective in learning interactions, thereby paving the way for future research on more approximate and scalable extensions.

**Overview and Motivation of the Following Examples** In this appendix, we present several illustrative examples that showcase how *single-layer linear self-attention* can capture important pairwise interactions across diverse domains: multi-agent collision settings, time series forecasting, genotype-phenotype mapping, simple vision tasks, and more. Although there exist many possible configurations, we focus on relatively straightforward, yet representative cases that make it clear how to embed domain-specific features into our theoretical framework.

## B.1. Colliding Agents Environment

We consider $L$ *identical* agents on a cylindrical grid of size $[N] \times [N]$, with initial position vector $\mathbf{r}_i = \begin{bmatrix} n_i^x & n_i^y \end{bmatrix} \in [N]^2$, where $y$ axis is looped, that is, the distance between $n_i^y = 1$ and $n_i^y = N - 1$ is just 2. An analogy for such a grid world is that ants moving on the surface of a water pipe. Each agent is modeled as a circle of radius $R$ executes a simple "move-right" policy (one step to the right per time step if there is space to move, $n_i^x < N - 1$, otherwise stays where it is). We denote the value function for agent $i$ as $V_i$, which corresponds to the expected return (sum of discounted rewards) from a particular configuration. Our goal is to capture how each agent's value function $V_i$ depends on the initial states (positions) of other agents: whenever agent $i$ collides with agent $j$ (distinct), it receives a penalty of $-1$. Leveraging Theorem 3.1, we show how this reward structure (and hence the value function if continuing in time) can be exactly represented by a single-layer linear self-attention mechanism.

Under this setting, adding $-1$ bias to all value functions for simplicity, the value function for each agent can be expressed as

$$V_{\mathbf{r}_i} = \sum_{\substack{j \in [L] \\ j \neq i}} \mathbb{I}\{\min\left(\left|n_i^y - n_j^y\right|, N - \left|n_i^y - n_j^y\right|\right) \leq 2R\} \cdot (-1) \tag{13}$$

Seeing that the value function depends only on the $y$-coordinates, we focus our discussion on a one-dimensional case for simplicity. The extension to value functions with higher dimension dependence follows naturally and is illustrated after the one-dimensional case discussed here. In addition, we provide two versions of the similar representation construction for different positional embeddings.

### B.1.1. ONE-HOT EMBEDDING

One of the most straightforward approaches is to embed each agent's position $n_i \in [N]$ as the standard basis vector $\mathbf{e}_{n_i} \in \mathbb{R}^N$. Concretely, $n_i$-th entry of $\mathbf{e}_{n_i}$ is one and the rest are zeros. This guarantees $\mathbf{e}_{n_i}^\top \mathbf{e}_{n_j} = \delta_{n_i, n_j}$, making the embedding orthonormal.

**Representing the Collision Value Function.** By Theorem 3.1, there exists a single-layer linear self-attention model (dimension $d = N$) that encodes the collision-based value function $V_i$ via an appropriate choice of $\mathbf{C} \in \mathbb{R}^{N \times N}$ and

$\mathbf{W}^V \in \mathbb{R}^{N \times 1}$. To represent the value function in Eq. (13) using a single-layer linear self-attention mechanism, we explicitly construct the attention weight matrix $\mathbf{C}$ and value projection matrix $\mathbf{W}^V$. Define $\mathbf{W}^V \in \mathbb{R}^{N \times 1}$ as a vector of all $-1$s:

$$\mathbf{W}^V = -\mathbf{1}_N = \begin{bmatrix} -1 & -1 & \ldots & -1 \end{bmatrix}^\top. \tag{14}$$

This ensures that the output of self-attention sums over the interactions weighted by $\mathbf{C}$. The interaction matrix $\mathbf{C} \in \mathbb{R}^{N \times N}$ is defined to capture the penalty for collisions:

$$C_{mn} = \begin{cases} 1, & \text{if } \min(|m - n|, N - |m - n|) \leq 2R, \\ 0, & \text{otherwise.} \end{cases} \tag{15}$$

Here, $C_{mn} = 1$ whenever the $y$-coordinates $m$ and $n$ are within a radius of $2R$, accounting for cylindrical wrapping. The resulting attention computation effectively sums over all nearby agents within the collision range.

### B.1.2. SINUSOIDAL EMBEDDING

We now consider sinusoidal embeddings, inspired by various positional embedding techniques that leverage sinusoidal representations. These include absolute positional embeddings from (Vaswani et al., 2023) and the more recent rotational embeddings in (Su et al., 2023). Specifically, assuming $N$ is an even number, we embed the position of the $i$-th agent as:

$$\mathbf{x}(\mathbf{r}_i) = \mathbf{p}_i = \begin{bmatrix} \frac{1}{\sqrt{2}} & \ldots & \sin\left(\frac{2\pi k}{N} n_i\right) & \cos\left(\frac{2\pi k}{N} n_i\right) & \ldots & \frac{1}{\sqrt{2}} \cos\left(\frac{2\pi}{N}\left(\frac{N}{2} - 1\right) n_i\right) & \frac{1}{\sqrt{2}} \cos\left(\frac{2\pi}{N}\frac{N}{2} n_i\right) \end{bmatrix}^\top \in \mathbb{R}^N. \tag{16}$$

It is a simple exercise to check that $\mathbf{p}_i^\top \mathbf{p}_j = \frac{N}{2} \delta_{i,j}$

The *main result* of this section is that: due to Lemma B.4 and Threorem B.6, a single layer linear self-attention 1 with $\mathbf{W}^V = \begin{bmatrix} -\sqrt{2} & 0 & 0 & \ldots \end{bmatrix}^\top \in \mathbb{R}^{N \times 1}$ and diagonal $\mathbf{C} \in \mathbb{R}^{N \times N}$, such that

$$C_{nn} = \begin{cases} \frac{4}{N}\left(2R + \frac{1}{2}\right) & \text{if } n = 0, \\ \frac{2}{N} \dfrac{\sin\left[\frac{2\pi}{N}\left(2R + \frac{1}{2}\right)\left(\frac{n+1}{2}\right)\right]}{\sin\left[\frac{2\pi}{N}\frac{1}{2}\left(\frac{n+1}{2}\right)\right]} & \text{if } n \text{ odd}, \\ \frac{2}{N} \dfrac{\sin\left[\frac{2\pi}{N}\left(2R + \frac{1}{2}\right)\left(\frac{n}{2}\right)\right]}{\sin\left[\frac{2\pi}{N}\frac{1}{2}\left(\frac{n}{2}\right)\right]} & \text{if } n \text{ even}, \end{cases}$$

can represent the value functions at Eq. (13) exactly, **for any** $L$, which is plotted at Figure 1.

**Definition B.3** (Discrete-Time Fourier Series (DTFS)). DTFS of a function $f : [N] \to \mathbb{R}$ is defined as

$$F[k] = \sum_{n=0}^{N-1} f[n] e^{-j \frac{2\pi}{N} kn}.$$

**Lemma B.4** (Discrete-Time Fourier Series Expansion of Window Function). *Let $f$ be:*

$$f[n] = \mathbb{I}\left[|n| \leq 2R\right]$$

*Then the DFS, $F$ is:*

$$F[k] = \begin{cases} \frac{1}{N} \dfrac{\sin\left[\frac{2\pi}{N}\left(2R + \frac{1}{2}\right)k\right]}{\sin\left[\frac{2\pi}{N}\frac{1}{2}k\right]} & \text{if } k \neq 0 \\ \frac{2}{N}\left(2R + \frac{1}{2}\right) & \text{if } k = 0 \end{cases}$$

*Proof.* Straightforward. $\square$

**Lemma B.5** (Real Valued, i.e. Sinusoidal, Expression of Discrete-Time Fourier Series). *The Discrete Fourier Series (DFS) of a periodic discrete function $x[n]$ with period $N$ is given by:*

$$x[n] = \sum_{k=0}^{N-1} X[k] e^{i \frac{2\pi}{N} kn}$$

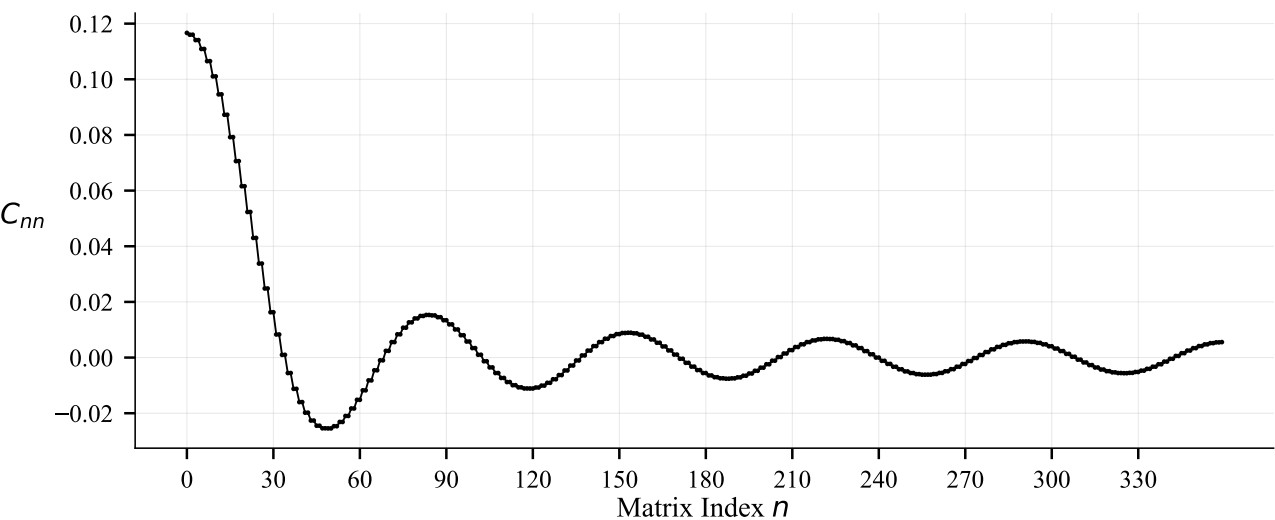

**Figure 1.** Diagonal entries of the interaction matrix $C$ for $N = 360$ agents with radius $R = 5$. The pattern emerges from Fourier analysis of collision dynamics in the agent movement model. The matrix indices are from 0 to $N - 1$.

*Letting $a_k = 2\,\mathrm{Re}\{X[k]\}$ and $b_k = -2\,\mathrm{Im}\{X[k]\}$, the same function can be expressed as*

$$x[n] = \begin{cases} \dfrac{a_0}{2} + \displaystyle\sum_{k=1}^{\frac{N-1}{2}} \left( a_k \cos\left(\dfrac{2\pi}{N} kn\right) + b_k \sin\left(\dfrac{2\pi}{N} kn\right) \right) & \text{, for odd } N \\[2em] \dfrac{a_0}{2} + \dfrac{a_{N/2}}{2} \cos\left(\dfrac{2\pi}{N}\dfrac{N}{2} n\right) + \displaystyle\sum_{k=1}^{\frac{N}{2}-1} \left( a_k \cos\left(\dfrac{2\pi}{N} kn\right) + b_k \sin\left(\dfrac{2\pi}{N} kn\right) \right) & \text{, for even } N \end{cases}$$

*Proof.* Using Euler's formula, the complex exponential term can be rewritten as:

$$e^{i\frac{2\pi}{N} kn} = \cos\left(\frac{2\pi}{N} kn\right) + i \sin\left(\frac{2\pi}{N} kn\right)$$

Substitute this expression into the DTFS equation:

$$x[n] = \sum_{k=0}^{N-1} X[k] \left( \cos\left(\frac{2\pi}{N} kn\right) + i \sin\left(\frac{2\pi}{N} kn\right) \right)$$

Separate the real and imaginary parts of the equation:

$$x[n] = \sum_{k=0}^{N-1} \left( \mathrm{Re}\{X[k]\} \cos\left(\frac{2\pi}{N} kn\right) - \mathrm{Im}\{X[k]\} \sin\left(\frac{2\pi}{N} kn\right) \right)$$

$$+ i \sum_{k=0}^{N-1} \left( \mathrm{Im}\{X[k]\} \cos\left(\frac{2\pi}{N} kn\right) + \mathrm{Re}\{X[k]\} \sin\left(\frac{2\pi}{N} kn\right) \right)$$

$x[n]$ is a real function, so the imaginary part sums to zero. Therefore, for odd $N$ we have:

- $\mathrm{Im}\{X[0]\} = 0$

- $\mathrm{Im}\{X[k]\} + \mathrm{Im}\{X[N-k]\} = 0$ because cosine function has period of $\pi$

- $\mathrm{Re}\{X[k]\} - \mathrm{Re}\{X[N-k]\} = 0$ because sine function has period of $\pi$

As for the even $N$ we have an additional requirement:

- $\mathrm{Im}\{X[N/2]\} = 0$

Thus $x[n]$ for odd $N$ is:

$$x[n] = \mathrm{Re}\{X[0]\} + \sum_{k=1}^{\frac{N-1}{2}} \left( 2\,\mathrm{Re}\{X[k]\} \cos\left(\frac{2\pi}{N}kn\right) - 2\,\mathrm{Im}\{X[k]\} \sin\left(\frac{2\pi}{N}kn\right) \right),$$

As for even $N$ we have:

$$x[n] = \mathrm{Re}\{X[0]\} + \mathrm{Re}\{X[N/2]\}(-1)^n$$
$$+ \sum_{k=1}^{\frac{N}{2}-1} \left( 2\,\mathrm{Re}\{X[k]\} \cos\left(\frac{2\pi}{N}kn\right) - 2\,\mathrm{Im}\{X[k]\} \sin\left(\frac{2\pi}{N}kn\right) \right),$$

Letting $a_k = 2\,\mathrm{Re}\{X[k]\}$ and $b_k = -2\,\mathrm{Im}\{X[k]\}$, we get the final form as:

$$x[n] = \begin{cases} \dfrac{a_0}{2} + \displaystyle\sum_{k=1}^{\frac{N-1}{2}} \left( a_k \cos\left(\frac{2\pi}{N}kn\right) + b_k \sin\left(\frac{2\pi}{N}kn\right) \right) & \text{, for odd } N \\[3ex] \dfrac{a_0}{2} + \dfrac{a_{N/2}}{2} \cos\left(\frac{2\pi}{N}\frac{N}{2}n\right) + \displaystyle\sum_{k=1}^{\frac{N}{2}-1} \left( a_k \cos\left(\frac{2\pi}{N}kn\right) + b_k \sin\left(\frac{2\pi}{N}kn\right) \right) & \text{, for even } N \end{cases}$$

$\square$

**Theorem B.6.** *Let us denote DTFS of a function $f[n]$ as $F[k]$. Defining $a_k = 2\,\mathrm{Re}\{F[k]\}$ and $b_k = -2\,\mathrm{Im}\{F[k]\}$ a attention score matrix $\mathbf{C}$ of the form*

$$\mathbf{C} = \begin{bmatrix} a_0 & & & & & & \\ & a_1 & b_1 & & & & \\ & -b_1 & a_1 & & & & \\ & & & \ddots & & & \\ & & & & a_{N/2-1} & b_{N/2-1} & \\ & & & & -b_{N/2-1} & a_{N/2-1} & \\ & & & & & & a_{N/2} \end{bmatrix}, \tag{17}$$

*can represent any function $f[n_i - n_j] = \mathbf{p}_i \mathbf{C} \mathbf{p}_j^T$, where $\mathbf{p}_i$ is given at Eq. (16).*

*Proof.* Setting $\mathbf{C}$ as in (17),

$$\mathbf{p}_i \mathbf{C} \mathbf{p}_j^T = \frac{a_0}{2} + \frac{a_{N/2}}{2} \cos\left(\frac{2\pi}{N}\frac{N}{2}n\right) + \sum_{k=1}^{\frac{N}{2}-1} \left( a_k \cos\left(\frac{2\pi}{N}kn\right) + b_k \sin\left(\frac{2\pi}{N}kn\right) \right).$$

That is equal to $f[n]$ owing to Lemma B.5. $\square$

*Remark B.7.* If $f$ is symmetric, then the corresponding $\mathbf{C}$ is diagonal.

### B.1.3. MORE COMPLEX COLLIDING AGENTS ENVIRONMENTS

**Value Functions That Depend on Both Coordinates.** Assume the same basic grid setup described earlier, but now suppose each agent's value function depends on *both* coordinates. A trivial example is when agents receive huge penalties $-10$ if they collide even though initially they are far away, i.e, $\text{dist}(n_i, n_j) \geq R_2$ for some $R_2 \in \mathbb{R}$ and any $\text{dist}(\cdot)$ function. They receive small $-1$ penalties, if initially they are very close, so escaping from the collision is difficult. Thus, the value function becomes

$$V_i = -\sum_{j \neq i} \mathbb{I}\{|n_i^y - n_j^y| \leq 2R\} \cdot (1 + 9 \cdot \mathbb{I}\{\text{dist}(n_i, n_j) \leq R_2\}),$$

where the $x$-coordinate condition encodes temporal proximity. This value function, along with similar ones, can be expressed in a general form as follows. Consider $L$ agents on a two-dimensional $[N] \times [N]$ grid, with each agent $i$ occupying a cell at integer coordinates $\mathbf{r}_i = \begin{bmatrix} n_i^x & n_i^y \end{bmatrix}^\top \in [N]^2$. Each agent's value function is then influenced by pairwise interactions of the form

$$V_i = \sum_{\substack{j \in [L] \\ j \neq i}} f(\mathbf{r}_i - \mathbf{r}_j)\, w_{\mathbf{r}_j} = \sum_{\substack{j \in [L] \\ j \neq i}} f(n_i^x - n_j^x,\ n_i^y - n_j^y)\, w_{\mathbf{r}_j},$$

where $f : [N]^2 \to \mathbb{R}$ measures *how* agent $j$ (via its relative position) influences agent $i$. To represent $f$ exactly as a linear self-attention kernel (as in Theorem 3.1), we embed *each* 2D position $\mathbf{r}_i$ into a vector of dimension $N^2$, defined by the Kronecker (outer) product

$$\mathbf{p}_i = \mathbf{p}_i^x \otimes \mathbf{p}_i^y \in \mathbb{R}^{N^2}.$$

Here, $\mathbf{p}_i^x, \mathbf{p}_i^y \in \mathbb{R}^N$ are the 1D sinusoidal embeddings from Equation (16), applied separately to the $x$- and $y$-coordinates. By construction, $\{\mathbf{p}_i\}_{i=1}^L$ spans an orthogonal set in $\mathbb{R}^{N^2}$, with one vector per distinct grid cell.

**From 1D to 2D Discrete-Time Fourier Series.** Recall from Lemma B.4 and Theorem B.6 that any function $g(n)$ on a 1D grid $[N]$ can be written as $\mathbf{p}_i^\top \mathbf{C} \mathbf{p}_j$ for a suitably chosen matrix $\mathbf{C}$. Precisely the same reasoning applies in two dimensions via a 2D Discrete-Time Fourier Series (DTFS). Specifically, one writes

$$f(n_i^x - n_j^x,\ n_i^y - n_j^y) = \sum_{k_x=0}^{N-1} \sum_{k_y=0}^{N-1} F[k_x, k_y]\, e^{i \frac{2\pi}{N} \left( k_x\, (n_i^x - n_j^x) + k_y\, (n_i^y - n_j^y) \right)},$$

and then rewrites each exponential in terms of sines/cosines matching the tensor-product embedding $\mathbf{p}_i^x \otimes \mathbf{p}_i^y$. Consequently, there exists a block-structured $\mathbf{C} \in \mathbb{R}^{N^2 \times N^2}$ such that

$$\mathbf{p}_i^\top \mathbf{C} \mathbf{p}_j = f(\mathbf{r}_i - \mathbf{r}_j).$$

Hence, by choosing $\mathbf{W}^V$ so that $\mathbf{p}_i \mapsto w_{\mathbf{r}_i}$, a *single-layer linear self-attention* (dimension $d = N^2$) recovers exactly the mapping

$$V_i = \sum_{j \neq i} f(\mathbf{r}_i - \mathbf{r}_j)\, w_{\mathbf{r}_j}.$$

All the 1D collision arguments from earlier carry over: once the domain is embedded into $\mathbb{R}^{N^2}$, Theorem 3.1 implies that single-layer linear self-attention can represent any pairwise function $f(\mathbf{r}_i - \mathbf{r}_j)$. Of course, this comes at the cost of embedding dimension $N^2$, so the resulting kernel $\mathbf{C}$ has size $N^2 \times N^2$, i.e. an $\mathcal{O}(N^4)$ parameter count for a fully general 2D interaction.

**Non-Identical Agents.** One can likewise handle agents with *different* behavior or policies. For example, suppose half of the agents always move right until they reach the boundary, while the others always move up. Label these two behaviors (or policies) via a discrete set $\mathcal{S}_q = \{\text{R, U}\}$, and let each agent $i$ carry an extra label $q_i \in \mathcal{S}_q$. Then the value function is

$$V_{\mathbf{r}_i, q_i} = \sum_{\substack{j \in [L] \\ j \neq i}} f\left( \mathbf{r}_i - \mathbf{r}_j,\ q_i,\ q_j \right) w_{\mathbf{r}_j, q_j}. \tag{18}$$

Following steps similar to the value functions that depend on both coordinates example, we now view the domain of each agent as

$$\mathcal{S} \;=\; \mathcal{S}_q \;\times\; \mathcal{S}_x \;\times\; \mathcal{S}_y,$$

where $\mathcal{S}_x$ and $\mathcal{S}_y$ are each $[N]$. Its cardinality is $|\mathcal{S}| = 2\,N^2$, in this simple example. We then embed each agent's label and position as a vector in $\mathbb{R}^{2N^2}$. For instance, if $q_i = \mathrm{R}$ we might set

$$q_i \;=\; \begin{bmatrix} 1 \\ 0 \end{bmatrix}, \quad \mathbf{p}_i^x, \; \mathbf{p}_i^y \;\in\; \mathbb{R}^N, \quad \mathbf{z}_i \;=\; \mathbf{q}_i \otimes \mathbf{p}_i^x \otimes \mathbf{p}_i^y \;\in\; \mathbb{R}^{2N^2}.$$

Likewise if $q_i = \mathrm{U}$, we flip those two bits in $\mathbf{q}_i$. By expanding the definition of $f(\mathbf{r}_i - \mathbf{r}_j, q_i, q_j)$ via a suitable DTFS, one again obtains a matrix $\mathbf{C}$ of size $(2N^2) \times (2N^2)$ that captures all pairwise interactions. Thus a single-layer linear self-attention with $d = 2N^2$ dimensions suffices to represent Eq. (18). The parameter count grows to $\mathcal{O}(4\,N^4)$ in the fully general case, but the construction exactly parallels the identical-agent scenario.

## B.2. Genotype–Phenotype Mapping Task

Each *allele* (or gene variant) in the domain $\mathcal{S}$ is labeled with a unique integer and embedded using a one-hot vector. We consider a DNA sequence of length $L$ in which every position corresponds to a unique allele. In other words, each allele appears at most once in the sequence (no duplicates). Our experiments randomly sample *activation relations*: an allele $\mu \in \mathcal{S}$ is *activated* if another allele $\nu$ exists somewhere in the sequence. Symbolically, we might store these relations in a dictionary of the form

$$\mu : [\nu_1, \nu_2, ...],$$

meaning "allele $i$ is activated by the set of alleles-$\nu_k$" If $i : [\,]$ (an empty list), then allele $i$ is *always* active, while alleles not present in the dictionary behave like redundant or inactive genetic material.

**Constructing C and $\mathbf{W}^V$.** To model these activation patterns via a single-layer linear self-attention, we build $\mathbf{C} \in \mathbb{R}^{d \times d}$ and $\mathbf{W}^V \in \mathbb{R}^{d \times 1}$ as follows (assuming each allele is one-hot embedded into $\mathbb{R}^d$, with $d = |\mathcal{S}|$):

* For every dictionary entry of the form $i : [j_1, \ldots, j_m]$, set

$$\mathbf{C}_{i,\,j_k} \;=\; 1/m \quad \text{for each } k = 1, \ldots, m,$$

  and set all other entries in row $i$ to zero.

* For entries $i : [\,]$ (i.e., allele $i$ is always active), assign each entry in row $i$ to $\frac{1}{L}$. This ensures allele $i$ gains a constant contribution from the entire sequence.

* Set every entry of $\mathbf{W}^V$ to 1 (i.e., $\mathbf{W}^V \in \mathbb{R}^{d \times 1}$ is a vector of all ones).

An example dictionary may be $\{1{:}[3],\ 2{:}[],\ 4{:}[2]\}$. Seeing that we have one hot encoding

$$C = \begin{bmatrix} 0 & 0 & 0 & 0 & 0 \\ 0 & 0 & 0 & 1 & 0 \\ 1/L & 1/L & 1/L & 1/L & 1/L \\ 0 & 0 & 0 & 0 & 0 \\ 0 & 0 & 1 & 0 & 0 \end{bmatrix}$$

This is an example task for which a single layer linear self-attention cannot length generalize. However, a simple dense layer with ReLU activation addition after single layer linear self-attention, trivially enables it to generalize any length

## B.3. Vision Task

In this example, each *token* or *entity* corresponds to a single pixel in an image. Suppose there are $d$ possible pixel *positions* in the image, each with a positional embedding $\mathbf{p}_i \in \mathbb{R}^d$. We also have a binary indicator $b_i \in \{0, 1\}$ denoting whether pixel $i$ is black ($b_i = 1$) or white ($b_i = 0$). We embed each pixel *jointly* as

$$\mathbf{x}_i \;=\; \begin{bmatrix} b_i, & \mathbf{p}_i \end{bmatrix} \;\in\; \mathbb{R}^{1+d}.$$

Thus, the first coordinate captures color, and the remaining coordinates capture position. We want to detect whether a specific *pattern of black pixels* occurs around the position $\mathbf{p}_i$. Formally, we want to detect if pixels in a set of relative offsets

$$\Delta \;=\; \{\Delta_1, \Delta_2, \dots\}$$

are all black around position $i$. For example, $\Delta$ might define a small pattern (e.g., a $3 \times 3$ cross), such that for each $\Delta_k \in \Delta$, the pixel at $\mathbf{p}_i + \Delta_k$ must be black for us to declare "shape present at $i$."

**Block-Structured C Matrix.** Consider a single-layer linear self-attention of dimension $d + 1$. Let $\mathbf{C} \in \mathbb{R}^{(d+1)\times(d+1)}$ be block-structured, meaning:

$$\mathbf{C} \;=\; \begin{bmatrix} \mathbf{0}_{1\times 1} & \mathbf{0}_{1\times d} \\ \mathbf{0}_{d\times 1} & \widetilde{\mathbf{C}}_{d\times d} \end{bmatrix},$$

where $\mathbf{0}$ denotes a zero block (ensuring that the binary coordinate $b_i$ does not directly alter *which* positions are relevant). If we want to detect a shape around $\mathbf{p}_i$ by checking offsets $\Delta = \{\Delta_1, \dots, \Delta_m\} \subseteq \mathbb{R}^d$, then for each row $i$ (representing $\mathbf{p}_i$ in the $\widetilde{\mathbf{C}}$ submatrix), we set

$$\widetilde{\mathbf{C}}_{i,j} \;=\; \begin{cases} 1, & \text{if } \mathbf{p}_j \text{ is in } \mathbf{p}_i + \Delta, \\ 0, & \text{otherwise.} \end{cases}$$

This ensures $\mathbf{p}_i$ attends to exactly those pixel positions $\mathbf{p}_j$ that lie within the shape region around $i$.

**Capturing the Binary Color.** To incorporate the notion that only black pixels ($b_j = 1$) contribute, we define $\mathbf{W}^V \in \mathbb{R}^{(d+1)\times 1}$ so that its entries corresponding to the *binary part* of $\mathbf{x}_j$ are nonzero, while its entries corresponding to the *positional part* are zero. More formally,

$$\mathbf{W}^V \;=\; \begin{bmatrix} 1 \\ \mathbf{0}_d \end{bmatrix} \;\in\; \mathbb{R}^{1+d}.$$

Hence, when we multiply $\mathbf{x}_j$ by $\mathbf{W}^V$, the outcome is simply $b_j$. In other words, if $b_j = 1$, the pixel contributes; if $b_j = 0$, it does not.

**Overall Operation.** Putting it together, our single-layer linear self-attention

$$\left(\mathbf{X}\,\mathbf{C}\,\mathbf{X}^\top\right)\mathbf{X}\,\mathbf{W}^V$$

behaves as follows: (i) $\widetilde{\mathbf{C}}$ identifies the relevant offsets $\mathbf{p}_j$ around each $\mathbf{p}_i$, i.e., it checks which pixels could participate in a pattern at $i$. (ii) $\mathbf{W}^V$ converts the embedding $\begin{bmatrix} b_j, & \mathbf{p}_j \end{bmatrix}$ into $b_j$, effectively a blackness indicator. (iii) Summing across the image yields, for each position $i$, the count of black pixels at the appropriate offsets $\Delta$.

If this count matches the size of $\Delta$, we conclude that the shape is present around pixel $i$. Thus, the linear self-attention mechanism can recognize patterns of arbitrary size and structure *without changing the kernel size* or other architectural hyperparameters.

**CNN Versus Transformer: Theoretical Insight.** A single-layer Convolutional Neural Network (CNN) with a fixed $3 \times 3$ kernel cannot detect patterns larger than $3 \times 3$ within that same layer. Extending the receptive field would require either deeper networks or larger kernels. However, we do not know even the task of a layer in a large and deep neural network. Therefore, generally we do not know the required kernel for the task in advance, so we just cross validate different over different trials. This design constraint makes CNNs less flexible when the optimal pattern scale is not known a priori. By contrast, Transformers can attend to any subset of positions in the input —whether nearby or distant— in a single layer. In our example, the shape offsets $\Delta$ might be large or irregular, yet the same $\widetilde{\mathbf{C}}$ submatrix can be adapted to capture those long-range relationships. Consequently, Transformers provide a more *versatile* framework for shape detection (which is simply learning how different parts of image interact with each other) or other vision tasks where the required pattern scale or geometry may vary significantly from one scenario to another.

### B.4. Time Series Prediction

Consider a univariate or multivariate time series $\{\mathbf{m}[t]\}_{t=1}^{L+1}$, where each $\mathbf{m}[t] \in \mathbb{R}^{d_2}$ is the observed value at time $t$. We assume $\mathbf{m}[t]$ depends on a set of specific past delays $D = \{t_1, t_2, \dots\} \subset \mathbb{N}$, along with scalar multipliers $\{a_k\}_{k \in D}$ and a

linear transform $\mathbf{A} \in \mathbb{R}^{d_2 \times d_2}$ capturing how past values affect the current state:

$$\mathbf{m}[t] \;=\; \sum_{k \in D} a_k \, \mathbf{A} \, \mathbf{m}[t - k].$$

For instance, if $D = \{2, 5\}$, then $\mathbf{m}[t] = 3\,\mathbf{A}\,\mathbf{m}[t - 2] + 7\,\mathbf{A}\,\mathbf{m}[t - 5]$. For example, if $D = \{2, 5\}$, we get $\mathbf{m}[t] = 3\,\mathbf{A}\,\mathbf{m}[t - 2] + 7\,\mathbf{A}\,\mathbf{m}[t - 5]$.

To embed each time step $t$ as an entity, define

$$\mathbf{x}_t \;=\; \big[\mathbf{m}[t],\, \mathbf{p}[t]\big] \;\in\; \mathbb{R}^{d + d_2},$$

where $\mathbf{m}[t] \in \mathbb{R}^{d_2}$ is the observed state, and $\mathbf{p}[t] \in \mathbb{R}^d$ is a positional embedding (e.g., one-hot or continuous encoding). An indicator-based formulation ensures the attention mechanism recovers delays in $D$:

$$\mathbf{m}[L + 1] \;=\; \sum_{j \in [L]} \bigg( \sum_{k \in D} a_k \, \mathbb{I}\big\{ (L + 1) - j = k \big\} \bigg) \mathbf{A} \, \mathbf{m}[j].$$

Our goal is to show how a *single-layer linear self-attention* can represent this dependency structure exactly, following Theorem 3.1.

We treat each time step $t$ as a distinct entity in the sequence. Its embedding $\mathbf{x}_t \in \mathbb{R}^{d + d_2}$ combines:

$$\mathbf{x}_t \;=\; \big[\mathbf{m}[t],\, \mathbf{p}[t]\big],$$

where:

- $\mathbf{m}[t] \in \mathbb{R}^{d_2}$ is the observed state at time $t$ (univariate or multivariate).

- $\mathbf{p}[t] \in \mathbb{R}^d$ is a *positional embedding* encoding the index $t$. This could be a one-hot vector of length $L$, or a sinusoidal embedding.

Define a block-structured matrix $\mathbf{C} \in \mathbb{R}^{(d + d_2) \times (d + d_2)}$ to separate the $\mathbf{m}[t]$ coordinates from the positional coordinates $\mathbf{p}[t]$. We can denote:

$$\mathbf{C} \;=\; \begin{bmatrix} \mathbf{0}_{d_2 \times d_2} & \mathbf{0}_{d_2 \times d} \\ \mathbf{0}_{d \times d_2} & \widetilde{\mathbf{C}}_{d \times d} \end{bmatrix},$$

so that the *positional* part $\widetilde{\mathbf{C}}$ encodes which time delays matter, while the $\mathbf{m}[t]$ portion does not directly determine *which* indices to attend to. Concretely, let $\widetilde{\mathbf{C}}_{u,v} = 1$ if position $v$ is a valid activator for position $u$ under one of the delays in $D$, and $0$ otherwise. Equivalently, we can define

$$\widetilde{\mathbf{C}}_{u,v} \;=\; \sum_{k \in D} a_k \mathbf{I}\big[ u - v = k \big],$$

if we index the positional embedding such that $u, v \in \{1, \ldots, L\}$. This ensures that time step $u$ attends to time step $v$ if $v$ is exactly one of the valid delays $k$ behind $u$.

Next, we define $\mathbf{W}^V \in \mathbb{R}^{(d + d_2) \times d_2}$ to extract the actual state $\mathbf{m}[t]$ from each token:

$$\mathbf{W}^V \;=\; \begin{bmatrix} \mathbf{A} \\ \mathbf{0}_{d \times d_2} \end{bmatrix},$$

where $\mathbf{A} \in \mathbb{R}^{d_2 \times d_2}$ is precisely the transformation from the autoregressive model. This construction means that when we multiply $\mathbf{x}_t$ by $\mathbf{W}^V$, we obtain $\mathbf{A}\,\mathbf{m}[t]$, and the positional coordinates are ignored in this step (since their block is zero).

Putting it all together, a single-layer linear self-attention computes

$$\big(\mathbf{X} \, \mathbf{C} \, \mathbf{X}^\top \big) \mathbf{X} \, \mathbf{W}^V.$$

For the row corresponding to the final time step $L + 1$, the multiplication by $\mathbf{C}$ picks out those time steps $j$ such that $(L + 1) - j \in D$. Then multiplying by $\mathbf{W}^V$ retrieves $\mathbf{A}\,\mathbf{m}[j]$. Summing the contributions yields precisely

$$\mathbf{m}[L + 1] \;=\; \sum_{k \in D} a_k \, \mathbf{A} \, \mathbf{m}[L + 1 - k],$$

mirroring the autoregressive formula.

# C. Proof of Theorem 4.4: Convergence to Zero Training Error

Recall that $\mathbf{B} \in \mathbb{R}^{|\mathcal{S}| \times d}$ is the embedding base matrix whose rows are embeddings of each element in the domain. We start with the following simple observation.

**Lemma C.1.** *If the domain embeddings are orthonormal, that is ,* $\langle x(\alpha), x(\beta) \rangle = \delta_{a,b}$, $\forall \alpha$, $\beta \in \mathcal{S}$, *then* $\mathbf{X}^\top \mathbf{X}$ *is diagonal in the embedding basis, and* $\mathbf{B} \mathbf{X}^{(n)\top} \mathbf{X}^{(n)} \mathbf{B}^\top = \mathrm{diag}(\mathbf{s}^{(n)})$, *where* $s_\mu^{(n)}$ *is the number of times the element* $\mu$ *appears in the* $n$-*th sample.*

Since we set $d = |\mathcal{S}|$, we can freely adopt an orthonormal domain embedding, ensuring that

$$\langle \mathbf{B}_{i,:}, \mathbf{B}_{j,:} \rangle = \delta_{i,j}.$$

For the remainder of this section, we conduct all calculations in the domain embedding basis. To formalize this, we define the following orthonormal transformations:

$$\mathbf{X}^{(n)\,\mathrm{one-hot}} = \mathbf{X}^{(n)} \mathbf{B}^\top,$$
$$\mathbf{C}^{\mathrm{one-hot}} = \mathbf{B} \mathbf{C} \mathbf{B}^\top,$$
$$\mathbf{w}^{\mathrm{one-hot}} = \mathbf{B} \mathbf{w}.$$

For notational simplicity, we omit the $\mathrm{one-hot}$ superscripts in the **rest of this section**. This will not cause any confusion as we are always referring to the one-hot transformed versions. That is, whenever we write $\mathbf{X}^{(n)}, \mathbf{C}, \mathbf{w}$, they actually represent $\mathbf{X}^{(n)\,\mathrm{one-hot}}, \mathbf{C}^{\mathrm{one-hot}}, \mathbf{w}^{\mathrm{one-hot}}$. Also, seeing that $\mathbf{B}$ is orthonormal matrix, when we establish the convergence of the one-hot parameter representations, it directly implies the convergence of the original parameters, which can be recovered by applying the inverse transformation in the $\mathbf{B}$ basis. What we do is simply change our perspective to how we look at coordinate system. Lastly, in this section, we denote $\mathbf{e}_\mu \in \mathbb{R}^{|\mathcal{S}|}$ as unique one-hot encoded vector for all $\mu \in \mathcal{S}$, i.e. the base vector.

We will firstly derive the gradients of the loss function with respect to the parameters than we will state a lemma that will be useful for the proof of Theorem 3.1.

**Gradients of the** $L^{\mathrm{MSE}}(\mathbf{C}, \mathbf{w})$ **with Respect to C and w.** For convenience, denote $\mathbf{W}^V \in \mathbb{R}^{d \times 1}$ as $\mathbf{w} \in \mathbb{R}^d$. It is easy to verify the following equation:

$$\frac{\partial L^{\mathrm{MSE}}(\mathbf{C}, \mathbf{w})}{\partial \mathbf{C}} = \frac{1}{B} \sum_{n=1}^{B} (\mathbf{D}^{(n)})^\top \frac{\partial}{\partial \mathbf{C}} \mathbf{SA}^{\mathrm{lin}}(\mathbf{X}^{(n)}), \qquad \frac{\partial L^{\mathrm{MSE}}(\mathbf{C}, \mathbf{w})}{\partial \mathbf{w}} = \frac{1}{B} \sum_{n=1}^{B} (\mathbf{D}^{(n)})^\top \frac{\partial}{\partial \mathbf{w}} \mathbf{SA}^{\mathrm{lin}}(\mathbf{X}^{(n)}).$$

Since $d_2 = 1$ the linear self-attention in Eq.1 can be written as

$$\mathbf{SA}^{\mathrm{lin}}(\mathbf{X}^{(n)}) = (\mathbf{X} \mathbf{C} \mathbf{X}^\top) \mathbf{X} \mathbf{w},$$

where $\mathbf{w} \in \mathbb{R}^d$. We have

$$\frac{\partial \mathbf{SA}^{\mathrm{lin}}(\mathbf{X}^{(n)})}{\partial C_{\mu\nu}} = \mathbf{X}_{:\mu} \left[ \mathbf{X}^\top \mathbf{X} \mathbf{w} \right]_\nu.$$

Hence,

$$\frac{\partial L^{\mathrm{MSE}}(\mathbf{C}, \mathbf{w})}{\partial C_{\mu\nu}} = \frac{2}{B} \sum_{n=1}^{B} \left[ \mathbf{X}^{(n)\top} \mathbf{X}^{(n)} \mathbf{w} \right]_\nu \left( \mathbf{X}_{:\mu}^{(n)} \right)^\top \mathbf{D}^{(n)}.$$

Similarly,

$$\frac{\partial \mathbf{SA}^{\mathrm{lin}}(\mathbf{X}^{(n)})}{\partial w_\alpha} = (\mathbf{X} \mathbf{C} \mathbf{X}^\top) \mathbf{X}_{:\alpha},$$

$$\frac{\partial L^{\mathrm{MSE}}(\mathbf{C}, \mathbf{w})}{\partial w_\alpha} = \frac{2}{B} \sum_{n=1}^{B} \left( \mathbf{X}^{(n)\top} \right)_{\alpha:} \left( \mathbf{X}^{(n)} \mathbf{C}^\top \mathbf{X}^{(n)\top} \right) \mathbf{D}^{(n)}.$$

We can write the same gradient equations as

$$\frac{\partial L^{\mathrm{MSE}}\left(\mathbf{C}, \mathbf{w}\right)}{\partial \mathbf{C}} = \frac{2}{B} \sum_n \mathbf{X}^{(n)\,\top} \mathbf{D}^{(n)} \mathbf{w}^\top \mathbf{X}^{(n)\,\top} \mathbf{X}^{(n)},$$

$$\frac{\partial L^{\mathrm{MSE}}\left(\mathbf{C}, \mathbf{w}\right)}{\partial \mathbf{w}} = \frac{2}{B} \sum_n \mathbf{X}^{(n)\,\top} \mathbf{X}^{(n)} \mathbf{C}^\top \mathbf{X}^{(n)\,\top} \mathbf{D}^{(n)}.$$

**Lemma C.2.** *If we choose initial parameters as $\mathbf{C}(0) = \mathbf{0}$ and $w_\alpha(0) \geq b > 0$, then $w_\alpha(t) \geq b > 0$, $\forall \alpha$ and $\forall t \geq 0$.*

*Proof.* Firstly, we will show that $w_\alpha(t)^2 \geq w_\alpha(0)^2$, $\forall t$ and $\forall i$, than we will prove the statement in the lemma. We can copy the previous gradient derivations and gradient flow equations

$$\frac{d\mathbf{C}}{dt} = -\eta \frac{\partial L^{\mathrm{MSE}}\left(\mathbf{C}, \mathbf{w}\right)}{\partial \mathbf{C}} = -\eta \frac{2}{B} \sum_n \mathbf{X}^{(n)\,\top} \mathbf{D}^{(n)} \mathbf{w}^\top \mathbf{X}^{(n)\,\top} \mathbf{X}^{(n)},$$

$$\frac{d\mathbf{w}}{dt} = -\eta \frac{\partial L^{\mathrm{MSE}}\left(\mathbf{C}, \mathbf{w}\right)}{\partial \mathbf{w}} = -\eta \frac{2}{B} \sum_n \mathbf{X}^{(n)\,\top} \mathbf{X}^{(n)} \mathbf{C}^\top \mathbf{X}^{(n)\,\top} \mathbf{D}^{(n)}.$$

Let $\mathbf{\Lambda}$ be a matrix that is diagonal in the embedding base $\mathbf{B}$. However, we again abuse the notation. We do not rewrite the $\mathrm{one-hot}$ in $\mathbf{\Lambda}^{\mathrm{one-hot}} = \mathbf{B}\mathbf{C}\mathbf{B}^\top$ and denote it just as $\mathbf{\Lambda}$ in the rest of the proof. We can now write

$$\mathbf{C}^\top \frac{d\mathbf{C}}{dt} \mathbf{\Lambda} = -\eta \frac{2}{B} \sum_n \mathbf{C}^\top \mathbf{X}^{(n)\,\top} \mathbf{D}^{(n)} \mathbf{w}^\top \mathbf{X}^{(n)\,\top} \mathbf{X}^{(n)} \mathbf{\Lambda},$$

$$\mathrm{Tr}\left\{ \mathbf{C} \frac{d\mathbf{C}}{dt} \mathbf{\Lambda} \right\} = -\eta \frac{2}{B} \sum_n \mathrm{Tr}\left\{ \mathbf{C}^\top \mathbf{X}^{(n)\,\top} \mathbf{D}^{(n)} \mathbf{w}^\top \mathbf{X}^{(n)\,\top} \mathbf{X}^{(n)} \mathbf{\Lambda} \right\},$$

$$\mathbf{\Lambda} \frac{d\mathbf{w}}{dt} \mathbf{w}^\top = -\eta \frac{2}{B} \sum_n \mathbf{\Lambda} \mathbf{X}^{(n)\,\top} \mathbf{X}^{(n)} \mathbf{C}^\top \mathbf{X}^{(n)\,\top} \mathbf{D}^{(n)} \mathbf{w}^\top,$$

$$\mathrm{Tr}\left\{ \mathbf{\Lambda} \frac{d\mathbf{w}}{dt} \mathbf{w}^\top \right\} = -\eta \frac{2}{B} \sum_n \mathrm{Tr}\left\{ \mathbf{C}^\top \mathbf{X}^{(n)\,\top} \mathbf{D}^{(n)} \mathbf{w}^\top \mathbf{X}^{(n)\,\top} \mathbf{X}^{(n)} \mathbf{\Lambda} \right\}.$$

Thus, we have

$$\mathrm{Tr}\left\{ \mathbf{C}^\top \frac{d\mathbf{C}}{dt} \mathbf{\Lambda} \right\} = \mathrm{Tr}\left\{ \mathbf{\Lambda} \frac{d\mathbf{w}}{dt} \mathbf{w}^\top \right\},$$

$$\mathrm{Tr}\left\{ \frac{d\mathbf{C}}{dt} \mathbf{\Lambda} \mathbf{C}^\top \right\} = \mathrm{Tr}\left\{ \mathbf{w}^\top \mathbf{\Lambda} \frac{d\mathbf{w}}{dt} \right\}. \tag{19}$$

Seeing that $\mathbf{\Lambda}$ is diagonal $\mathbf{\Lambda}^\top = \mathbf{\Lambda}$. Thus, from Equation 19 we can get

$$\frac{d}{dt}\left( \mathrm{Tr}\{\mathbf{C}\mathbf{\Lambda}\mathbf{C}^\top\} - \mathrm{Tr}\{\mathbf{w}^\top \mathbf{\Lambda} \mathbf{w}\} \right) = 0$$

$$\mathrm{Tr}\{\mathbf{C}(t)\mathbf{\Lambda}\mathbf{C}(t)^\top\} - \mathrm{Tr}\{\mathbf{w}^\top(t)\mathbf{\Lambda}\mathbf{w}(t)\} = \mathrm{Tr}\{\mathbf{C}(0)\mathbf{\Lambda}\mathbf{C}(0)^\top\} - \mathrm{Tr}\{\mathbf{w}^\top(0)\mathbf{\Lambda}\mathbf{w}(0)\}$$

Letting $\mathbf{\Lambda} = \mathrm{diag}(\mathbf{e}_\alpha)$, where $\mathbf{e}_\alpha$ is the unit basis vector corresponding to $\alpha$, we reach to

$$w_\alpha^2(t) = w_\alpha^2(0) + \|\mathbf{C}_{:,i}(t)\|_2^2 - \|\mathbf{C}_{:,\alpha}(0)\|_2^2 = w_\alpha^2(0) + \|\mathbf{C}_{:,\alpha}(t)\|_2^2,$$

where the last equality follows because $\mathbf{C}(0) = \mathbf{0}$. As a result we reach to

$$w_\alpha^2(t) \geq w_\alpha^2(0) \geq b^2 \tag{20}$$

Seeing that $\frac{dw_\alpha}{dt}$ is finite $\forall t$, $w_\alpha(t)$ is continuous. As a result if $w_\alpha(0) \geq b > 0$, then $w_\alpha(t) \geq b$, $\forall t$ which can be proven by contradiction. Assume $\exists t^* > 0$ such that $w_\alpha(t^*) \leq b$. By Equation 20, $w_\alpha(t^*) \leq -b < 0$. By intermediate value theorem $\exists \tau \in (0, t^*)$ such that $w_\alpha(\tau) = 0$, so $w_\alpha^2(\tau) = 0 < w_\alpha^2(0) \geq b^2$, which contradicts with (20) $\qquad\square$

*Proof of Theorem 4.4 (Convergence to Zero Training Error).* **Gradient Flow for the Residuals and the Loss.** We define the residual (error) on the $n$-th example

$$\mathbf{D}^{(n)} = \mathbf{SA}^{\text{lin}}\left(\mathbf{X}^{(n)}\right) - \mathbf{y}^{(n)}.$$

Let $t$ denote the (continuous) gradient-descent time, with

$$\frac{d\mathbf{C}}{dt} = -\eta\,\frac{\partial L}{\partial \mathbf{C}}, \quad \frac{d\mathbf{w}}{dt} = -\eta\,\frac{\partial L}{\partial \mathbf{w}}.$$

Consider the time derivative of the residual

$$\frac{d\mathbf{D}^{(m)}}{dt} = \frac{\partial \mathbf{SA}^{\text{lin}}\left(\mathbf{X}^{(m)}\right)}{\partial \mathbf{C}}\frac{d\mathbf{C}}{dt} + \frac{\partial \mathbf{SA}^{\text{lin}}\left(\mathbf{X}^{(m)}\right)}{\partial \mathbf{w}}\frac{d\mathbf{w}}{dt}.$$

Expanding each term, we substitute

$$\frac{\partial \mathbf{SA}^{\text{lin}}\left(\mathbf{X}^{(m)}\right)}{\partial C_{\mu\nu}} = \mathbf{X}^{(m)}_{:\mu}\left[\mathbf{X}^{(m)\top}\mathbf{X}^{(m)}\mathbf{w}\right]_{\nu} \text{ and } \frac{\partial \mathbf{SA}^{\text{lin}}\left(\mathbf{X}^{(m)}\right)}{\partial w_{\alpha}} = \left(\mathbf{X}^{(m)}\mathbf{C}\mathbf{X}^{(m)\top}\right)\mathbf{X}^{(m)}_{:\alpha}.$$

As for the gradient updates we subsitute

$$\frac{dC_{\mu\nu}}{dt} = -\frac{2\,\eta}{B}\sum_{n=1}^{B}\left[\mathbf{X}^{(n)\top}\mathbf{X}^{(n)}\mathbf{w}\right]_{\nu}\left(\mathbf{X}^{(n)}_{:\mu}\right)^{\top}\mathbf{D}^{(n)} \text{ and } \frac{dw_{\alpha}}{dt} = -\frac{2\,\eta}{B}\sum_{n=1}^{B}\left(\mathbf{X}^{(n)\top}\right)_{\alpha:}\left(\mathbf{X}^{(n)}\mathbf{C}\mathbf{X}^{(n)\top}\right)\mathbf{D}^{(n)},$$

Thus, we arrive to

$$\frac{d\mathbf{D}^{(m)}}{dt} = \sum_{\mu,\nu}\mathbf{X}^{(m)}_{:\mu}\left[\mathbf{X}^{(m)\top}\mathbf{X}^{(m)}\mathbf{w}\right]_{\nu}\left(-\frac{2\,\eta}{B}\right)\sum_{n=1}^{B}\left[\mathbf{X}^{(n)\top}\mathbf{X}^{(n)}\mathbf{w}\right]_{\nu}\left(\mathbf{X}^{(n)}_{:\mu}\right)^{\top}\mathbf{D}^{(n)}$$

$$+ \sum_{\alpha}\left(\mathbf{X}^{(m)}\mathbf{C}\mathbf{X}^{(m)\top}\right)\mathbf{X}^{(m)}_{:\alpha}\left(-\frac{2\,\eta}{B}\right)\sum_{n=1}^{B}\left(\mathbf{X}^{(n)\top}\right)_{\alpha:}\left(\mathbf{X}^{(n)}\mathbf{C}\mathbf{X}^{(n)\top}\right)\mathbf{D}^{(n)}.$$

Rearranging terms,

$$\frac{d\mathbf{D}^{(m)}}{dt} = -\frac{2\eta}{B}\sum_{n=1}^{B}\left[\left(\mathbf{w}^{\top}\mathbf{X}^{(m)\top}\mathbf{X}^{(m)}\mathbf{X}^{(n)\top}\mathbf{X}^{(n)}\mathbf{w}\right)\mathbf{X}^{(m)}\mathbf{X}^{(n)\top}\right.$$

$$\left. + \mathbf{X}^{(m)}\mathbf{C}\mathbf{X}^{(m)\top}\mathbf{X}^{(m)}\mathbf{X}^{(n)\top}\mathbf{X}^{(n)}\mathbf{C}\mathbf{X}^{(n)\top}\right]\mathbf{D}^{(n)}$$

Seeing that the term in the parenthesis is a scalar, we can write the same equation in terms for kronocker product for future convenience, which leads to

$$\frac{d\mathbf{D}^{(m)}}{dt} = -\frac{2\,\eta}{B}\sum_{n=1}^{B}\left[\left(\mathbf{w}^{\top}\mathbf{X}^{(m)\top}\mathbf{X}^{(m)}\mathbf{X}^{(n)\top}\mathbf{X}^{(n)}\mathbf{w}\right)\otimes\left(\mathbf{X}^{(m)}\mathbf{X}^{(n)\top}\right)\right.$$

$$\left. + \mathbf{X}^{(m)}\mathbf{C}\mathbf{X}^{(m)\top}\mathbf{X}^{(m)}\mathbf{X}^{(n)\top}\mathbf{X}^{(n)}\mathbf{C}^{\top}\mathbf{X}^{(n)\top}\right]\mathbf{D}^{(n)}. \quad (21)$$

Stacking different samples with the following definitions

$$\mathbf{D} = \begin{bmatrix} \vdots \\ \mathbf{D}^{(n)} \\ \vdots \end{bmatrix}, \ \mathbf{M} = \begin{bmatrix} \vdots \\ \left(\mathbf{w}^{\top}\mathbf{X}^{(n)\top}\mathbf{X}^{(n)}\right)\otimes\mathbf{X}^{(n)} \\ \vdots \end{bmatrix}, \ \mathbf{M_2} = \begin{bmatrix} \vdots \\ \mathbf{X}^{(n)}\mathbf{C}\mathbf{X}^{(n)\top}\mathbf{X}^{(n)} \\ \vdots \end{bmatrix},$$

we can write the Eq. 21

$$\frac{d\mathbf{D}}{dt} = -\frac{2\eta}{B} \left[ \mathbf{M}\mathbf{M}^\top + \mathbf{M}_2\mathbf{M}_2^\top \right] \mathbf{D}.$$

We can write the derivative of the loss as

$$\frac{dL^{\mathrm{MSE}}(\mathbf{C}, \mathbf{w})}{dt} = \frac{2}{B} \sum_{m=1}^{B} \mathbf{D}^{(m)\top} \frac{d\mathbf{D}^{(m)}}{dt} = -\frac{4\eta}{B^2} \mathbf{D}^\top \left[ \mathbf{M}\mathbf{M}^\top + \mathbf{M}_2\mathbf{M}_2^\top \right] \mathbf{D}.$$

Clearly, both $\mathbf{M}\mathbf{M}^\top$ and $\mathbf{M}_2\mathbf{M}_2^\top$ are positive semidefinite, so

$$\frac{dL^{\mathrm{MSE}}(\mathbf{C}, \mathbf{w})}{dt} \leq -\frac{4\eta}{B^2} \mathbf{D}^\top \mathbf{M}\mathbf{M}^\top \mathbf{D}. \tag{22}$$

Now, we will write Eq. 22 differently by re-expressing $\mathbf{D}$. Thanks to the realizability, we can write,

$$\mathbf{D}^{(n)} = \left( \mathbf{X}^{(n)} \mathbf{C} \mathbf{X}^{(n)\top} \right) \mathbf{X}^{(n)} \mathbf{w} - \left( \mathbf{X}^{(n)} \mathbf{C}^* \mathbf{X}^{(n)\top} \right) \mathbf{X}^{(n)} \mathbf{w}^*.$$

Due to Lemma C.2 $\mathbf{w}_i \neq 0$, so $\mathbf{w}^*{}_i/\mathbf{w}_i$ is defined for all $i \in [d]$. Thus we can define, $\mathrm{diag}\left( \frac{\mathbf{w}^*}{\mathbf{w}} \right)$ to be the diagonal matrix, whose entries are $\mathbf{w}^*{}_i/\mathbf{w}_i$ in order.

$$\mathbf{D}^{(n)} = \left( \mathbf{X}^{(n)} \mathbf{C} \mathbf{X}^{(n)\top} \right) \mathbf{X}^{(n)} \mathbf{w} - \left( \mathbf{X}^{(n)} \mathbf{C}^* \mathbf{X}^{(n)\top} \right) \mathbf{X}^{(n)} \mathrm{diag}\left( \frac{\mathbf{w}^*}{\mathbf{w}} \right) \mathbf{w}.$$

In the orthonormal basis, $\mathbf{X}^{(n)\top}\mathbf{X}^{(n)}$ is diagonal, which allows reordering to obtain

$$\mathbf{D}^{(n)} = \mathbf{X}^{(n)} \left[ \mathbf{C} - \mathbf{C}^* \mathrm{diag}\left( \frac{\mathbf{w}^*}{\mathbf{w}} \right) \right] \mathbf{X}^{(n)\top} \mathbf{X}^{(n)} \mathbf{w}.$$

Vectorizing (using $\mathrm{vec}(\mathbf{A}\mathbf{X}\mathbf{B}) = (\mathbf{B}^\top \otimes \mathbf{A}) \mathrm{vec}(\mathbf{X})$) yields

$$\mathbf{D}^{(n)} = \mathbf{M}^{(n)} \mathrm{vec}\left[ \mathbf{C} - \mathbf{C}^* \mathrm{diag}\left( \frac{\mathbf{w}^*}{\mathbf{w}} \right) \right].$$

Stacking over $n$ produces $\mathbf{D} = \mathbf{M} \mathrm{vec}\left( \mathbf{C} - \mathbf{C}^*\mathrm{diag}(\mathbf{w}^*/\mathbf{w}) \right)$. Thus, Eq. 22 can be written as

$$\frac{dL^{\mathrm{MSE}}(\mathbf{C}, \mathbf{w})}{dt} \leq -\frac{4\eta}{B^2} \mathbf{D}^\top \mathbf{M}\mathbf{M}^\top \mathbf{D} = -\frac{4\eta}{B^2} \mathrm{vec}\left[ \mathbf{C} - \mathbf{C}^* \mathrm{diag}\left( \frac{\mathbf{w}^*}{\mathbf{w}} \right) \right]^\top \mathbf{M}^\top \mathbf{M}\mathbf{M}^\top \mathbf{M} \mathrm{vec}\left[ \mathbf{C} - \mathbf{C}^* \mathrm{diag}\left( \frac{\mathbf{w}^*}{\mathbf{w}} \right) \right]$$

Using the Lemma C.3, the same inequality can be written as

$$\frac{dL^{\mathrm{MSE}}(\mathbf{C}, \mathbf{w})}{dt} \leq -\frac{4\eta}{B^2} \lambda_{\min}\left( \mathbf{M}^\top \mathbf{M} \right) \left\| \mathbf{M} \mathrm{vec}\left[ \mathbf{C} - \mathbf{C}^* \mathrm{diag}\left( \frac{\mathbf{w}^*}{\mathbf{w}} \right) \right] \right\|^2 = -\frac{4\eta}{B^2} \lambda_{\min}\left( \mathbf{M}^\top \mathbf{M} \right) \|\mathbf{D}\|^2,$$

where $\lambda_{\min}\left( \mathbf{M}^\top \mathbf{M} \right)$ is the minimum eigenvalue of $\mathbf{M}^\top \mathbf{M}$. Thus, if there exists a constant $\psi$ such that $\lambda_{\min}\left( \mathbf{M}^\top(t)\mathbf{M}(t) \right) \geq \psi > 0, \forall t$, then the training loss stops decreasing only when $\mathbf{D}$ reaches to all zero vector, i.e, training loss stops decreasing only when it reaches to zero, which is stated more rigorously in Lemma C.4.

**Lower Bound on the Eigenvalues of $\mathbf{M}^\top\mathbf{M}$.** We can show that

$$\lambda_{\min}\left( \mathbf{M}^\top \mathbf{M} \right) = \sigma_{\min}\left( \mathbf{M}^\top \mathbf{M} \right) = \min_{\mathbf{u}:\|\mathbf{u}\|_2=1} \|\mathbf{M}^\top \mathbf{M}\mathbf{u}\|_2, \tag{23}$$

where the first equality follow because $\mathbf{M}^\top \mathbf{M}$ is symmetric and positive semi definite. We also know

$$\mathbf{M}^\top \mathbf{M} = \sum_n \left( \mathbf{X}^{(n)\top}\mathbf{X}^{(n)} \mathbf{w}\mathbf{w}^\top \mathbf{X}^{(n)\top}\mathbf{X}^{(n)} \right) \otimes \mathbf{X}^{(n)\top}\mathbf{X}^{(n)}.$$

Defining

$$\mathbf{u} = \sum_{\mu \in \mathcal{S}} \mathbf{u}_\mu \otimes \mathbf{e}_\mu,$$

where each $\mathbf{u}_\mu \in \mathbb{R}^d$, so

$$\|\mathbf{u}\|_2^2 = \sum_{\mu \in \mathcal{S}} \|\mathbf{u}_\mu\|_2^2 = 1. \tag{24}$$

Recalling

$$\mathbf{X}^{(n)} = \begin{pmatrix} \vdots \\ \mathbf{e}_{\mathcal{X}(i)} \\ \vdots \end{pmatrix}_{i \in [L]},$$

$\mathbf{X}^{(n)\top}\mathbf{X}^{(n)}$ acts as a projection matrix onto the space $\{\mathbf{e}_{\mathcal{X}(0)}, \mathbf{e}_{\mathcal{X}(1)}, \ldots, \mathbf{e}_{\mathcal{X}(L-1)}\}$. Thus, using the mixed-product property of Kronecker product, we can write

$$\|\mathbf{M}^\top\mathbf{M}\mathbf{u}\|_2^2 = \left\|\sum_{n \in \mathcal{B}}\sum_{i \in [L]} \left(\left[\mathbf{X}^{(n)\top}\mathbf{X}^{(n)}\mathbf{w}\mathbf{w}^\top\mathbf{X}^{(n)\top}\mathbf{X}^{(n)}\right]\mathbf{u}_{\mathcal{X}(i)}\right) \otimes \mathbf{e}_{\mathcal{X}(i)}\right\|^2 \tag{25}$$

$$\|\mathbf{M}^\top\mathbf{M}\mathbf{u}\|_2^2 = \left\|\sum_{n \in \mathcal{B}}\sum_{\mu \in \mathcal{S}} \left(\left[\mathbf{X}^{(n)\top}\mathbf{X}^{(n)}\mathbf{w}\mathbf{w}^\top\mathbf{X}^{(n)\top}\mathbf{X}^{(n)}\right]\mathbf{u}_\mu\right) \otimes s_\mu^{(n)}\mathbf{e}_\mu\right\|^2$$

Recall the definition $\mathcal{B}_\mu = \{n \in \mathcal{B} : \mu \in \mathcal{X}^{(n)}\}$, so

$$\|\mathbf{M}^\top\mathbf{M}\mathbf{u}\|_2^2 = \left\|\sum_{\mu \in \mathcal{S}}\sum_{n \in \mathcal{B}_\mu} \left(s_\mu^{(n)}\left[\mathbf{X}^{(n)\top}\mathbf{X}^{(n)}\mathbf{w}\mathbf{w}^\top\mathbf{X}^{(n)\top}\mathbf{X}^{(n)}\right]\mathbf{u}_\mu\right) \otimes \mathbf{e}_\mu\right\|_2^2 \tag{26}$$

$$= \sum_{\mu \in \mathcal{S}}\left\|\sum_{n \in \mathcal{B}_\mu} s_\mu^{(n)}\left[\mathbf{X}^{(n)\top}\mathbf{X}^{(n)}\mathbf{w}\mathbf{w}^\top\mathbf{X}^{(n)\top}\mathbf{X}^{(n)}\right]\mathbf{u}_\mu\right\|_2^2$$

$$= \sum_{\mu \in \mathcal{S}}\left\|\sum_{n \in \mathcal{B}_\mu} s_\mu^{(n)}\left[\mathrm{diag}\left(\mathbf{s}^{(n)}\right)\mathbf{w}\mathbf{w}^\top\mathrm{diag}\left(\mathbf{s}^{(n)}\right)\right]\mathbf{u}_\mu\right\|_2^2$$

$$= \sum_{\mu \in \mathcal{S}}\left\|\sum_{n \in \mathcal{B}_\mu} s_\mu^{(n)}\left[\mathrm{diag}\left(\mathbf{w}\right)\mathbf{s}^{(n)}\mathbf{s}^{(n)\top}\mathrm{diag}\left(\mathbf{w}\right)\right]\mathbf{u}_\mu\right\|_2^2$$

$$= \sum_{\mu \in \mathcal{S}}\left\|\mathrm{diag}\left(\mathbf{w}\right)\left(\sum_{n \in \mathcal{B}_\mu} s_\mu^{(n)}\mathbf{s}^{(n)}\mathbf{s}^{(n)\top}\right)\mathrm{diag}\left(\mathbf{w}\right)\mathbf{u}_\mu\right\|_2^2. \tag{27}$$

Recall the $\mathbf{S}_{\mathcal{B}_\mu} = \left[\ldots \quad \mathbf{s}^{(n)} \quad \ldots\right]_{n \in \mathcal{B}_\mu}^\top$, and define

$$\mathrm{diag}\left(\mathbf{s}_\mu\right) = \begin{pmatrix} \ddots & & \\ & s_\mu^{(n)} & \\ & & \ddots \end{pmatrix}_{n \in \mathcal{B}_\mu}.$$

Thus, we reach to

$$\|\mathbf{M}^\top\mathbf{M}\mathbf{u}\|_2^2 = \sum_{\mu \in \mathcal{S}}\left\|\mathrm{diag}\left(\mathbf{w}\right)\mathbf{S}_{\mathcal{B}_\mu}^\top\mathrm{diag}\left(\mathbf{s}_\mu\right)\mathbf{S}_{\mathcal{B}_\mu}\mathrm{diag}\left(\mathbf{w}\right)\mathbf{u}_\mu\right\|_2^2.$$

Repeatedly applying the identity $\|\mathbf{A}\mathbf{z}\|_2 \geq \sigma_{\min}(\mathbf{A})\|\mathbf{z}\|_2$, where $\mathbf{A}$ and $\mathbf{z}$ are any matrix and vectors with suitable shapes,

$$\|\mathbf{M}^\top\mathbf{M}\mathbf{u}\|_2^2 \geq \sum_{\mu \in \mathcal{S}} \sigma_{\min}^4\left\{\mathrm{diag}\left(\mathbf{w}\right)\right\}\sigma_{\min}^2\left\{\mathrm{diag}\left(\mathbf{s}_\mu\right)\right\}\sigma_{\min}^4\left(\mathbf{S}_{\mathcal{B}_\mu}\right)\|\mathbf{u}_\mu\|_2^2$$

Due to Lemma C.2, $\sigma_{\min}^2\left\{\mathrm{diag}\left(\mathbf{w}\right)\right\} \geq b^2 > 0$. By definition, $\sigma_{\min}^2\left\{\mathrm{diag}\left(\mathbf{s}_\mu\right)\right\} \geq 1$. By Assumption 4.2, $\sigma_{\min}^2\left(\mathbf{S}_{\mathcal{B}_\mu}\right) \geq \zeta^2 > 0$. Because of the Equations 23 and 24, we reach to

$$\lambda_{\min}\left(\mathbf{M}^\top\mathbf{M}\right) \geq b^2\zeta^2 > 0.$$

$\square$

**Lemma C.3.** *For any matrix* $\mathbf{M}$*, and any vector* $\mathbf{x}$ *such that* $\mathbf{M}\mathbf{x}$ *is defined, then*

$$\mathbf{x}^\top \mathbf{M}^\top \mathbf{M} \mathbf{M}^\top \mathbf{M} \mathbf{x} \geq \lambda_{\min}\left(\mathbf{M}^\top \mathbf{M}\right) \|\mathbf{M}\mathbf{x}\|^2, \tag{28}$$

*where* $\lambda_{\min}\left(\mathbf{M}^\top \mathbf{M}\right)$ *corresponds to minimum eigenvalue of* $\mathbf{M}^\top \mathbf{M}$ *matrix.*

*Proof.* Obviously $\mathbf{A} = \left(\mathbf{M}^\top \mathbf{M}\right)$ is a symmetric and positive semi definite matrix, so it can be diagonalized as $\mathbf{A} = \mathbf{Q}\mathbf{\Lambda}\mathbf{Q}^\mathbf{T}$ and its square root $\mathbf{A} = \mathbf{A}^{\frac{1}{2}}\mathbf{A}^{\frac{1}{2}}$ defined uniquely $\mathbf{A}^{\frac{1}{2}} = \mathbf{Q}\mathbf{\Lambda}^{\frac{1}{2}}\mathbf{Q}^\top$. It follows that

$$\begin{aligned}
\mathbf{x}^\top \mathbf{A}^2 \mathbf{x} &= \mathbf{x}^\top \mathbf{A}^{\frac{1}{2}} \mathbf{A} \mathbf{A}^{\frac{1}{2}} \mathbf{x} \\
&= \mathbf{x}^\top \mathbf{A}^{\frac{1}{2}\top} \mathbf{Q}\mathbf{\Lambda}\mathbf{Q}^\top \mathbf{A}^{\frac{1}{2}} \mathbf{x} \\
&= \left(\mathbf{Q}^\top \mathbf{A}^{\frac{1}{2}}\mathbf{x}\right)^\top \mathbf{\Lambda} \left(\mathbf{Q}^\top \mathbf{A}^{\frac{1}{2}}\mathbf{x}\right) \\
&\geq \lambda_{\min}\|\mathbf{Q}^\top \mathbf{A}^{\frac{1}{2}}\mathbf{x}\|^2 = \lambda_{\min}\mathbf{x}^\top \mathbf{A}\mathbf{x} = \lambda_{\min}\|\mathbf{M}\mathbf{x}\|^2.
\end{aligned}$$

$\square$

**Lemma C.4** (Convergence Lemma). *Consider the equation*

$$\dot{\mathbf{x}}(t) = -\eta \mathbf{A}(t)\mathbf{x}(t). \tag{29}$$

*Assume the following conditions hold*

(i) $\mathbf{A}(t)$ *is symmetric for all* $t$.

(ii) *The eigenvalues of* $\mathbf{A}(t)$ *are lower bounded by a positive constant, i.e.,* $\lambda_{\min}(\mathbf{A}(t)) \geq \psi > 0$ *for all* $t$, *for some* $\psi > 0$.

*Then,* $\mathbf{x}(t) \rightarrow 0$ *as* $t \rightarrow \infty$.

## D. Generalization Analysis of Linear Self-Attention

Let's denote the parameters we get from training as $\hat{\mathbf{C}}$ and $\hat{\mathbf{W}}^V$. Also, to simplify the notation, just for this section, we define

$$\begin{aligned}
C_{\mu\nu} &= \mathbf{x}^\top\left(\mu\right)\mathbf{C}\mathbf{x}\left(\nu\right), \\
W_{\nu k} &= \mathbf{x}^\top\left(\nu\right)\mathbf{W}_{:,k},
\end{aligned}$$

### D.1. Proof of Theorem 4.6: Generalization

Remember that $\mathbf{B}$ is the domain embedding matrix seen in (6). Also, remember that $\{\mathbf{W}^{V\dagger}, \mathbf{C}^\dagger\}$ represent a set of parameters that generalize to population distribution.

*Proof of Theorem 4.6.* Let $\hat{\mathbf{C}}$ and $\hat{\mathbf{W}}^V$ be the parameters we get from the training. The zero training error condition corresponds to $\forall n \in \mathcal{B}$

$$\mathbf{X}^{(n)}\hat{\mathbf{C}}\mathbf{X}^{(n)\top}\mathbf{X}^{(n)}\hat{\mathbf{W}}^V = \mathbf{X}^{(n)}\mathbf{C}^\dagger\mathbf{X}^{(n)\top}\mathbf{X}^{(n)}\mathbf{W}^{V\dagger},$$

writing the same equation for each column separately, we get

$$\mathbf{X}^{(n)}\hat{\mathbf{C}}\mathbf{X}^{(n)\top}\mathbf{X}^{(n)}\hat{\mathbf{w}}^k = \mathbf{X}^{(n)}\mathbf{C}^\dagger\mathbf{X}^{(n)\top}\mathbf{X}^{(n)}\mathbf{w}^{k\dagger},$$

where $\mathbf{w}^{k\dagger}$ is defined as $k$-th column of $\mathbf{W}^{V\dagger}$. Using $\mathbf{B}^\top\mathbf{B} = \mathbf{I}$, we write it in the domain embedding base

$$\mathbf{X}^{(n)}\mathbf{B}^\top\mathbf{B}\hat{\mathbf{C}}\mathbf{B}^\top\mathbf{B}\mathbf{X}^{(n)\top}\mathbf{X}^{(n)}\mathbf{B}^\top\mathbf{B}\hat{\mathbf{w}}^k = \mathbf{X}^{(n)}\mathbf{B}^\top\mathbf{B}\mathbf{C}^\dagger\mathbf{B}^\top\mathbf{B}\mathbf{X}^{(n)\top}\mathbf{X}^{(n)}\mathbf{B}^\top\mathbf{B}\mathbf{w}^{k\dagger}$$

By Lemma C.1,

$$\mathbf{B}\mathbf{X}^{(n)\top}\mathbf{X}^{(n)}\mathbf{B}^\top = \text{diag}\left(s_\alpha^{(n)}, s_\beta^{(n)}, \dots\right)$$

Remembering the definition $\hat{C}_{\mu\nu} = \mathbf{x}(\mu)^\top \hat{\mathbf{C}} \mathbf{x}(\nu)$ and $\hat{w}_\alpha = \mathbf{x}(\alpha)^\top \hat{\mathbf{w}}$ and similar definitions for $\mathbf{C}^\dagger$ and $\mathbf{W}^{V\dagger}$, for any $\mu, \nu, \alpha \in \mathcal{S}$,

$$\sum_{\nu \in \mathcal{S}} \hat{C}_{\mu\nu} s_\nu^{(n)} \hat{w}_\nu^k = \sum_{\nu \in \mathcal{S}} C_{\mu\nu}^\dagger s_\nu^{(n)} w_\nu^{k\dagger},$$

$$\sum_{\nu \in \mathcal{S}} s_\nu^{(n)} \left( \hat{C}_{\mu\nu} \hat{w}_\nu^k - C_{\mu\nu}^\dagger w_\nu^{k\dagger} \right) = 0, \quad \forall \mu \in \mathcal{X}^{(n)} \text{ and } \forall n \in \mathcal{B} \tag{30}$$

For each specific $\mu$ and $k$, the terms in the last equation in parentheses can be construed as a vector with indices over $\nu$. Denoting that vector as $\mathbf{a}^{\mu k}$ and remembering $\mathbf{S}_{\mathcal{B}_\mu}$ definition (from Appendix A), Eq. 30 can be written as

$$\mathbf{S}_{\mathcal{B}_\mu} \mathbf{a}^{\mu k} = 0$$

Thanks to the Assumption 4.2, $\mathbf{S}_{\mathcal{B}_\mu}$ is full column rank so only solution to last equation is $\mathbf{a}^{\mu k} = \mathbf{0}$, which corresponds to $\hat{C}_{\mu\nu} \hat{w}_\nu^k = C_{\mu\nu}^\dagger w_\nu^{k\dagger} \ \forall \mu, \nu \in \mathcal{S}, \ k \in [d]$. Consequently, for any $\mathbf{X}$ which is embedding of a corresponding $\mathcal{X} \sim \mathcal{D}$

$$\mathbf{X} \hat{\mathbf{C}} \mathbf{X}^\top \mathbf{X} \hat{\mathbf{W}}^V = \mathbf{X} \mathbf{C}^\dagger \mathbf{X}^\top \mathbf{X} \mathbf{W}^{V\dagger} = \mathbf{Y}$$

Thus the learned parameters generalizes to all population, that is,

$$\mathbb{E}_{\mathcal{D}_{\mathcal{X} \times \mathcal{Y}}} \left[ \left\| \mathbf{Y} - \left( \mathbf{X} \hat{\mathbf{C}} \mathbf{X}^\top \right) \mathbf{X} \hat{\mathbf{W}}^V \right\| \right] = 0$$

$\square$

From the above proof, we can see that Assumption 4.2 not just plays a critical role in ensuring zero training error, but also leads to generalization to the entire population distribution. Specifically, this assumption ensures that any set of parameters achieving zero training error must also yield zero population error, meaning there exists no parameter set that perfectly fits the training data while still incurring some error at the population level.

### D.2. Length Generalization

In the preceding analysis, we showed that under some mild assumptions, achieving zero training error leads to robust population-level generalization for the specific length $L$ from which training data is sampled. However, many of our interaction-based experiments naturally suggest *length generalization*: the outputs of these models are not inherently tied to a fixed sequence length, nor did our experimental design depend on a specific number of entities. That brings us to the Assumption 4.7. Despite this, our test-time generalization results do not formally guarantee out-of-distribution generalization to unseen sequence lengths. In this section, we analyze the conditions on $\mathcal{D}^{L^*}$ under which the parameters that generalize to the distribution $\mathcal{D}^{L^*}$ generalize to $\mathcal{D}^{\forall L}$.

Remember that we denote $(\mathbf{C}^{\forall L}, \mathbf{W}^{V, \forall L})$ as the set of parameters that generalize to any length for the task of interest Also, we denote $k$-th column of the matrix $\mathbf{W}^{V, \forall L}$ as $\mathbf{w}^{k, \forall L}$ Let us first look at possibility of parameters that has zero error for $\mathcal{X}^{L^*} \sim \mathcal{D}^{L^*}$ but does not generalize to any other length $L$. Writing them in terms of a specific length generalizing parameters and $\mathbf{C}^\Delta$, $\mathbf{w}^\Delta$ -defined accordingly to satisfy the following equalities-, $\mathbf{C}^{L^*} = \mathbf{C}^{\forall L} + \mathbf{C}^\Delta$ and $\mathbf{w}^{k, L^*} = \mathbf{w}^{k, \forall L} + \mathbf{w}^\Delta$, $\forall \mu \in \mathcal{X}^{L^*}$ and denoting $k$-th column of the true output for the input tuple $\mathcal{X}^{L^*}$ as $\mathbf{y}^{k, L^*} \left( \mathcal{X}^{L^*} \right)$.

$$\begin{aligned} y_\mu^{k, L^*} \left( \mathcal{X}^{L^*} \right) &= \sum_{\nu \in \mathcal{X}^{L^*}} \left( C_{\mu\nu}^{\forall L} + C_{\mu\nu}^\Delta \right) \left( w_\nu^{k, \forall L} + w_\nu^{k\Delta} \right) \\ &= \sum_{\nu \in \mathcal{X}^{L^*}} \left( C_{\mu\nu}^{\forall L} w_\nu^{k, \forall L} + C_{\mu\nu}^{\forall L} w_\nu^{k, \Delta} + C_{\mu\nu}^\Delta w_\nu^{k, \forall L} + C_{\mu\nu}^\Delta w_\nu^{k, \forall L} \right) \end{aligned} \tag{31}$$

Defining $\Delta_{\mu\nu}^{L^*} = C_{\mu\nu}^{\forall L} w_\nu^{k, \Delta} + C_{\mu\nu}^\Delta w_\nu^{k, \forall L} + C_{\mu\nu}^\Delta w_\nu^{k, \forall L}$, and combining the effect of biasses on the output,

$$y_\mu^{k, L^*} \left( \mathcal{X}^{L^*} \right) = \sum_{\nu \in \mathcal{X}^{L^*}} C_{\mu\nu}^{\forall L} w_\nu^{k, \forall L} + \sum_{\nu \in \mathcal{X}^{L^*}} \Delta_{\mu\nu}^{L^*}$$

We can write the bias on the function output that does not change the function output for the inputs such that $|\mathcal{X}| = L^*$, but may change the output for inputs with other $L$, as

$$\sum_{\nu \in \mathcal{X}} \Delta_{\mu\nu}^{L^*}.$$

To illustrate this point, consider $\Delta_{\mu,\mu} = a$ and $\Delta_{\mu,\nu} = \frac{-a}{L^*-1} \; \forall \; \nu \neq \mu$, seeing that Eq. 31 is written $\forall \; \mu \in \mathcal{X}$,

$$\sum_{\nu \in \mathcal{X}^{L^*}} \Delta_{\mu\nu}^{L^*} = \Delta_{\mu\nu}^{L^*} + \sum_{\substack{\nu \in \mathcal{X}^{L^*} \\ \nu \neq \mu}} \Delta_{\mu\nu}^{L^*} = a - \sum_{\substack{\nu \in \mathcal{X}^{L^*} \\ \nu \neq \mu}} \frac{a}{L^*-1} = 0.$$

However, when we feed an input $\mathcal{X}^L \sim \mathcal{D}^L$ that $L \neq L^*$, we get

$$
\begin{aligned}
y_\mu^{k,L^*}(\mathcal{X}^L) &= \sum_{\nu \in \mathcal{X}^L} C_{\mu\nu}^{\forall \mathrm{L}} w_\nu^{k,\forall \mathrm{L}} + \sum_{\nu \in \mathcal{X}^L} \Delta_{\mu\nu}^{L^*} \\
&= y_\mu^k + \sum_{\nu \in \mathcal{X}^L} \Delta_{\mu\nu}^{L^*} \\
&= y_\mu^k + \Delta_{\mu\nu}^{L^*} + \sum_{\substack{\nu \in \mathcal{X}^L \\ \nu \neq \mu}} \Delta_{\mu\nu}^{L^*} \\
&= y_\mu^k + a - \sum_{\substack{\nu \in \mathcal{X}^L \\ \nu \neq \mu}} \frac{a}{L^*-1} \\
&= y_\mu^k + a \frac{L^*-L}{L^*-1}
\end{aligned}
$$

In particular, the last equation shows that an unseen sequence of length $L \neq L^*$ incurs an additive deviation of $a \dfrac{L^*-L}{L^*-1}$ from the desired target $y_\mu^{k,L^*}$, so the model generalizes to *every* length if and only if this bias term vanishes, i.e., $a = 0$. This will be useful in the following proof.

*Proof of Theorem 4.8.* To generalize to any length we should make sure that the population distribution for any length $L^*$ does not allow such a bias, that is, $\Delta_{\mu\nu}^{L^*} = 0 \, , \forall \, \mu, \nu$. From the previous discussion, for the set of parameters that generalize to population distribution $\mathcal{D}^{L^*}$ we know that,

$$\sum_{\nu \in \mathcal{X}^{L^*}} \Delta_{\mu\nu}^{L^*} = 0 \, , \;\; \forall \, \mathcal{X}^{L^*} \sim \mathcal{D}^{L^*} \tag{32}$$

For each $\mu$, we can think of Eq. 32 as a linear system of equations. Defining, $\delta_\mu^{L^*} = \begin{bmatrix} \Delta_{\mu\alpha}^{L^*} & \Delta_{\mu\beta}^{L^*} & \Delta_{\mu\gamma}^{L^*} & \dots \end{bmatrix}^\top \in \mathbb{R}^{|\mathcal{S}|}$, and an infinite sample extension of $\mathbf{S}_{\mathcal{B}_\mu}$ that is

$$\mathbf{S}_{\mathcal{B}_\mu}^{L^*} = \begin{bmatrix} \dots & \mathbf{s}^{L^*} & \dots \end{bmatrix}^\top_{s_\nu \neq 0}, \tag{33}$$

where $\mathbf{S}_{\mathcal{B}_\mu}^{L^*}$ is such a matrix that each row of it sums to $L^*$, Eq. 32 can be written as

$$\mathbf{S}_{\mathcal{B}_\mu^\infty} \delta^\mu = 0, \forall \; \mu \in \mathcal{S}.$$

If $\mathbf{S}_{\mathcal{B}_\mu^\infty}$ is full column rank for all $\mu$, which is satisfied by Assumption 4.2, then $\boldsymbol{\Delta} = 0$. Thus, it generalizes to any length. $\qquad\square$

Seeing that Assumption 4.2 inherently encompasses "$\mathbf{S}_{\mathcal{B}_\mu^\infty}$ is full column rank", it may seem that test generalization would always lead to length generalization, without any assumption. However, it is important to note that Assumption 4.2 is not the minimal requirement for ensuring test generalization.

*Remark* D.1. For clarity, let us look at an example class of tasks that for which $\mathbf{S}_{\mathcal{B}_\mu^\infty}^{L^*}$ is not full column rank. If there is a constraint on the distribution $\mathcal{X} \sim \mathcal{D}^{L^*}$ that the elements within $\mathcal{X}$ tuple are unique, than $\mathbf{S}_{\mathcal{B}_\mu^\infty}^{L^*}$ is not full rank since the column of $\mathbf{S}_{\mathcal{B}_\mu^\infty}^{L^*}$ that corresponds to element $\mu$ will be all 1 vector. Which is means, that there are some possible parameters that ensure zero error on $\mathcal{D}^{L^*}$ but does not generalize to $\mathcal{D}^L$ for any $L$.

*Remark* D.2. Under the realizability assumption, skip connections do not affect the values of the residues $\mathbf{D}^{(n)}$, allowing the same proof on generalization to apply in the skip-connected scenario.

**Corollary D.3.** *Defining*

$$C_{\mu\nu} = \mathbf{x}^\top (\mu) \, \mathbf{C}\mathbf{x} (\nu) \,,$$
$$W_{\nu k} = \mathbf{x}^\top (\nu) \, \mathbf{W}_{:,k},$$

*it follows from the proof of Theorem 4.8 that any* $\mathbf{C}, \mathbf{W}^V$ *that generalizes* $\forall L$ *satisfies*

$$C_{\mu\nu} W_{\nu k}^V = C_{\mu\nu}^{\forall \mathrm{L}} W_{\nu k}^{V, \forall \mathrm{L}}.$$

*Consequently, if you apply a nontrivial transformation to the parameters, all of the length generalizing parameters lead to a specific matrix that depends on the task at hand.*

If two sets of parameters for linear self-attention lead to the functionally equivalent self-attentions, i.e. they will lead to the same outputs for the same inputs for any input output pairs, then under this kind of nontrivial transformation they lead to the same matrix. Thus with this transformation we simply show that although the parameters we get after training are the different than the length generalizing parameters we designed they are functionally equivalent to the length generalizing parameters we have.

## E. Justification for Data Versatility Assumption

In this section we justify Assumption 4.2, by showing it holds under an even milder assumption shown below.

**Assumption E.1** (Positive-Definite Covariance and Bounded Norm). Let $\{\mathbf{s}^{(n)}\}_{n=1}^B \subset \mathbb{R}^{|\mathcal{S}|}$ be i.i.d. random row vectors. Suppose there exist constants $\zeta_{\min} > 0$ and $M > 0$ such that:

(A1) **Positive-Definite Covariance:** The covariance matrix

$$\Sigma := \mathrm{Cov}(\mathbf{s}^{(n)}) = \mathbb{E}\big[(\mathbf{s}^{(n)} - \mathbb{E}[\mathbf{s}^{(n)}])(\mathbf{s}^{(n)} - \mathbb{E}[\mathbf{s}^{(n)}])^\top\big] \quad \text{satisfies} \quad \Sigma \succeq \zeta_{\min}^2 \, \mathbf{I}.$$

That is, $\lambda_{\min}(\Sigma) \geq \zeta_{\min}^2 > 0$.

(A2) **Bounded Norm:** The centered vectors satisfy

$$\|\mathbf{s}^{(n)} - \mathbb{E}[\mathbf{s}^{(n)}]\|_2 \;\leq\; M \quad \text{almost surely.} \tag{34}$$

**Theorem E.2** (Full Column Rank with High Probability). *Under Assumption E.1, let*

$$\mathbf{S} \;=\; \begin{bmatrix} \mathbf{s}_1^{(n)} \\ \mathbf{s}_2^{(n)} \\ \vdots \\ \mathbf{s}_B^{(n)} \end{bmatrix} \;\in\; \mathbb{R}^{B \times |\mathcal{S}|}.$$

*Then there exist positive constant* $\gamma > 0$ *(depending on* $M$, $\zeta_{\min}$*, and* $|\mathcal{S}|$*) such that for all sufficiently large* $B$,

$$\mathbb{P}\Big[\mathrm{rank}(\mathbf{S}) < |\mathcal{S}|\Big] \;\leq\; e^{-\gamma B}.$$

*Equivalently,* $\mathbf{A}$ *is full column rank with probability at least* $1 - e^{-\gamma B}$.

*Proof.* **Step 1: Center the rows.** Define $\mathbf{a}_n := \mathbf{s}^{(n)} - \mathbb{E}[\mathbf{s}^{(n)}]$, so that $\mathbb{E}[\mathbf{a}_n] = \mathbf{0}$ and

$$\mathrm{Cov}(\mathbf{a}_n) \;=\; \mathbb{E}[\mathbf{a}_n\mathbf{a}_n^\top] \;=\; \Sigma.$$

Stack these centered rows into

$$\mathbf{X} \;=\; \begin{bmatrix} \mathbf{a}_1^\top \\ \mathbf{a}_2^\top \\ \vdots \\ \mathbf{a}_B^\top \end{bmatrix} \;\in\; \mathbb{R}^{B \times |\mathcal{S}|}.$$

Since each row of $\mathbf{S}$ differs from $\mathbf{X}$ by a constant shift, $\mathrm{rank}(\mathbf{S}) = \mathrm{rank}(\mathbf{X})$. Hence it suffices to show $\mathbf{X}$ is full column rank with high probability.

**Step 2: Expected Gram matrix lower bound.** We have

$$\mathbf{A}^\top\mathbf{A} \;=\; \sum_{n=1}^{B} \mathbf{a}_n\mathbf{a}_n^\top$$

Taking expectation,

$$\mathbb{E}[\mathbf{A}^\top\mathbf{A}] \;=\; \sum_{n=1}^{B} \mathbb{E}[\mathbf{a}_n\mathbf{a}_n^\top] \;=\; B\,\Sigma.$$

By Assumption E.1(A1), $\Sigma \succeq \zeta_{\min}^2\mathbf{I}$, hence

$$\mathbb{E}[\mathbf{A}^\top\mathbf{A}] \;\succeq\; B\,\zeta_{\min}^2\,\mathbf{I}.$$

**Step 3: Concentration via Matrix Bernstein.** Define the centered matrix

$$\mathbf{Z}_n \;:=\; \mathbf{a}_n\mathbf{a}_n^\top - \Sigma.$$

Note $\mathbb{E}[\mathbf{Z}_n] = \mathbf{0}$. Summing,

$$\mathbf{A}^\top\mathbf{A} - \mathbb{E}[\mathbf{A}^\top\mathbf{A}] \;=\; \sum_{n=1}^{B}(\mathbf{a}_n\mathbf{a}_n^\top - \Sigma) \;=\; \sum_{n=1}^{B}\mathbf{Z}_n.$$

Each $\mathbf{Z}_n$ is bounded in operator norm since

$$\|\mathbf{a}_n\mathbf{a}_n^\top\|_{\mathrm{op}} \;=\; \|\mathbf{a}_n\|_2^2 \;\leq\; M^2, \quad \|\Sigma\|_{\mathrm{op}} \;\leq\; \|\Sigma\|_{\mathrm{F}} \;\text{(finite)}.$$

Thus for a constant $R \in \mathbb{R}$,

$$\|\mathbf{Z}_n\|_{\mathrm{op}} \leq \|\mathbf{a}_n\mathbf{a}_n^\top\|_{\mathrm{op}} + \|\Sigma\|_{\mathrm{op}} = R.$$

In addition,

$$\|\mathbf{Z}_n^2\|_{\mathrm{op}} \leq \|\mathbf{Z}_n\|_{\mathrm{op}}^2 \leq R^2,$$

$$\left\|\sum_{n=1}^{B}\mathbb{E}\left[\mathbf{Z}_n^2\right]\right\|_{\mathrm{op}} \leq BR^2$$

Hence by a standard matrix Bernstein inequality (self-adjoint version) from (Tropp, 2015), there exist constant $\gamma > 0$ such that

$$\mathbb{P}\big[\|\mathbf{A}^\top\mathbf{A} - \mathbb{E}[\mathbf{A}^\top\mathbf{A}]\|_{\mathrm{op}} \;\geq\; \tfrac{1}{2}\,B\,\zeta_{\min}^2\big] = \mathbb{P}\left[\left\|\sum_{n=1}^{B}\mathbf{Z}_n\right\|_{\mathrm{op}} \;\geq\; \tfrac{1}{2}\,B\,\zeta_{\min}^2\right] \;\leq\; e^{-\gamma\,B}.$$

In other words, with high probability, $\mathbf{A}^\top\mathbf{A}$ stays within half its expected value in spectral norm.

**Step 4: Weyl's inequality implies strict positivity.** On this high-probability event,

$$\lambda_{\min}(\mathbf{A}^\top\mathbf{A}) \;\geq\; \lambda_{\min}\big(\mathbb{E}[\mathbf{A}^\top\mathbf{A}]\big) \;-\; \big\|\mathbf{A}^\top\mathbf{A} - \mathbb{E}[\mathbf{A}^\top\mathbf{A}]\big\|_{\mathrm{op}} \;\geq\; B\,\zeta_{\min}^2 \;-\; \tfrac{1}{2}\,B\,\zeta_{\min}^2 \;=\; \tfrac{1}{2}\,B\,\zeta_{\min}^2.$$

Hence $\mathbf{A}^\top\mathbf{A}$ is strictly positive-definite, implying $\mathrm{rank}(\mathbf{A}) = |\mathcal{S}|$. Consequently, $\mathbf{A}$ is full column rank with probability at least $1 - e^{-\gamma B}$. $\qquad\square$

*Remark* E.3 (Justification for Assumption E.1). In many natural data-generation processes, these assumptions hold:

- **Count Vectors from a Dictionary.** Suppose each sample $\mathcal{X}^{(n)}$ is a tuple of $L$ elements drawn from a vocabulary $\mathcal{S}$. The row vector $\mathbf{s}^{(n)}$ may represent counts $(s_\alpha^{(n)}, s_\beta^{(n)}, \dots)$ of how many times each element $\alpha, \beta, \dots$ appears. If we focus on the subset $\mathcal{B}_\mu$ of samples that *contain* $\mu$, then $s_\mu^{(n)} \geq 1$, while the other $L - 1$ slots of the sequence are drawn from $\mathcal{S} \setminus \{\mu\}$ according to some distribution.

- **Positive-Definite Covariance.** When these $(L - 1)$ "remaining" elements are distributed in a *non-degenerate* way (e.g., at least some variability in how the other vocabulary items appear), the resulting count vectors $\mathbf{s}^{(n)}$ will have a covariance $\Sigma$ whose minimum eigenvalue is strictly positive. For instance, under a uniform choice of the $L - 1$ positions among the $|\mathcal{S}| - 1$ possible elements, straightforward calculations show each coordinate has nonzero variance and $\lambda_{\min}(\Sigma) > 0$.

- **Bounded Norm.** Since $0 \leq s_\nu^{(n)} \leq L$ for each element $\nu \in \mathcal{S}$, the count vector $\mathbf{a}_n$ is trivially bounded by $\sqrt{|\mathcal{S}|}\, L$ in Euclidean norm. Thus we can take $M = \sqrt{|\mathcal{S}|}\, L$, satisfying Assumption E.1(A2).

- **General Distributions.** Even more general scenarios (e.g., non-uniform sampling, correlated draws) satisfy the same assumptions, *provided* negative correlations are not too extreme to force $\Sigma$ to have a zero eigenvalue. In practice, real-world data tends to have enough variability so that $\text{Cov}(\mathbf{s}^{(n)})$ is well-conditioned, meeting the requirement $\lambda_{\min}(\Sigma) \geq \zeta_{\min}^2$ for some $\zeta_{\min} > 0$.

# F. HyperFeatureAttention

### F.1. Motivation

The preceding discussions in Appendix B.1 on value functions that depend on single coordinates, both coordinates, and both coordinates with nonidentical agents reveal a fundamental pattern: as the number of features describing agents increases, the embedding dimension sufficient to fully capture pairwise interactions grows *exponentially*. As a quick recap of representations:

- For a single coordinate (e.g., $x$), the embedding dimension is proportional to $|\mathcal{S}_x| = N$, the number of possible positions along the $x$-axis.

- When extending to two coordinates ($x$ and $y$), the domain becomes $\mathcal{S} = \mathcal{S}_x \times \mathcal{S}_y$, resulting in an embedding dimension of $|\mathcal{S}| = N^2$, reflecting all possible 2D positions.

- Adding agent-specific policies (e.g., $\mathcal{S}_q = \{\text{R}, \text{U}\}$) introduces an additional multiplicative factor, so the domain expands to $\mathcal{S} = \mathcal{S}_q \times \mathcal{S}_x \times \mathcal{S}_y$, with $|\mathcal{S}| = 2N^2$.

- We can even add a new feature called "species", from a different nature, such as species $\in \{\text{H}, \text{P}\}$, which would make the domain size even larger.

Now let us look at the colliding agents environments from Appendix B.1 and Appendix B.1.3, with a slightly more complex example, to motivate the *novel* HyperFeatureAttention.

**Non-Identical Agents Revisited.** Consider $L$ agents on a $\mathcal{S}_\mathbf{r} = [N] \times [N]$ grid, each with initial coordinates $\mathbf{r}_i = \begin{bmatrix} n_i^x & n_i^y \end{bmatrix} \in [N]^2$. The agents have identities from $\ell_i \in \mathcal{S}_{\text{spcs}} = \{\text{H}, \text{P}\}$. The Hs get $+1$ reward if they catch (collide with) a P, but the Ps get $-1$ reward when they are caught. The agents also have fixed policies $q_i \in \mathcal{S}_q = \{\text{R}, \text{U}\}$, that agents with policy R always moves to right and policy U always moves to up. From our earlier step by step results in Appendix B.1.3 the value functions for the agents can be written as

$$V_i = \sum_{\substack{j \in [L] \\ j \neq i}} f\left(n_i^x, n_i^y, q_i, \ell_i, n_j^x, n_j^y, q_j, \ell_j\right) w_{n_j^x, n_j^y, q_j, \ell_j}. \tag{35}$$

With our earlier discussion (Appendix B), we know that a linear self-attention needs $\mathcal{O}(|\mathcal{S}|^2 |\mathcal{S}_{\text{spcs}}|^2 |\mathcal{S}_q|^2)$ parameters to represent such functions, which is exponential in the number of features.

However, after careful thinking on each case in our experiment, one can write the value functions as

$$
\begin{aligned}
V_i \;=\; \sum_{j \in [L]:\, j \neq i} \Big\{ &\mathbb{I}\{\ell_i = \mathrm{P},\, \ell_j = \mathrm{H}\}\,\mathbb{I}\{q_i = \mathrm{R},\, q_j = \mathrm{U}\}\,\mathbb{I}\{n_i^x - n_j^x \approx n_i^y - n_j^y\} \\
&+\mathbb{I}\{\ell_i = \mathrm{H},\, \ell_j = \mathrm{P}\}\,\mathbb{I}\{q_i = \mathrm{R},\, q_j = \mathrm{U}\}\,\mathbb{I}\{n_i^x - n_j^x \approx n_i^y - n_j^y\} \\
&+\mathbb{I}\{\ell_i = \mathrm{P},\, \ell_j = \mathrm{H}\}\,\mathbb{I}\{q_i = \mathrm{U},\, q_j = \mathrm{R}\}\,\mathbb{I}\{n_i^x - n_j^x \approx n_i^y - n_j^y\} \\
&+\mathbb{I}\{\ell_i = \mathrm{H},\, \ell_j = \mathrm{P}\}\,\mathbb{I}\{q_i = \mathrm{U},\, q_j = \mathrm{R}\}\,\mathbb{I}\{n_i^x - n_j^x \approx n_i^y - n_j^y\} \\
&+\mathbb{I}\{\ell_i = \mathrm{P},\, \ell_j = \mathrm{H}\}\,\mathbb{I}\{q_i = \mathrm{R},\, q_j = \mathrm{R}\}\,\mathbb{I}\{n_i^y \approx n_j^y\} \\
&+\mathbb{I}\{\ell_i = \mathrm{H},\, \ell_j = \mathrm{P}\}\,\mathbb{I}\{q_i = \mathrm{R},\, q_j = \mathrm{R}\}\,\mathbb{I}\{n_i^y \approx n_j^y\} \\
&+\mathbb{I}\{\ell_i = \mathrm{P},\, \ell_j = \mathrm{H}\}\,\mathbb{I}\{q_i = \mathrm{U},\, q_j = \mathrm{U}\}\,\mathbb{I}\{n_i^x \approx n_j^x\} \\
&+\mathbb{I}\{\ell_i = \mathrm{H},\, \ell_j = \mathrm{P}\}\,\mathbb{I}\{q_i = \mathrm{U},\, q_j = \mathrm{U}\}\,\mathbb{I}\{n_i^x \approx n_j^x\} \Big\} \left(\mathbb{I}\{\ell_j = \mathrm{P}\} - \mathbb{I}\{\ell_j = \mathrm{H}\}\right).
\end{aligned}
$$

Defining

$$
\begin{aligned}
f_{a_1}^{h_1}(\ell_i, \ell_j) &= \mathbb{I}\{\ell_i = \mathrm{P},\, \ell_j = \mathrm{H}\} + \mathbb{I}\{\ell_i = \mathrm{H},\, \ell_j = \mathrm{P}\}, \\
f_{a_2}^{h_1}(q_i, q_j) &= \mathbb{I}\{q_i = \mathrm{R},\, q_j = \mathrm{U}\} + \mathbb{I}\{q_i = \mathrm{U},\, q_j = \mathrm{R}\}, \\
f_{a_3}^{h_1}(\mathbf{r}_i, \mathbf{r}_j) &= \mathbb{I}\{n_i^x - n_j^x \approx n_i^y - n_j^y\}, \\
f_{a_1}^{h_2}(\ell_i, \ell_j) &= \mathbb{I}\{\ell_i = \mathrm{P},\, \ell_j = \mathrm{H}\} + \mathbb{I}\{\ell_i = \mathrm{H},\, \ell_j = \mathrm{P}\}, \\
f_{a_2}^{h_2}(q_i, q_j) &= \mathbb{I}\{q_i = \mathrm{R},\, q_j = \mathrm{R}\}, \\
f_{a_3}^{h_2}(n_i^y, n_j^y) &= \mathbb{I}\{n_i^y \approx n_j^y\}, \\
f_{a_1}^{h_3}(\ell_i, \ell_j) &= \mathbb{I}\{\ell_i = \mathrm{P},\, \ell_j = \mathrm{H}\} + \mathbb{I}\{\ell_i = \mathrm{H},\, \ell_j = \mathrm{P}\}, \\
f_{a_2}^{h_3}(q_i, q_j) &= \mathbb{I}\{q_i = \mathrm{U},\, q_j = \mathrm{U}\}, \\
f_{a_3}^{h_3}(n_i^x, n_j^x) &= \mathbb{I}\{n_i^x \approx n_j^x\}, \\
w_{a_j}^{h_i}(\ell_j) &= \mathbb{I}\{\ell_j = \mathrm{P}\} - \mathbb{I}\{\ell_j = \mathrm{H}\},
\end{aligned}
$$

and $\mathcal{H} = \{h_1, h_2, h_3\}$, $\mathcal{H} = \{a_1, a_2, a_3\}$ the same equation can be organized into

$$
V_i \;=\; \sum_{h \in \mathcal{H}} \left[ \sum_{j \in [L]} \left( \prod_{a \in \mathcal{A}} f_a^h\left(\phi_{a,i}^h, \theta_{a,j}^h\right) \right) \left( \prod_{a \in \mathcal{A}} w_a^h\left(\gamma_{a,j}^h\right) \right) \right], \tag{36}
$$

where $\phi_{a,i}^h$, $\theta_{a,j}^h$ $\gamma_{a,j}^h$ are the corresponding features picked by the corresponding functions.[10] Owing to our discussion from Appendix B, we can easily conclude that the functions $f_{a_1}^h$ can be represented *exactly* with an attention score matrix of $\mathbf{C}_{a_1}^h \in \mathbb{R}^{|\mathcal{S}_{\mathrm{spcs}}| \times |\mathcal{S}_{\mathrm{spcs}}|}$ similarly for $f_{a_2}^h$ we need $\mathbf{C}_{a_2}^h \in \mathbb{R}^{|\mathcal{S}_q| \times |\mathcal{S}_q|}$ and for $f_{a_3}^h$ we need $\mathbf{C}_{a_3}^h \in \mathbb{R}^{|\mathcal{S}_{\mathbf{r}}| \times |\mathcal{S}_{\mathbf{r}}|}$. As a result total number of parameters required is $\mathcal{O}(|\mathcal{S}_{\mathrm{spcs}}|^2 + |\mathcal{S}_q|^2 + |\mathcal{S}_{\mathbf{r}}|^2)$ which is linear in the number of features, much better than linear self-attention which was exponential. We can calculate the exact number of parameters instead of the big-$\mathcal{O}$ versions. For $N = 360$, approximately $10^6$ parameters is sufficient for HyperFeatureAttention, while self-attention requires approximately $10^7$ parameters.

**Implications for Attention Mechanisms.** While the linear self-attention mechanism discussed in Theorem 3.1 can represent pairwise interactions exactly, its parameter requirements also scale with the embedding dimension $d$. This limits its applicability in cases where the number of features (or their cardinality) is very large. For instance, in scenarios with additional discrete features such as temporal information, behavioral categories, or hierarchical roles, the exponential growth in $|\mathcal{S}|$ quickly becomes a bottleneck.

The exponential growth observed here highlights the need for alternative attention mechanisms that can efficiently handle high-dimensional domains without explicitly embedding all feature combinations. This sets the stage for the introduction of

---

[10]Equation 36 is more general than the toy example discussed above: in that example a single $(a, h)$ pair selects the same feature for its keys and queries, whereas the equation is written to accommodate cases where the selected features may differ. Therefore, we used different symbols $\phi, \theta$

a novel attention module *HyperFeatureAttention*, designed specifically to address exponential embedding growth while preserving the ability to model complex interactions across diverse features.

Also, one may concern that in practice we use layers of multi-head attention, which may possibly express Eq. 3, without exponential embedding dimension. However, we briefly go over in Theorem F.6 and the following justification that even two layer multihead linear self-attention cannot express (3) (which is easily expressed by a single layer single head HyperFeatureAttention).[11]

### F.2. HyperFeatureAttention Definition

The non-identical agents revised discussion was just a toy example for setting the stage for our novel *HyperFeatureAttention* model. Let us first generalize the discussion. We have $\Phi$ as the set of all features and $M = |\Phi|$ as total number of features in our setting that are distinct in nature. For feature $\phi$, denote its domain $\mathcal{S}_\phi$, domain size $|\mathcal{S}_\phi|$, and allocated embedding size $d_\phi$. From the previous discussion, to represent any function of the form Eq (36) the total embedding dimension for linear self-attention grows exponentially as $d = \prod_{\phi \in \Phi} |\mathcal{S}_\phi|$ and the corresponding number of parameters $\mathcal{O}\left(\prod_{\phi \in \Phi} |\mathcal{S}_\phi|^2\right)$. However, using a function of the form,

$$\mathbf{HFA}^{\text{lin}}(\mathbf{X}) \ = \ \sum_{h \in \mathcal{H}} \left[ \left( \prod_{a \in \mathcal{A}}^{\odot} \mathbf{X}\mathbf{C}^{(h,a)}\mathbf{X}^\top \right) \left( \prod_{a \in \mathcal{A}}^{\odot} \mathbf{X}\mathbf{W}^{V,(h,a)} \right) \right],$$

we only need embedding dimension of $d = \sum_{\phi \in \Phi} |\mathcal{S}_\phi|$ and the corresponding number of parameters grows linearly in the number of features $\mathcal{O}(M)$.

*Remark* F.1 (Note on Approximate Embeddings). Although the following discussion is somewhat *perpendicular* to our main focus on how entities (or features) interact, we include it briefly for completeness. In principle, self-attention may require an embedding dimension exponential in the number of features, but in practice, one often leverages the Johnson–Lindenstrauss lemma: in high-dimensional spaces, random projections yield vectors that are *approximately* orthonormal with embedding dimension only linear in the number of features. Concretely, each feature $\phi$ typically contributes $\mathcal{O}(\log |\mathcal{S}_\phi|)$ dimensions rather than $\mathcal{O}(|\mathcal{S}_\phi|)$, so total embedding dimension becomes $d = \mathcal{O}(|\Phi|)$.[12] However, in HyperFeatureAttention module, this same Johnson-Lindenstrauss argument implies a dimension requirement of $\mathcal{O}(\log |\Phi|)$ (rather than $\mathcal{O}(|\Phi|)$). Hence, even though both methods rely on approximate embeddings, *the exponential gap remains*: standard self-attention requires dimension linear in the number of features, whereas our module reduces it to logarithmic.

After all these motivations, we can now *formally* define the Multihead *HyperFeature* Attention. First, we provide the definition of Multihead Self Attention in (Vaswani et al., 2023) here for comparison.

**Definition F.2** (Multihead Self-Attention). Let $\mathbf{X} \in \mathbb{R}^{T \times d}$ denote the input sequence of $T$ tokens, where $d$ is the embedding dimension. Multihead self-attention computes a sequence of contextualized embeddings as follows:

$$\mathbf{SA}^h = \text{Softmax}\left(\mathbf{Q}^h (\mathbf{K}^h)^\top\right) \mathbf{V}^h, \quad \forall h \in \{1, \ldots, H\}, \tag{37}$$

$$\mathbf{MHA}(\mathbf{X}) = \text{Concat}(\mathbf{SA}^1, \ldots, \mathbf{SA}^H)\mathbf{W}^O, \tag{38}$$

where:

---

[11]In this comparison, we deliberately bias the setup toward standard self-attention by pitting a two-layer, multi-head SA model against a single-layer, single-head HFA.

[12]Of course, some features are more crucial than others and thus allocated more dimensions. In addition, while simpler or ordered features (e.g. spatial coordinates) can often be embedded in far fewer dimensions. For example, spatial $x$-coordinates, are intrinsically one-dimensional, allowing them to be embedded into a much lower-dimensional vector space than $|\mathcal{S}_x|$. This is because $x$-coordinates form an ordered set with a linear relationship between positions. Conversely, features like the nucleotide bases $A$, $G$, $T$, and $C$ lack such linear relationships —there is no linear relationship between "adenineness" or "guanineness"— and therefore, these features tend to occupy a 4-dimensional vector space. Some features, such as color, lie somewhere in between. Scientifically, a single frequency (e.g., 400–484 THz) is sufficient to specify the color "rose red," but in practical encodings like RGB, it is represented by three integers (e.g., [255, 3, 62]). However, in the language it is even more complex. As explained in (Jensen, 2022), the qualities of the color red, such as its warmth or contrast with green, cannot be deduced solely from its frequency as an electromagnetic wave. Even less can its emotional associations —such as romance or warmth— be reduced to a property of the wave. Instead, these properties emerge from the collective processes generated by light absorbed through the eyes, which trigger a hierarchy of neural processes in the brain. These processes ultimately lead to thoughts and emotions that we experience as color perception.

- $\mathbf{Q}^h = \mathbf{X}\mathbf{W}^{Q^h}$, $\mathbf{K}^h = \mathbf{X}\mathbf{W}^{K^h}$, and $\mathbf{V}^h = \mathbf{X}\mathbf{W}^{V^h}$ are the query, key, and value matrices for the $h$-th head, respectively.

- $d_h = \frac{d}{H}$ is the dimension of each attention head.

- $\mathbf{W}^{Q^h}, \mathbf{W}^{K^h}, \mathbf{W}^{V^h} \in \mathbb{R}^{d \times d_h}$ and $\mathbf{W}^O \in \mathbb{R}^{d \times d}$ are learnable weight matrices.

- Softmax$(\cdot)$ is applied along the last dimension.

**Definition F.3** (Multihead HyperFeatureAttention). We define two versions of Multihead HyperFeatureAttention, "value-product" and "non-value-product" respectively. Let $\mathbf{X} \in \mathbb{R}^{T \times d}$ denote the input sequence of $T$ tokens, where $d$ is the embedding dimension.

$$\mathbf{HFA}^h = \begin{cases} \text{Softmax}\left(\prod^{\odot}_{a \in [A]} \mathbf{Q}^{(h,a)}(\mathbf{K}^{(h,a)})^{\top}\right) \prod^{\odot}_{a \in [A]} \mathbf{V}^{(h,a)}, & \text{if value product,} \\ \text{Softmax}\left(\prod^{\odot}_{a \in [A]} \mathbf{Q}^{(h,a)}(\mathbf{K}^{(h,a)})^{\top}\right) \mathbf{V}^{(h)}, & \text{otherwise,} \end{cases} \tag{39}$$

$$\mathbf{MHFA}(\mathbf{X}) = \text{Concat}(\mathbf{HFA}^1, \ldots, \mathbf{HFA}^H)\mathbf{W}^O, \tag{40}$$

where:

- $\mathbf{Q}^{(h,a)} = \mathbf{X}\mathbf{W}^{Q^{(h,a)}}$, $\mathbf{K}^{(h,a)} = \mathbf{X}\mathbf{W}^{K^{(h,a)}}$, and $\mathbf{V}^{(h,a)} = \mathbf{X}\mathbf{W}^{V^{(h,a)}}$ are the query, key, and value matrices for the $h$-th head $a$-th attention score, respectively ($\mathbf{V}^{(h)} = \mathbf{X}\mathbf{W}^{V^{(h)}}$ for no value product case).

- For each head we specify $d_h$ such that $\sum_h d_h = d$ and the attention size for that head is $d_a^h = d_h/A$.

- $\mathbf{W}^{Q^{(h,a)}}, \mathbf{W}^{K^{(h,a)}} \in \mathbb{R}^{d \times d_a^h}$, $\mathbf{W}^{V^{(h,a)}}, \mathbf{W}^{V^{(h)}} \in \mathbb{R}^{d \times d_h}$ and $\mathbf{W}^O \in \mathbb{R}^{d \times d}$ are learnable weight matrices.

- $\prod^{\odot}$ represents *Hadamard* product of matrices and $[A] = \{1, \ldots, A\}$

- Softmax$(\cdot)$ is applied along the last dimension.

*Remark* F.4. In the above definition the no value product HFA has the same number of parameters as SA (assuming same $d$), yet value product version has slightly more parameters depending on the orders.

*Remark* F.5 (Rotary Positional Embedding). One can easily incorporate rotary positional embedding into this module by applying the corresponding rotation matrices $\mathbf{R}$ as $\mathbf{RQ}^{(h,a)}$ and $\mathbf{RK}^{(h,a)}$ to keys and queries just as they are explained in (Su et al., 2023).

Lastly, one may concern that in practice we use layers of multi-head attention, which may possibly express Eq. 3, without exponential embedding dimension. However, we show in the following remark that even two layer multihead linear self-attention cannot express (3) (which is easily expressed by a single layer single head HyperFeatureAttention). In this comparison, we deliberately bias the setup toward standard self-attention by pitting a two-layer, multi-head SA model against a single-layer, single-head HFA.

*Remark* F.6 (Limitations of Two-Layer Multihead Linear Self-Attention). Two-layer multihead linear self-attention cannot represent factorized cross-feature interaction functions of the form

$$\sum_j f^{(1)}(a_i, a_j)f^{(2)}(b_i, b_j), \quad \text{or} \quad \sum_j f^{(1)}(a_i, b_j)f^{(2)}(b_i, a_j), \quad \text{or similar variants,} \tag{41}$$

where the token embedding is defined as $\mathbf{x}_i = \text{concat}(a_i, b_i)$.

**Brief justification of Remark F.6.** Consider a two-layer multihead linear self-attention model. Each layer consists of $H$ attention heads, where the output of each head is computed as:

$$\mathbf{SA}_i^{(h)} = \sum_{j=1}^{L} (\mathbf{x}_i^{\top}\mathbf{C}^{(h)}\mathbf{x}_j)\mathbf{w}^{(h)}(\mathbf{x}_j), \quad \text{for } m = 1, \ldots, H,$$

with $\mathbf{C}^{(h)} \in \mathbb{R}^{d \times d}$ as the attention matrix for head $h$, and $\mathbf{w}^{(h)}(\cdot)$ as a linear map. The outputs of the heads are concatenated and optionally projected using a weight matrix $\mathbf{W}^O$. For a single-layer multihead attention, the overall output for token $i$ is a linear combination of bilinear terms of the form:

$$\mathbf{MHA}_i = \sum_{h=1}^{H} \sum_{j=1}^{L} \left(\mathbf{x}_i^\top \mathbf{C}^{(h)} \mathbf{x}_j\right) \mathbf{w}^{(h)}(\mathbf{x}_j).$$

In a two-layer model, the first layer computes:

$$\mathbf{h}_i = \text{LinearCombine}\left\{ \sum_{j=1}^{L} \left(\mathbf{x}_i^\top \mathbf{C}^{(h)} \mathbf{x}_j\right) \mathbf{w}^{(h)}(\mathbf{x}_j), \; h = 1, \ldots, H_1 \right\},$$

and passes $\{\mathbf{h}_i\}_{i \in [T]}$ as input to the second layer. The second layer then computes:

$$\mathbf{SA}_i^{(p)} = \sum_{k=1}^{L} \left(\mathbf{h}_i^\top \mathbf{C}^{(p)} \mathbf{h}_k\right) \mathbf{v}^{(p)}(\mathbf{h}_k), \quad \text{for } p = 1, \ldots, H_2.$$

By definition, $\mathbf{h}_i$ is a linear combination of sums of bilinear terms in $\mathbf{x}_i$ and $\mathbf{x}_j$. Therefore, $\mathbf{h}_i$ remains a sum of bilinear expressions across $\{\mathbf{x}_i, \mathbf{x}_j\}$.

The second layer operates on $\mathbf{h}_i$ and computes terms of the form $\mathbf{h}_i^\top \mathbf{C}^{(p)} \mathbf{h}_k$. Since $\mathbf{h}_i$ itself is bilinear, this results in expressions that are *multi-bilinear* in $\mathbf{x}_i, \mathbf{x}_j, \mathbf{x}_k$. However, it does not introduce multiplicative interactions between independent subsets of features (e.g., $a_i, a_j$ vs. $b_i, b_j$).

*Factorization is absent:* The target function $\sum_j f^{(1)}(a_i, a_j) f^{(2)}(b_i, b_j)$ requires the output to be a product of two independent terms: one depending solely on $(a_i, a_j)$ and the other on $(b_i, b_j)$. Multihead attention combines bilinear terms additively, not multiplicatively, so it cannot achieve this factorization.

Suppose $f^{(1)}$ and $f^{(2)}$ are chosen such that $f^{(1)}(a_i, a_j)$ is orthogonal to $f^{(2)}(b_i, b_j)$. In this case, any additive combination of bilinear terms cannot approximate the product $f^{(1)}(a_i, a_j) f^{(2)}(b_i, b_j)$, regardless of how many layers or heads are used.

Even with two layers of multihead linear self-attention, the mechanism remains additive and cannot express functions requiring multiplicative factorization of independent feature interactions. This limitation highlights the inability of 2 layer multihead self-attention to represent cross-feature factorized interactions such as $\sum_j f^{(1)}(a_i, a_j) f^{(2)}(b_i, b_j)$ or its variants.

## G. HyperAttention

### G.1. Defining HyperAttention

Generalizing the Definition 6.1 to order $n$,

$$A_{ij_1 j_2 \ldots j_{n-1}} = \sum_{\alpha \zeta_1 \zeta_2 \ldots \zeta_{n-1}}^{d} C_{\alpha \zeta_1 \zeta_2 \ldots \zeta_{n-1}} X_{i\alpha} X_{j_1 \zeta_1} X_{j_2 \zeta_2} \ldots X_{j_{n-1} \zeta_{n-1}}$$

$$V_{j_1 j_2 \ldots j_{n-1} \tau} = \sum_{\xi_1 \xi_2 \ldots \xi_{n-1}}^{d} X_{j_1 \xi_1} X_{j_2 \xi_2} \ldots X_{j_{n-1} \xi_{n-1}} W^V_{\xi_1 \xi_2 \ldots \xi_{n-1} \tau}$$

$$\text{HA}_{i\tau}^{\text{lin}}(\mathbf{X}) = \sum_{j_1 \leq j_2 \leq \ldots j_{n-1}}^{L} A_{ij_1 j_2 \ldots j_{n-1}} V_{j_1 j_2 \ldots j_{n-1} \tau},$$

where $\mathbf{C}, \mathbf{W}^V \in \mathbb{R}^{d \times n}$ and we denote $(i, j_1, j_2, \ldots)$-th entry of a tensor $\mathbf{T}$ as $T_{ij_1 j_2 \ldots}$. Similar to how self-attention implements low rank approximation for efficiency, i.e,

$$\mathbf{C} = \mathbf{W}^Q \left(\mathbf{W}^K\right)^\top \quad \text{and} \quad C_{\alpha\beta} = \sum_{\sigma}^{R} W^Q_{\alpha\sigma} W^K_{\beta\sigma}$$

where $\mathbf{W}^Q$ and $\mathbf{W}^K \in \mathbb{R}^{d \times R}$ are chosen low rank $R < d$, where d is the maximum rank can a $d \times d$ matrix have, we can have low rank approximation of HyperAttention as

$$C_{\alpha \zeta_1 \zeta_2 \ldots \zeta_{n-1}} = \sum_{\sigma}^{R} W_{\alpha\sigma}^{Q} W_{\zeta_1\sigma}^{K^1} W_{\zeta_2\sigma}^{K^2} \ldots W_{\zeta_{n-1}\sigma}^{K^{n-1}}$$

$$W_{\xi_1 \xi_2 \ldots \xi_{n-1}\tau}^{V} = \sum_{\sigma}^{R} W_{\xi_1\sigma}^{V^1} W_{\xi_2\sigma}^{V^2} \ldots W_{\xi_{n-1}\sigma}^{V^{n-1}} W_{\tau\sigma}^{V^n},$$

where each $\mathbf{W}^Q$, $\mathbf{W}^{K^i}$, $\mathbf{W}^{V^i} \in \mathbb{R}^{d \times R}$. Similarly, we choose $R$ less than the maximum rank can an $n$ dimensional tensor have. Finally adding the non-linearity, the full definition becomes the following.

**Definition G.1** (HyperAttention with parameter sharing). Let $\mathbf{X} \in \mathbb{R}^{T \times d}$ be the input sequence of $T$ tokens, where $d$ is the embedding dimension. For the $n$-th order HyperAttention, define:

$$\mathbf{Q} = \mathbf{X}\mathbf{W}^Q \in \mathbb{R}^{T \times R},$$
$$\mathbf{K} = \mathbf{X}\mathbf{W}^K \in \mathbb{R}^{T \times R},$$
$$\mathbf{V}^1 = \mathbf{X}\mathbf{W}^{V^1} \in \mathbb{R}^{T \times R},$$
$$\mathbf{V}^2 = \mathbf{W}^{V^2} \in \mathbb{R}^{d \times R}.$$

We also define permutation mask $\mathbf{M} \in \mathbb{R}^{T \times n}$ such that,

$$M_{i,j_1,\ldots,j_{n-1}} = -\infty \left(1 - \mathbb{I}\left[j_1 \geq j_2 \geq \cdots \geq j_{n-1}\right]\right),$$

where $\mathbb{I}$ is the indicator function that is equal to $1$ if the condition satisfied, $0$ otherwise. Then, for each token index $i$ and output dimension $\tau$,

$$A_{i,j_1,\ldots,j_{n-1}} = \mathrm{Softmax}_{(j_1,\ldots,j_{n-1})}\left(M_{i,j_1,\ldots,j_{n-1}} \sum_{\sigma=1}^{R} Q_{i,\sigma} K_{j_1,\sigma} K_{j_2,\sigma} \ldots K_{j_{n-1},\sigma}\right),$$

$$V_{j_1,\ldots,j_{n-1},\tau} = \sum_{\sigma=1}^{R} V_{j_1,\sigma}^1 V_{j_2,\sigma}^1 \ldots V_{j_{n-1},\sigma}^1 V_{\tau,\sigma}^2.$$

The HyperAttention output is computed as:

$$\mathrm{HA}_{i,\tau}(\mathbf{X}) = \sum_{j_1,\ldots,j_{n-1}=1}^{T} A_{i,j_1,\ldots,j_{n-1}} V_{j_1,\ldots,j_{n-1},\tau},$$

which is equivalent to

$$\mathrm{HA}_{i,\tau}(\mathbf{X}) = \sum_{j_1 \geq \cdots \geq j_{n-1}}^{T} A_{i,j_1,\ldots,j_{n-1}} V_{j_1,\ldots,j_{n-1},\tau}.$$

For multihead HyperAttention, we compute multiple heads indexed by $h \in \{1,\ldots,H\}$, each with independent learnable weights $\mathbf{W}^{Q^h}$, $\mathbf{W}^{K^h}$, $\mathbf{W}^{V^{h,1}}$, and $\mathbf{W}^{V^{h,2}}$. The multihead HyperAttention output is given by:

$$\mathbf{MutiHeadHA}(\mathbf{X}) = \mathrm{Concat}\left(\mathbf{HA}^1(\mathbf{X}),\ldots,\mathbf{HA}^H(\mathbf{X})\right)\mathbf{W}^O,$$

where $\mathbf{HA}^h(\mathbf{X}) \in \mathbb{R}^{T \times d_h}$, $d_h = d/H$, and $\mathbf{W}^O \in \mathbb{R}^{d \times d}$ is a learnable projection matrix.

We also have another version which is very similar to the previous definition but the parameters corresponding to different keys and values are not shared.

*Remark* G.2. To increase the implicit bias one could use $M_{i,j_1,\ldots,j_{n-1}} = -\infty\left(1 - \mathbb{I}\left[j_1 > j_2 > \cdots > j_{n-1}\right]\right)$ instead of the mask stated in the definition. In this version the block would be more specialized to learning $n$-way interactions instead of lower order interactions.

**Definition G.3** (HyperAttention without parameter sharing). Let $\mathbf{X} \in \mathbb{R}^{T \times d}$ be the input sequence of $T$ tokens, where $d$ is the embedding dimension. For the $n$-th order HyperAttention, define:

$$
\begin{aligned}
\mathbf{Q} &= \mathbf{X}\mathbf{W}^Q \in \mathbb{R}^{T \times R}, \\
\mathbf{K}^m &= \mathbf{X}\mathbf{W}^{K^m} \in \mathbb{R}^{T \times R}, \quad \forall m \in \{1, \ldots, n-1\}, \\
\mathbf{V}^m &= \mathbf{X}\mathbf{W}^{V^m} \in \mathbb{R}^{T \times R}, \quad \forall m \in \{1, \ldots, n-1\}, \\
\mathbf{V}^n &= \mathbf{W}^{V^n} \in \mathbb{R}^{d \times R}.
\end{aligned}
$$

Then, for each token index $i$ and output dimension $\tau$,

$$
A_{i,j_1,\ldots,j_{n-1}} = \text{Softmax}_{(j_1,\ldots,j_{n-1})} \left( M_{i,j_1,\ldots,j_{n-1}} \sum_{\sigma=1}^{R} Q_{i,\sigma} K^1_{j_1,\sigma} K^2_{j_2,\sigma} \ldots K^{n-1}_{j_{n-1},\sigma} \right),
$$

$$
V_{j_1,\ldots,j_{n-1},\tau} = \sum_{\sigma=1}^{R} V^1_{j_1,\sigma} V^2_{j_2,\sigma} \ldots V^{n-1}_{j_{n-1},\sigma} V^n_{\tau,\sigma}.
$$

The rest of the definition is the same as the previous definition G.1.

## G.2. Representation Abilities of HyperAttention

**Theorem G.4.** *A single layer of order n linear HyperAttention (without parameter sharing), with embedding dimension $d = |\mathcal{S}|$, can represent any function of the form*

$$
\mathbf{F}_i = \sum_{j_1, j_2, \ldots, j_{n-1} \in [L]} f\left( \mathcal{X}(i), \mathcal{X}(j_1), \mathcal{X}(j_2), \ldots, \mathcal{X}(j_{n-1}) \right) w_{s(j_1), s(j_2), s(j_{n-1})}
$$

*for all elements in the sequence, i.e, $i \in [L]$. The parameter–sharing variant of HyperAttention can express the same class of functions provided that (i) the weight tensor $w$ is fully symmetric, i.e. invariant under any permutation of its indices, and (ii) the kernel $f$ is symmetric in its last $n - 1$ arguments, $\mathcal{X}(j_1), \ldots, \mathcal{X}(j_{n-1})$.*

*Proof.* Seeing that $d = |\mathcal{S}|$, the embeddings can be orthonormal. Consequently, the same arguments of the Proof of Theorem 3.1 in Appendix B follows. $\square$

### G.2.1. SKIP-TRIGRAM BUG

Let us now illustrate how higher-order dependencies may arise with a more practical example on *skip-trigrams*.

**Next-Token Prediction** A common strategy for training language models is *next-token prediction*: given a sequence of tokens $\left( \mathcal{X}(0), \mathcal{X}(1), \ldots, \mathcal{X}(L-1) \right)$, the model learns to predict the next token. In our setting, the label for training is $\mathcal{X}(L-1)$. For simplicity, we focus on the final row of the softmax self-attention output (corresponding to $\mathcal{X}(L-1)$). Concretely, the model produces a probability distribution $l$ over the vocabulary, defined as:

$$
l = \frac{\sum_{j \in [L]} \exp\big( f\big( \mathcal{X}(i), \mathcal{X}(j) \big) \big) w_{\mathcal{X}(j)}}{\sum_{j \in [L]} \exp\big( f\big( \mathcal{X}(i), \mathcal{X}(j) \big) \big)}.
$$

During inference, the next token $\hat{y}$ is then sampled from this distribution.

As shown by (Elhage et al., 2021), self-attention can learn *skip-trigrams*. For example, in a sentence fragment such as "... keep ... in [ ]," the model might predict the next token "mind," completing the phrase as "... keep ... in mind." In the context of our work, we can interpret this phenomenon in two steps: (1) self-attention identifies that "in" is influenced by "keep" (i.e., $f(\text{in}, \text{keep})$ is large), and (2) it leverages that influence to generate the token "mind."

However, as shown in (Elhage et al., 2021), the same mechanism that raises the probabilities of the correct skip-trigrams "... keep ... in mind" and "... keep ... at bay" also inadvertently increases the probabilities of the erroneous skip-trigrams "... keep ... in bay" and "... keep ... at mind." This phenomenon, known as the "skip-trigram bug," arises from how

attention influences these completions. From our interaction perspective, increasing the probability of "...keep...in mind" can be done in two ways (1) Increasing $f(\text{in, keep})$, which unfortunately also boosts the probability of "...keep...in bay." (2) Modifying $w_{\text{keep}}$ to bias the model more strongly towards "mind," which reduces the probability of "...keep...at bay." Either approach makes it challenging for the model to consistently prefer the correct completions without also amplifying incorrect ones, thereby explaining the skip-trigram bug.

**Hyper-Attention for Avoiding Skip-Trigram Bugs.** The crux of the skip-trigram bug is that a *single* self-attention head (or pairwise interaction) tries to capture the entire phrase "...keep...in mind" by boosting $f(\text{in, keep})$ alone. This inadvertently increases the probability of other completions like "...keep...in bay" whenever $w_{\text{keep}}$ also points toward "bay." However, many real-world contexts contain additional tokens that disambiguate the correct completion. For instance, the sentence "*...keep the deadline in [ ]*" strongly suggests "*mind*" over "*bay*".

In our framework of *hyper-attention*, one can introduce a *ternary* interaction term

$$f(\text{in, keep, deadline})$$

that focuses specifically on the triplet {"in", "keep", "deadline"}, allowing the model to favor "mind" without simultaneously boosting "bay." Concretely, if we let

$$f\big(\mathcal{X}(i), \mathcal{X}(j), \mathcal{X}(k)\big) \quad \text{and} \quad w_{\mathcal{X}(j), \mathcal{X}(k)}$$

govern three-way effects (rather than just pairwise $f(\text{in, keep})$), the probability of "mind" can be increased via a *higher-order* interaction $f(\text{in, keep, the\_deadline})$ specifically tailored to that context. In doing so, we need not raise *all* completions of "...keep...in [ ]," and thus avoid inadvertently increasing "...keep...in bay."

Hence, by modeling *triplet* or *higher-order* interactions, hyper-attention more flexibly captures context-specific phrases like "*keep the deadline in mind*," while suppressing incorrect ones like "*keep the deadline in bay*," mitigating the skip-trigram bug highlighted.

### G.3. Efficient Strategy to Mitigate $\mathcal{O}(L^n)$ Computation Complexity

For simplicity, we focus on third order HyperAttention but the same arguments generalize to any order. Let's first look at linear version with recalling the definition of third order linear HyperAttention (Definition 6.1) in the tensor decomposition format of Definition G.1.

$$A_{i,j_1,j_2} = \sum_{\sigma=1}^{R} Q_{i,\sigma} K^1_{j_1,\sigma} K^2_{j_2,\sigma},$$

$$V_{j_1,j_2,\tau} = \sum_{\sigma=1}^{R} V^1_{j_1,\sigma} V^2_{j_2,\sigma} V^3_{\tau,\sigma},$$

$$\text{HA}^{\text{lin}}_{i\tau}(\mathbf{X}) = \sum_{j_1 j_2}^{L} A_{ij_1 j_2} V_{j_1 j_2 \tau}.$$

Here, each equation has $\mathcal{O}(L^3 R)$ computational complexity, so the total calculation has $\mathcal{O}(L^3 R)$ complexity. We can write the same expression as

$$\text{HA}^{\text{lin}}_{i\tau}(\mathbf{X}) = \sum_{j_1 j_2}^{L} \Big\{ \sum_{\sigma=1}^{R} Q_{i,\sigma} K^1_{j_1,\sigma} K^2_{j_2,\sigma} \Big\} \Big\{ \sum_{\tilde{\sigma}=1}^{R} V^1_{j_1,\tilde{\sigma}} V^2_{j_2,\tilde{\sigma}} V^3_{\tau,\tilde{\sigma}} \Big\}. \tag{42}$$

Changing the order of summations (take the summations over $j$s first),

$$\text{HA}^{\text{lin}}_{i\tau}(\mathbf{X}) = \sum_{\tilde{\sigma}=1}^{R} \sum_{\tilde{\sigma}=1}^{R} Q_{i,\sigma} \Big\{ \sum_{j_1}^{L} K^1_{j_1,\sigma} V^1_{j_1,\tilde{\sigma}} \Big\} \Big\{ \sum_{j_2}^{L} K^2_{j_2,\sigma} V^2_{j_2,\tilde{\sigma}} \Big\} V^3_{\tau,\tilde{\sigma}}, \tag{43}$$

its computational complexity can is reduced to $\mathcal{O}(LR^2)$. Generally $R \ll L$ because $R$ simply corresponds to attention head dimension. Thus, computational complexity reduces significantly.

As for the softmax or general nonlinear version, we use techniques similar to those in (Katharopoulos et al., 2020; Choromanski et al., 2022). For general nonlinear case Eq.42 can be written as,

$$\mathrm{HA}_{i\tau}^{\mathrm{lin}}(\mathbf{X}) = \frac{\sum_{j_1 j_2}^{L} \mathrm{sim}\left(Q_{i,:}, K_{j_1,:}^1, K_{j_2,:}^2\right)\left\{\sum_{\sigma=1}^{R} V_{j_1,\sigma}^1 V_{j_2,\sigma}^2 V_{\tau,\sigma}^3\right\}}{\sum_{j_1 j_2}^{L} \mathrm{sim}\left(Q_{i,:}, K_{j_1,:}^1, K_{j_2,:}^2\right)},$$

where $\mathrm{sim}(.)$ is classical non-linearities (for softmax it is exponentiation of the attention scores). This can be approximated with a function of the form

$$\widehat{\mathrm{HA}}_{i\tau}^{\mathrm{lin}}(\mathbf{X}) = \frac{\sum_{j_1 j_2}^{L}\left(\sum_{\sigma=1}^{R_2}\phi(Q_{i:})_\sigma \phi(K_{j_1:}^1)_\sigma \phi(K_{j_2:}^2)_\sigma\right)\left\{\sum_{\sigma=1}^{R} V_{j_1,\sigma}^1 V_{j_2,\sigma}^2 V_{\tau,\sigma}^3\right\}}{\sum_{j_1 j_2}^{L}\left(\sum_{\sigma=1}^{R_2}\phi(Q_{i:})_\sigma \phi(K_{j_1:}^1)_\sigma \phi(K_{j_2:}^2)_\sigma\right)},$$

where $\phi : \mathbb{R}^R \to \mathbb{R}^{R_2}$ is a nonlinear transformation chosen accordingly to the sim and $R_2 \in \mathbb{Z}^+$ is $\mathcal{O}(R)$. (Alman & Song, 2023) show that if entries of the input matrices $\mathbf{Q}, \mathbf{K}$ are less than $o(\sqrt[3]{\log L})$ than for $\epsilon = 1/\mathrm{poly}(L)$

$$\max_{i,\tau}\left|\mathrm{HA}_{i\tau}^{\mathrm{lin}} - \widehat{\mathrm{HA}}_{i\tau}^{\mathrm{lin}}\right| \le \epsilon$$

Consequently, the same summation order change trick (43) applies for nonlinear HyperAttentions, too.

### G.4. HyperAttention Learning

For simplicity we prove the convergence of order three linear HyperAttention. However, after this proof, extension to any order is trivial. Recall Definition 6.1, which is copied here.

**Definition G.5** (Third order Linear HyperAttention)**.**

$$A_{ij_1 j_2} = \sum_{\alpha\nu_1\nu_2}^{d} C_{\alpha\zeta_1\zeta_2} X_{i\alpha} X_{j_1\zeta_1} X_{j_2\zeta_2}$$

$$V_{j_1 j_2 \tau} = \sum_{\xi_1\xi_2}^{d} X_{j_1\xi_1} X_{j_2\xi_2} W_{\xi_1\xi_2\tau}^V$$

$$\mathrm{HA}_{i\tau}^{\mathrm{lin}}(\mathbf{X}) = \sum_{j_1 \le j_2}^{L} A_{ij_1 j_2} V_{j_1 j_2 \tau},$$

where we **denote** $(i,j,k)$-th entry of a tensor $\mathbf{T}$ as $T_{ijk}$ and $\mathbf{C} \in \mathbb{R}^{d \times d \times d}$, $\mathbf{W}^V \in \mathbb{R}^{d \times d \times d_2}$.

Seeing that our main aim is understanding the attention scores, the core mechanism defining self-attention, we choose $d_2 = 1$ to simplify the convergence analysis. Thus, $\mathbf{W}^V$ is two dimensional tensor which we denote as $\mathbf{w}$ in this subsection.

**Assumption G.6** (Weak Realizability)**.** The task is realizable, i.e, there exist $\mathbf{C}^*$ and $\mathbf{w}^*$ that perfectly fits the training data.

**Theorem G.7** (Convergence of HyperAttention to Zero Training Error)**.** *Let the dimensions $d = |\mathcal{S}|$ and $d_2 = 1$. Also, let the initial parameters $\mathbf{C}(t) = \mathbf{0}$, $w_{\alpha\beta}^2(0) \ge b > 0$, $\forall i$. Then, under the assumptions G.10 and G.6, gradient flow on*

$$L^{\mathrm{MSE}}(\mathbf{C}, \mathbf{W}^V) \;=\; \frac{1}{B}\sum_{n=1}^{B}\left\|\mathbf{HA}_{\mathbf{C},\mathbf{W}^V}^{\mathrm{lin}}(\mathbf{X}^{(n)}) \;-\; \mathbf{Y}^{(n)}\right\|^2,$$

*converges to zero training error.*

Before proving the theorem we will derive the gradients and state some lemmas that are going to be useful in the proof.

**Gradients with Respect to C and w.**

$$\frac{\partial L^{\mathrm{MSE}}(\mathbf{C},\mathbf{w})}{\partial C_{\mu\nu\sigma}} = \frac{2}{B}\sum_{n=1}^{B}\left(\mathbf{HA}^{\mathrm{lin}}(\mathbf{X}^{(n)}) - \mathbf{y}^{(n)}\right)^\top \frac{\partial \mathbf{HA}^{\mathrm{lin}}(\mathbf{X}^{(n)})}{\partial C_{\mu\nu\sigma}}$$

$$\frac{\partial \mathrm{HA}_i^{\mathrm{lin}}\left(\mathbf{X}^{(n)}\right)}{\partial C_{\mu\nu\sigma}} = \sum_{\gamma}\sum_{\sigma}\sum_{k}\sum_{j\leq k} X_{i\mu}^{(n)} X_{j\nu}^{(n)} X_{j\gamma}^{(n)} X_{k\sigma}^{(n)} X_{k\theta}^{(n)} w_{\gamma\theta}$$

$$\frac{\partial L^{\mathrm{MSE}}\left(\mathbf{C},\mathbf{w}\right)}{\partial C_{\mu\nu\sigma}} = \frac{2}{B}\sum_{n}\sum_{\gamma\theta} w_{\gamma\theta}\left(\sum_{k}\sum_{j\leq k} X_{j\nu}^{(n)} X_{j\gamma}^{(n)} X_{k\sigma}^{(n)} X_{j\theta}^{(n)}\right)\left(\mathbf{X}^{(n)}\right)_{\mu,:}^{\top}\mathbf{D}^{(n)}$$

Similarly we can find the gradient with respect to $\mathbf{w}$.

$$\frac{\partial L^{\mathrm{MSE}}\left(\mathbf{C},\mathbf{w}\right)}{\partial w_{\gamma\theta}} = \frac{2}{B}\sum_{n=1}^{B}\left(\mathbf{HA}^{\mathrm{lin}}\left(\mathbf{X}^{(n)}\right) - \mathbf{y}^{(n)}\right)^{\top}\frac{\partial \mathbf{HA}^{\mathrm{lin}}\left(\mathbf{X}^{(n)}\right)}{\partial w_{\gamma\theta}}$$

$$\frac{\partial \mathbf{HA}^{\mathrm{lin}}\left(\mathbf{X}^{(n)}\right)}{\partial w_{\gamma\theta}} = \sum_{k}\sum_{j\leq k}\left(X_{i\mu} X_{j\nu} X_{k\sigma} C_{\mu\nu\sigma}\right) X_{j\gamma} X_{k\theta}$$

$$\frac{\partial L^{\mathrm{MSE}}\left(\mathbf{C},\mathbf{w}\right)}{\partial w_{\gamma\theta}} = \frac{2}{B}\sum_{n}\sum_{\mu\nu\sigma} C_{\mu\nu\sigma}\left(\sum_{k}\sum_{j\leq k} X_{j\nu}^{(n)} X_{j\gamma}^{(n)} X_{k\sigma}^{(n)} X_{j\theta}^{(n)}\right)\left(\mathbf{X}^{(n)\,\top}\right)_{\mu,:}\mathbf{D}^{(n)}$$

In this subsection, we will change our perspective and prove the things in the one hot encoding basis, similarly to what we did in Section C. That is, we define

$$X_{i\mu}^{(n)\,\mathrm{one-hot}} = \sum_{k} X_{ik}^{(n)} B_{\mu k},$$

$$C_{\mu\nu\sigma}^{\mathrm{one-hot}} = \sum_{ijk} B_{\mu i} B_{\nu j} B_{\sigma k} C_{ijk},$$

$$w_{\gamma\theta}^{\mathrm{one-hot}} = \sum_{ij} B_{\gamma i} B_{\theta j} w_{ij},$$

and use the $\mathrm{one-hot}$ encoded versions. However, again, we abuse the notation and omit the subscript in this subsection, e.g. we write $C$ but we mean the $\mathrm{one-hot}$ version $C^{\mathrm{one-hot}}$, in the rest of this subsection. Lastly, in this section, we denote $\mathbf{e}_{\mu} \in \mathbb{R}^{|\mathcal{S}|}$ as unique one-hot encoded vector for all $\mu \in \mathcal{S}$, i.e. the base vector.

Now we state some lemmas which are anologous to Lemmas C.1 and C.2.

**Lemma G.8.** *In the embedding base, $\sum_{k}\sum_{j\leq k} X_{j\nu}^{(n)} X_{j\gamma}^{(n)} X_{k\sigma}^{(n)} X_{j\theta}^{(n)}$ is diagonal in the sense that it can be written as*

$$\sum_{k}\sum_{j\leq k} X_{j\nu}^{(n)} X_{j\gamma}^{(n)} X_{k\sigma}^{(n)} X_{j\theta}^{(n)} = \Gamma_{\nu\sigma}^{(n)}\delta_{\nu\gamma}\delta_{\sigma\theta},$$

*where*

$$\Gamma_{\nu\sigma}^{(n)} = \sum_{k}\sum_{j\leq k} \delta_{\mathcal{X}^{(n)}(j),\nu}\delta_{\mathcal{X}^{(n)}(k),\sigma},$$

*and $\delta$ is kronocker delta function, that is,*

$$\delta_{\nu\gamma} = \begin{cases} 1, & \textit{if } \nu = \gamma \\ 0, & \textit{if } \nu \neq \gamma \end{cases}$$

*Proof.* Seeing that we are in the embedding base,

$$\sum_{k}\sum_{j\leq k} X_{j\nu}^{(n)} X_{j\gamma}^{(n)} X_{k\sigma}^{(n)} X_{j\theta}^{(n)} = \sum_{k}\sum_{j<k}\left[\mathbf{e}_{\mathcal{X}^{(n)}(k)}\right]_{\sigma}\left[\mathbf{e}_{\mathcal{X}^{(n)}(k)}\right]_{\theta}\left[\mathbf{e}_{\mathcal{X}^{(n)}(j)}\right]_{\nu}\left[\mathbf{e}_{\mathcal{X}^{(n)}(j)}\right]_{\gamma}$$

$$= \sum_{k}\sum_{j\leq k} \delta_{\mathcal{X}^{(n)}(j),\nu}\delta_{\mathcal{X}^{(n)}(j),\gamma}\delta_{\mathcal{X}^{(n)}(k),\sigma}\delta_{\mathcal{X}^{(n)}(k),\theta}$$

$$= \left( \sum_k \sum_{j \le k} \delta_{\mathcal{X}^{(n)}(j),\nu} \delta_{\mathcal{X}^{(n)}(k),\sigma} \right) \delta_{\nu\gamma} \delta_{\sigma\theta}$$

the last equality follows from the identity $\delta_{ij}\delta_{ik} = \delta_{ij}\delta_{jk}$. $\qquad\square$

**Lemma G.9.** *If we choose initial parameters as* $\mathbf{C}(0) = \mathbf{0}$ *and* $w_{\alpha\beta}(0) \ge b > 0$, *then* $w_{\alpha\beta}(t) \ge b > 0$, $\forall \alpha, \beta$ *and* $\forall t \ge 0$.

*Proof.* Firstly, we will show that $w_\alpha(t)^2 \ge w_\alpha(0)^2$, $\forall t$ and $\forall i$, than the statement in the lemma will follow similar to the proof of Lemma C.2. Using the previous gradient derivations and Lemma G.8,

$$\frac{dC_{\mu\nu\sigma}}{dt} = -\eta \frac{\partial L^{\mathrm{MSE}}(\mathbf{C}, \mathbf{w})}{\partial C_{\mu\nu\sigma}} = -\eta \frac{2}{B} \sum_n \sum_{\gamma\theta} w_{\gamma\theta} \left( \sum_k \sum_{j \le k} X_{j\nu}^{(n)} X_{j\gamma}^{(n)} X_{k\sigma}^{(n)} X_{j\theta}^{(n)} \right) \left( \mathbf{X}^{(n)} \right)_{\mu,:}^\top \mathbf{D}^{(n)},$$

$$= -\eta \frac{2}{B} \sum_n \sum_{\gamma\theta} w_{\gamma\theta} X_{k\sigma}^{(n)} X_{j\theta}^{(n)} \Gamma_{\nu\sigma} \delta_{\nu\gamma} \delta_{\sigma\theta} \left( \mathbf{X}^{(n)} \right)_{\mu,:}^\top \mathbf{D}^{(n)},$$

$$\frac{dw_{\gamma\theta}}{dt} = -\eta \frac{\partial L^{\mathrm{MSE}}(\mathbf{C}, \mathbf{w})}{\partial w_{\gamma\theta}} = -\eta \frac{2}{B} \sum_n \sum_{\mu\nu\sigma} C_{\mu\nu\sigma} \left( \sum_k \sum_{j \le k} X_{j\nu}^{(n)} X_{j\gamma}^{(n)} X_{k\sigma}^{(n)} X_{j\theta}^{(n)} \right) \left( \mathbf{X}^{(n)\,\top} \right)_{\mu,:} \mathbf{D}^{(n)},$$

$$= -\eta \frac{2}{B} \sum_n \sum_{\mu\nu\sigma} C_{\mu\nu\sigma} \Gamma_{\nu\sigma} \delta_{\nu\gamma} \delta_{\sigma\theta} \left( \mathbf{X}^{(n)\,\top} \right)_{\mu,:} \mathbf{D}^{(n)}.$$

Let $\mathbf{\Lambda}^a$ and $\mathbf{\Lambda}^b$ be matrices that are diagonal in the embedding base $\mathbf{B}$. However, we again abuse the notation. We do not rewrite the one $-$ hot in $\mathbf{\Lambda}^{a,b\,\mathrm{one-hot}} = \mathbf{BCB}^\top$ and denote it just as $\mathbf{\Lambda}^{a,b}$ in the rest of the proof. We can now write

$$\sum_{\mu\nu\sigma} C_{\mu\nu\sigma} \frac{dC_{\mu\nu\sigma}}{dt} \Lambda_{\nu\nu}^a \Lambda_{\sigma\sigma}^b = -\eta \frac{2}{B} \sum_n \sum_{\mu\nu\sigma} C_{\mu\nu\sigma} \Lambda_{\nu\nu}^a \Lambda_{\sigma\sigma}^b \sum_{\gamma\theta} w_{\gamma\theta} \Gamma_{\nu\sigma}^{(n)} \delta_{\nu\gamma} \delta_{\sigma\theta} \left( \mathbf{X}^{(n)\,\top} \right)_{\mu,:} \mathbf{D}^{(n)},$$

$$= -\eta \frac{2}{B} \sum_n \sum_{\mu\nu\sigma} C_{\mu\nu\sigma} w_{\nu\theta} \Lambda_{\nu\nu}^a \Lambda_{\sigma\sigma}^b \Gamma_{\nu\sigma}^{(n)} \left( \mathbf{X}^{(n)\,\top} \right)_{\mu,:} \mathbf{D}^{(n)},$$

$$\sum_{\gamma\theta} w_{\gamma\theta} \frac{dw_{\gamma\theta}}{dt} \Lambda_{\nu\nu}^a \Lambda_{\sigma\sigma}^b = -\eta \frac{2}{B} \sum_n \sum_{\gamma\theta} \Lambda_{\gamma\gamma}^a \Lambda_{\theta\theta}^b w_{\gamma\theta} \sum_{\mu\nu\sigma} C_{\mu\nu\sigma} \Gamma_{\nu\sigma}^{(n)} \delta_{\nu\gamma} \delta_{\sigma\theta} \left( \mathbf{X}^{(n)\,\top} \right)_{\mu,:} \mathbf{D}^{(n)},$$

$$= -\eta \frac{2}{B} \sum_n \sum_{\mu\nu\sigma} C_{\mu\nu\sigma} w_{\nu\theta} \Lambda_{\nu\nu}^a \Lambda_{\sigma\sigma}^b \Gamma_{\nu\sigma}^{(n)} \left( \mathbf{X}^{(n)\,\top} \right)_{\mu,:} \mathbf{D}^{(n)}.$$

Consequently we have

$$\sum_{\gamma\theta} w_{\gamma\theta} \frac{dw_{\gamma\theta}}{dt} \Lambda_{\nu\nu}^a \Lambda_{\sigma\sigma}^b = \sum_{\mu\nu\sigma} C_{\mu\nu\sigma} \frac{dC_{\mu\nu\sigma}}{dt} \Lambda_{\nu\nu}^a \Lambda_{\sigma\sigma}^b,$$

$$\frac{d}{dt} \sum_{\gamma\theta} w_{\gamma\theta}^2 \Lambda_{\nu\nu}^a \Lambda_{\sigma\sigma}^b = \frac{d}{dt} \sum_{\mu\nu\sigma} C_{\mu\nu\sigma}^2 \Lambda_{\nu\nu}^a \Lambda_{\sigma\sigma}^b,$$

$$\sum_{\gamma\theta} w_{\gamma\theta}^2(t) \Lambda_{\nu\nu}^a \Lambda_{\sigma\sigma}^b = \sum_{\gamma\theta} w_{\gamma\theta}^2(0) \Lambda_{\nu\nu}^a \Lambda_{\sigma\sigma}^b + \sum_{\mu\nu\sigma} C_{\mu\nu\sigma}^2(t) \Lambda_{\nu\nu}^a \Lambda_{\sigma\sigma}^b - \sum_{\mu\nu\sigma} C_{\mu\nu\sigma}^2(0) \Lambda_{\nu\nu}^a \Lambda_{\sigma\sigma}^b.$$

Letting $\mathbf{\Lambda}^a = \mathrm{diag}(\mathbf{e}_\alpha)$ and $\mathbf{\Lambda}^b = \mathrm{diag}(\mathbf{e}_\beta)$,

$$w_{\alpha\beta}^2(t) = w_{\alpha\beta}^2(0) + \|\mathbf{C}_{:,\alpha,\beta}(t)\|^2 - \|\mathbf{C}_{:,\alpha,\beta}(0)\|^2 = w_{\alpha\beta}^2(0) + \|\mathbf{C}_{:,\alpha,\beta}(t)\|^2$$

where the last equality follows because $\mathbf{C}(0) = \mathbf{0}$. As a result we reach to

$$w_{\alpha\beta}^2(t) \ge w_{\alpha\beta}^2(0) \ge b^2 \tag{44}$$

Seeing that $\frac{dw_{\alpha\beta}}{dt}$ is finite $\forall t$, $w_{\alpha\beta}(t)$ is continuous. As a result if $w_\alpha(0) \geq b > 0$, then $w_{\alpha\beta}(t) \geq b$, $\forall t$ which can be proven by contradiction. Assume $\exists t^* > 0$ such that $w_{\alpha\beta}(t^*) \leq b$. By Equation 44, $w_{\alpha\beta}(t^*) \leq -b < 0$. By intermediate value theorem $\exists \tau \in (0, t^*)$ such that $w_{\alpha\beta}(\tau) = 0$, so $w_{\alpha\beta}^2(\tau) = 0 < w_{\alpha\beta}^2(0) \geq b^2$, which contradicts with (44) $\qquad\square$

*Proof of Theorem G.7.* Seeing that $d_2 = 1$ we denote two dimensional reduction of the three dimensional tensor $\mathbf{W}^V$ as $\mathbf{w}$. Thus the HyperAttention formula becomes,

$$\mathrm{HA}_i^{\mathrm{lin}}\left(\mathbf{X}\right) = \sum_{j_1 \leq j_2}^L \left(\sum_{\alpha\zeta_1\zeta_2} C_{\alpha\zeta_1\zeta_2} X_{i\alpha} X_{j_1\zeta_1} X_{j_2\zeta_2}\right) \left(\sum_{\xi_1\xi_2} X_{j_1\xi_1} X_{j_2\xi_2} w_{\xi_1\xi_2}\right)$$

**Gradient Flow for the Residuals and the Loss**

$$\frac{d\mathbf{C}}{dt} = -\eta \frac{\partial L^{\mathrm{MSE}}\left(\mathbf{C}, \mathbf{w}\right)}{\partial \mathbf{C}}, \quad \frac{d\mathbf{w}}{dt} = -\eta \frac{\partial L^{\mathrm{MSE}}\left(\mathbf{C}, \mathbf{w}\right)}{\partial \mathbf{w}}$$

Following similar steps as in the proof of Theorem 4.4,

$$\frac{dD_i^{(m)}}{dt} = -\eta \sum_{\mu\nu\sigma\gamma\theta} X_{i\mu}^{(m)} \sum_k \sum_{j \leq k} X_{j\nu}^{(m)} X_{k\sigma}^{(m)} X_{j\gamma}^{(m)} X_{k\theta}^{(m)} w_{\gamma\theta} \frac{\partial L^{\mathrm{MSE}}\left(\mathbf{C}, \mathbf{w}\right)}{\partial C_{\mu\nu\sigma}}$$

$$-\eta \sum_{\nu\mu\sigma\gamma\theta} X_{i\mu}^{(m)} \sum_k \sum_{j \leq k} X_{j\nu}^{(m)} X_{k\sigma}^{(m)} C_{\mu\nu\sigma} X_{j\gamma}^{(m)} X_{k\theta}^{(m)} \frac{\partial L^{\mathrm{MSE}}\left(\mathbf{C}, \mathbf{w}\right)}{\partial w_{\gamma\theta}}$$

Substituting the gradients

$$\frac{dD_i^{(m)}}{dt} = -\frac{2\eta}{B} \sum_{\mu\nu\sigma\gamma'\theta'} \left\{ X_{i\mu}^{(m)} w_{\gamma'\theta'} \left(\sum_k \sum_{j \leq k} X_{j\nu}^{(m)} X_{k\sigma}^{(m)} X_{j\gamma'}^{(m)} X_{k\theta'}^{(m)}\right)\right.$$

$$\left.\times \sum_n \sum_{\gamma\theta} w_{\gamma\theta} \left(\sum_k \sum_{j \leq k} X_{j\nu}^{(n)} X_{j\gamma}^{(n)} X_{k\sigma}^{(n)} X_{j\theta}^{(n)}\right) \left(\mathbf{X}^{(n)}\right)_{\mu,:}^\top \mathbf{D}^{(n)} \right\}$$

$$-\frac{2\eta}{B} \sum_{\mu'\nu'\sigma'\gamma\theta} \left\{ X_{i\mu'}^{(m)} C_{\mu'\nu'\sigma'} \left(\sum_k \sum_{j \leq k} X_{j\nu'}^{(m)} X_{k\sigma'}^{(m)} X_{j\gamma}^{(m)} X_{k\theta}^{(m)}\right)\right.$$

$$\left.\times \sum_n \sum_{\mu\nu\sigma} C_{\mu\nu\sigma} \left(\sum_k \sum_{j \leq k} X_{j\nu}^{(n)} X_{j\gamma}^{(n)} X_{k\sigma}^{(n)} X_{j\theta}^{(n)}\right) \left(\mathbf{X}^{(n)\top}\right)_{\mu,:} \mathbf{D}^{(n)} \right\}$$

By Lemma G.8,

$$\frac{dD_i^{(m)}}{dt} = -\eta \frac{2}{B} \sum_{\mu\nu\sigma\gamma'\theta'} X_{i\mu}^{(m)} w_{\gamma'\theta'} \Gamma_{\nu\sigma}^{(m)} \delta_{\nu\gamma'} \delta_{\sigma\theta'} \sum_n \sum_{\gamma\theta} w_{\gamma\theta} \Gamma_{\nu\sigma}^{(n)} \delta_{\nu\gamma} \delta_{\sigma\theta} \left(\mathbf{X}^{(n)}\right)_{\mu,:}^\top \mathbf{D}^{(n)}$$

$$-\eta \frac{2}{B} \sum_{\mu'\nu'\sigma'\gamma\theta} X_{i\mu'}^{(m)} C_{\mu'\nu'\sigma'} \Gamma_{\nu'\sigma'}^{(m)} \delta_{\nu'\gamma} \delta_{\sigma'\theta} \sum_n \sum_{\mu\nu\sigma} C_{\mu\nu\sigma} \Gamma_{\nu\sigma}^{(n)} \delta_{\nu\gamma} \delta_{\sigma\theta} \left(\mathbf{X}^{(n)\top}\right)_{\mu,:} \mathbf{D}^{(n)}$$

$$= -\eta \frac{2}{B} \sum_{\mu\nu\sigma} X_{i\mu}^{(m)} w_{\nu\sigma} \Gamma_{\nu\sigma}^{(m)} \sum_n w_{\nu\sigma} \Gamma_{\nu\sigma}^{(n)} \left(\mathbf{X}^{(n)}\right)_{\mu,:}^\top \mathbf{D}^{(n)}$$

$$-\eta \frac{2}{B} \sum_{\mu'\gamma\theta} X_{i\mu'}^{(m)} C_{\mu'\gamma\theta} \Gamma_{\gamma\theta}^{(m)} \sum_n \sum_\mu C_{\mu\gamma\theta} \Gamma_{\gamma\theta}^{(n)} \left(\mathbf{X}^{(n)\top}\right)_{\mu,:} \mathbf{D}^{(n)}$$

Now let us define,

$$M_{i\mu\nu\sigma}^{(n)} = X_{i\mu}^{(n)} w_{\nu\sigma} \Gamma_{\nu\sigma}^{(n)}$$

Also define matrixize $M_{i\mu\nu\sigma}^{(n)}$ as $\mathbf{M}$ such that first dimension of $\mathbf{M}$ is vectorization of $n$ and $i$ the second dimension is vectorization of $\mu, \nu, \sigma$, so $\mathbf{M} \in \mathbb{R}^{BL \times d^3}$. There exists a similar matrix $\mathbf{M}_2$ such that

$$\frac{d\mathbf{D}}{dt} = -\eta \frac{2}{B} \left[ \mathbf{M}\mathbf{M}^\top + \mathbf{M}_2\mathbf{M}_2^\top \right]$$

Similar to what we in the proof of Theorem 4.4, it follows that

$$\frac{dL^{\mathrm{MSE}}(\mathbf{C}, \mathbf{w})}{dt} \leq -\frac{4\eta}{B^2} \mathbf{D}^\top \mathbf{M}\mathbf{M}^\top \mathbf{D} \tag{45}$$

Now, we will write Eq. 45 differently, reexpressing $\mathbf{D}$. Thanks to Assumption G.6, i.e. the realizability, we can write

$$D_i^{(n)} = \sum_{\mu\nu\sigma\gamma\theta} X_{i\mu}^{(n)} X_{j\nu}^{(n)} X_{k\sigma}^{(n)} C_{\mu\nu\sigma} X_{j\gamma}^{(n)} X_{k\sigma}^{(n)} w_{\gamma\sigma} - \sum_{\mu\nu\sigma\gamma\theta} X_{i\mu}^{(n)} X_{j\nu}^{(n)} X_{k\sigma}^{(n)} C_{\mu\nu\sigma}^* X_{j\gamma}^{(n)} X_{k\sigma}^{(n)} w_{\gamma\theta}^*.$$

By Lemma G.8,

$$D_i^{(n)} = \sum_{\mu\nu\sigma} X_{i\mu} \left( C_{\mu\nu\sigma} \Gamma_{\nu\sigma}^{(n)} w_{\nu\sigma} - C_{\mu\nu\sigma}^* \Gamma_{\nu\sigma}^{(n)} w_{\nu\sigma}^* \right)$$

By Lemma G.9, $1/w_{\nu\sigma}$ is defined, so we can write

$$D_i^{(n)} = \sum_{\mu\nu\sigma} X_{i\mu} \Gamma_{\nu\sigma}^{(n)} w_{\nu\sigma} \left( C_{\mu\nu\sigma} - C_{\mu\nu\sigma}^* \frac{w_{\nu\sigma}^*}{w_{\nu\sigma}} \right) = \sum_{\mu\nu\sigma} M_{i\mu\nu\sigma}^{(n)} \left( C_{\mu\nu\sigma} - C_{\mu\nu\sigma}^* \frac{w_{\nu\sigma}^*}{w_{\nu\sigma}} \right).$$

Now again we can vectorize along $n, i$ and vectorize along $\mu, \nu, \sigma$ which leads to

$$\mathbf{D} = \mathbf{M}\mathrm{vec} \left( \mathbf{C} - \mathbf{C}^* \frac{\mathbf{w}^*}{\mathbf{w}^*} \right).$$

Following the same steps as in the proof of Theorem 4.4,

$$\frac{dL^{\mathrm{MSE}}(\mathbf{C}, \mathbf{w})}{dt} \leq -\frac{4\eta}{B^2} \mathbf{D}^\top \mathbf{M}\mathbf{M}^\top \mathbf{D} = -\frac{4\eta}{B^2} \mathrm{vec}\left[ \mathbf{C} - \mathbf{C}^* \mathrm{diag}\left( \frac{\mathbf{w}^*}{\mathbf{w}} \right) \right]^\top \mathbf{M}^\top \mathbf{M}\mathbf{M}^\top \mathbf{M}\mathrm{vec}\left[ \mathbf{C} - \mathbf{C}^* \mathrm{diag}\left( \frac{\mathbf{w}^*}{\mathbf{w}} \right) \right]$$

Using the Lemma C.3, the same inequality can be written as

$$\frac{dL^{\mathrm{MSE}}(\mathbf{C}, \mathbf{w})}{dt} \leq -\frac{4\eta}{B^2} \lambda_{\min}\left( \mathbf{M}^\top \mathbf{M} \right) \left\| \mathbf{M}\mathrm{vec}\left[ \mathbf{C} - \mathbf{C}^* \mathrm{diag}\left( \frac{\mathbf{w}^*}{\mathbf{w}} \right) \right] \right\|^2 = -\frac{4\eta}{B^2} \lambda_{\min}\left( \mathbf{M}^\top \mathbf{M} \right) \left\| \mathbf{D} \right\|^2,$$

where $\lambda_{\min}\left( \mathbf{M}^\top \mathbf{M} \right)$ is the minimum eigenvalue of $\mathbf{M}^\top \mathbf{M}$. Thus, if there exists a constant $\psi$ such that $\lambda_{\min}\left( \mathbf{M}^\top(t)\mathbf{M}(t) \right) \geq \psi > 0$, $\forall t$, then the training loss stops decreasing only when $\mathbf{D}$ reaches to all zero vector, i.e, training loss stops decreasing only when it reaches to zero, which is stated more rigorously in Lemma C.4.

**Lower Bound on the Eigenvalues of $\mathbf{M}^\top \mathbf{M}$.**

$$\lambda_{min}\left( \mathbf{M}^\top \mathbf{M} \right) = \sigma_{\min}\left( \mathbf{M}^\top \mathbf{M} \right) = \min_{\mathbf{u}: \|\mathbf{u}\|_2 = 1} \|\mathbf{M}^\top \mathbf{M}\mathbf{u}\|_2, \tag{46}$$

where the first equality follow because $\mathbf{M}^\top \mathbf{M}$ is symmetric and positive semi definite and $\mathbf{u} \in \mathbb{R}^{d^3}$. We also know

$$\left[ \mathbf{M}^\top \mathbf{M} \right]_{\mu\nu\sigma\mu'\nu'\sigma'} = \sum_n \sum_{i \in [L]} X_{i\mu}^{(n)} \Gamma_{\nu\sigma}^n w_{\nu\sigma} X_{i\mu'}^{(n)} \Gamma_{\nu'\sigma'}^n w_{\nu'\sigma'}.$$

For ease of notation we can tensorize the things back to the $\mu, \nu, \sigma$. Thus,

$$\|\mathbf{u}\| = \sum_{\mu\nu\sigma} u_{\mu\nu\sigma}^2 = 1$$

It follows that

$$\left[\mathbf{M}^\top \mathbf{M}\mathbf{u}\right]_{\mu\nu\sigma} = \sum_{n\in\mathcal{B}}\sum_{i\in[L]}\sum_{\mu'\nu'\sigma'} X_{i\mu}^{(n)}\Gamma_{\nu\sigma}^{(n)}w_{\mu\nu}X_{i\mu'}^{(n)}\Gamma_{\nu'\sigma'}^{(n)}w_{\nu'\sigma'}u_{\mu'\nu'\sigma'} \tag{47}$$

Remembering

$$\sum_{i\in[L]} X_{i\mu}X_{i\mu'} = \sum_{i\in[L]} \delta_{\mathcal{X}(i),\mu}\delta_{\mathcal{X}(i),\mu'},$$

Eq. 47 becomes

$$\begin{aligned}
\left[\mathbf{M}^\top \mathbf{M}\mathbf{u}\right]_{\mu\nu\sigma} &= \sum_{n\in\mathcal{B}}\Gamma_{\nu\sigma}^{(n)}w_{\nu\sigma}\sum_{\mu'\nu'\sigma'}\left(\sum_{i\in[L]}\delta_{\mathcal{X}(i),\mu}\delta_{\mathcal{X}(i),\mu'}\right)\Gamma_{\nu'\sigma'}^{(n)}w_{\nu'\sigma'}u_{\mu'\nu'\sigma'}\\
&= \sum_{n\in\mathcal{B}}\sum_{i\in[L]}\sum_{\nu'\sigma'}\Gamma_{\nu\sigma}^{(n)}w_{\nu\sigma}\Gamma_{\nu'\sigma'}^{(n)}w_{\nu'\sigma'}u_{\mathcal{X}(i)^n\nu'\sigma'}
\end{aligned}$$

Recalling the definition $\mathcal{B}_\mu = \left\{n\in\mathcal{B} : \mu\in\mathcal{X}^{(n)}\right\}$, we do the same trick we did when we were getting Eq. 26 form Eq. 25, so we get

$$\left[\mathbf{M}^\top \mathbf{M}\mathbf{u}\right]_{\mu\nu\sigma} = \sum_{\mu\in\mathcal{S}}\sum_{n\in\mathcal{B}_\mu}\left(s_\mu^{(n)}\sum_{\nu'\sigma'}\Gamma_{\nu\sigma}^{(n)}w_{\nu\sigma}\Gamma_{\nu'\sigma'}^{(n)}w_{\nu'\sigma'}\right)u_{\mu\nu'\sigma'}$$

Vectorizing along $\nu,\sigma$ and $\nu',\sigma'$ we reach to

$$\mathbf{M}^\top \mathbf{M}\mathbf{u} = \sum_{\mu\in\mathcal{S}}\operatorname{diag}\left(\mathbf{w}\right)\left(\sum_{n\in\mathcal{B}_\mu}s_\mu^{(n)}\operatorname{vec}\left(\boldsymbol{\Gamma}^{(n)}\right)\operatorname{vec}^\top\left(\boldsymbol{\Gamma}^{(n)}\right)\right)\mathbf{u}_{\mu,:}. \tag{48}$$

Notice the similarity between Eq. 48 and 27. Thus, defining

$$\mathbf{Z}_{\mathcal{B}_\mu} = \begin{pmatrix} \vdots \\ \operatorname{vec}^\top\left(\Gamma^{(n)}\right) \\ \vdots \end{pmatrix}_{n\in\mathcal{B}_\mu},$$

we follow the same steps seen after Eq. 27 and reach to

$$\lambda_{\min}\left(\mathbf{M}^\top \mathbf{M}\right) \geq b^2\zeta^2,$$

where we used the bound $w_{\nu\sigma} \geq b$ and the Assumption G.10 -$\sigma_{\min}^2\left(\mathbf{Z}_{\mathcal{B}_\mu}\right) \geq \zeta^2$. $\qquad\square$

**Assumption G.10** (Training Data Versatility). For all $\mu\in\mathcal{S}$,

$$\mathbf{Z}_{\mathcal{B}_\mu} = \begin{pmatrix} \vdots \\ \operatorname{vec}^\top\left(\Gamma^{(n)}\right) \\ \vdots \end{pmatrix}_{n\in\mathcal{B}_\mu},$$

is full column rank.

## H. Comparison Between the Attention Models

We have introduced two novel models (HyperFeatureAttention in Appendix F, HyperAttention in Appendix G) and mentioned some approximations to reduce computational complexity (in Appendix G.3). In this section, letting embedding dimension to be $d$, sequence length to be $L$, we compare those models in terms of number of parameters, computational complexity, and abilities in Table H, too.

Table 3. Comparison Between Attention Models

| Model | Computational Complexity | Captures |
|---|---|---|
| Self Attention | $\Theta\left(L^2\right)$ | Mutual interactions |
| HyperFeatureAttention | $\Theta\left(L^2\right)$ | Couplings of mutual interactions |
| HyperAttention of order n | $\Theta\left(L^n\right)$ | n-way interactions |
| Linear Self Attention | $\Theta\left(L\right)$ | Mutual interactions (approximate) |
| Linear HyperFeatureAttention | $\Theta\left(L\right)$ | Couplings of mutual interactions (approximate) |
| Linear HyperAttention of order n | $\Theta\left(L\right)$ | n-way interactions (approximate) |

Starting with self-attention, it captures mutual interactions between entities.[13] If it has multiple heads, it can capture summation over mutual interactions between features of the entities. It has $\Theta(d^2)$ parameters, and its computational complexity is $\Theta(L^2)$. In order to reduce the computational complexity to $\Theta(L)$, people came up with approximations called "Linear Attention" (Katharopoulos et al., 2020; Wang et al., 2020). However, despite the name, the method is generally used to approximate softmax self-attention.

HyperFeatureAttention captures couplings of mutual interactions between features.[14] If it has multiple heads, it can capture summation over couplings of mutual interactions between features. Same as self-attention, HyperFeatureAttention has $\Theta(d^2)$ parameters, and its computational complexity is $\Theta(L^2)$. The same Linear Attention approximations can be applied to HyperFeatureAttention, reducing its computational complexity to $\Theta(L)$. Seeing that the main goal of the paper is not this approximation for HyperFeatureAttention, we did not show it explicitly.

As for HyperAttention of order $n$, it captures up to $n^{th}$ order interactions.[15] If it has multiple heads, it can capture summation over up to $n^{th}$ order interactions between features of the tokens. It has $\Theta(d^2)$ parameters, and its computational complexity is $\Theta(L^n)$. Using similar Linear Attention approximations, in Appendix G.3, we explained how to reduce computational complexity of HyperAttention to $\Theta(L)$.

One might contend that standard self-attention can, in principle, capture these complex interactions "in surprising ways." The key difference is that our modules achieve comparable expressiveness with far fewer parameters—and therefore lower memory and compute overhead. While we do not advocate replacing conventional self-attention with HyperAttention or HyperFeatureAttention, we propose these mechanisms as complementary enhancements. In a typical multi-head architecture, certain heads may employ standard self-attention while others utilize HyperFeatureAttention (of varying orders) or HyperAttention to capture richer, higher-order interactions. Depending on the computational constraints, the HyperAttentions may leverage the linear approximations described in Appendix G.3.

## I. Figures

---

[13]Refer to Theorems 3.1, 4.4, 4.6, and 4.8
[14]Refer to Section 5 and Appendix F
[15]Refer to Section 6 and Appendix G

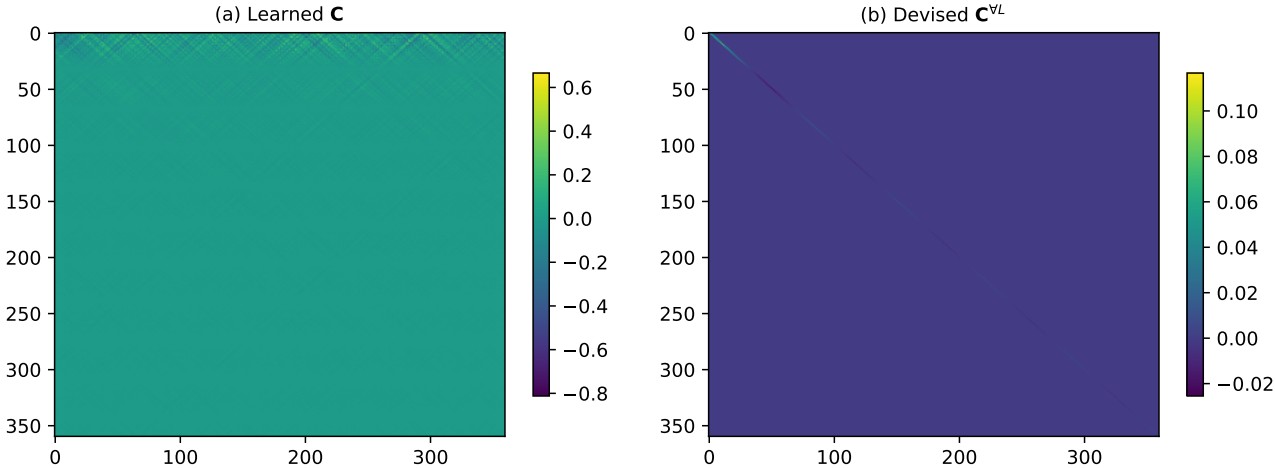

*Figure 2.* Comparison of learned vs. devised parameters for sinusoidal embedding: (a) Devised matrix $\mathbf{C}^{\forall L}$ showing the original devised structure, (b) Learned matrix $\mathbf{C}$ demonstrating the emergent but non-interpretable patterns. While visually distinct, both parameterizations lead to equivalent model behavior through different mathematical organizations.

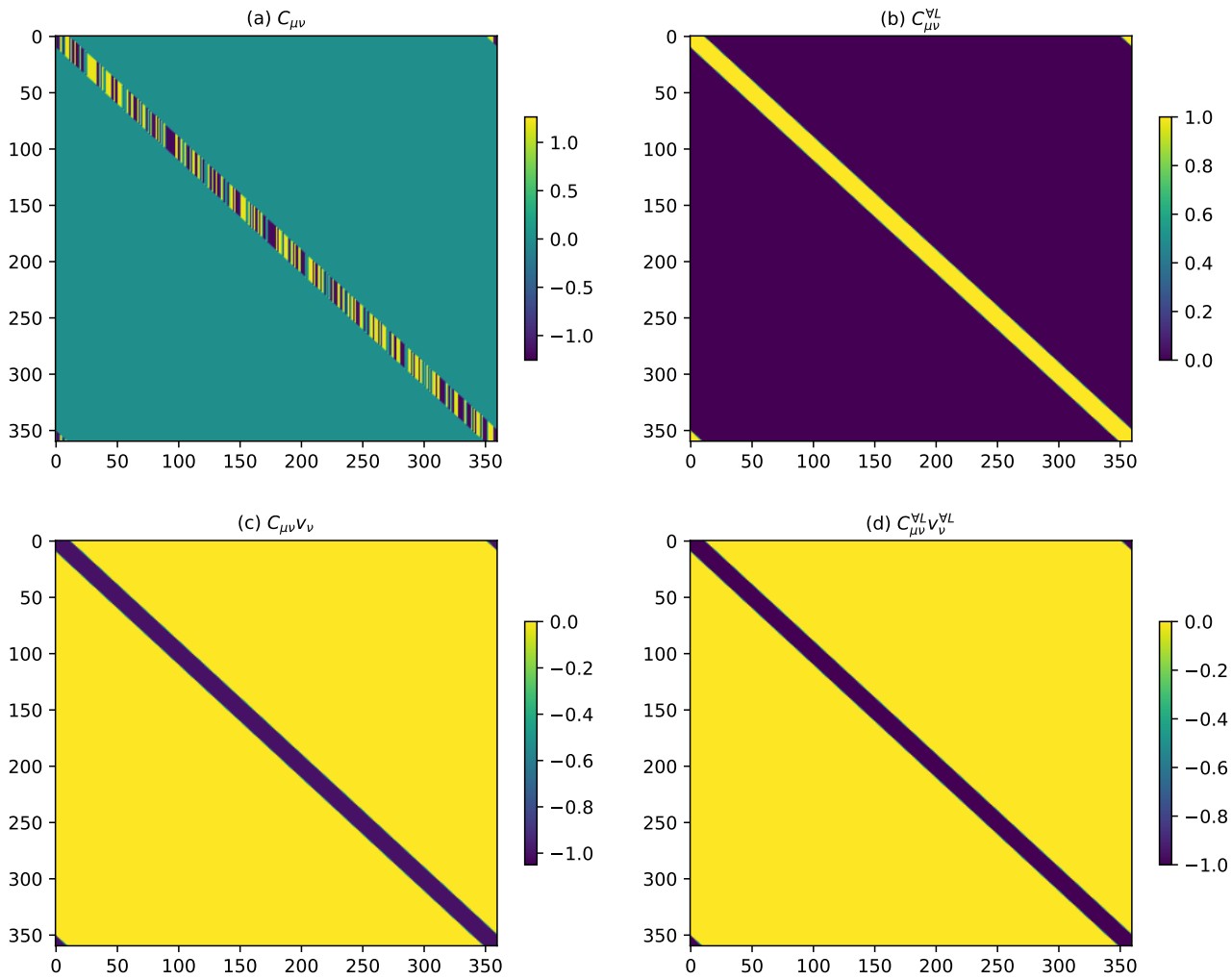

*Figure 3.* Equivalence of learned parameters with one-hot embedding in domain embedding base (here the parameters are already in the domain embedding base so we did not transfer them): (a) $C_{\mu\nu}$ learned parameters in domain embedding base, (b) $C^{\forall L}$ devised parameters in domain embedding base, (c) $C_{\mu\nu}W_{\nu 0}$ the interesting matrix in domain embedding base, (d) Transformed $C^{\forall L}_{\mu\nu}W^{\forall L}_{\nu 0}$ using original parameters. The mean squared difference between (c) and (d) is $\mathcal{O}(10^{-5})$, demonstrating functional equivalence despite different parameter organizations.

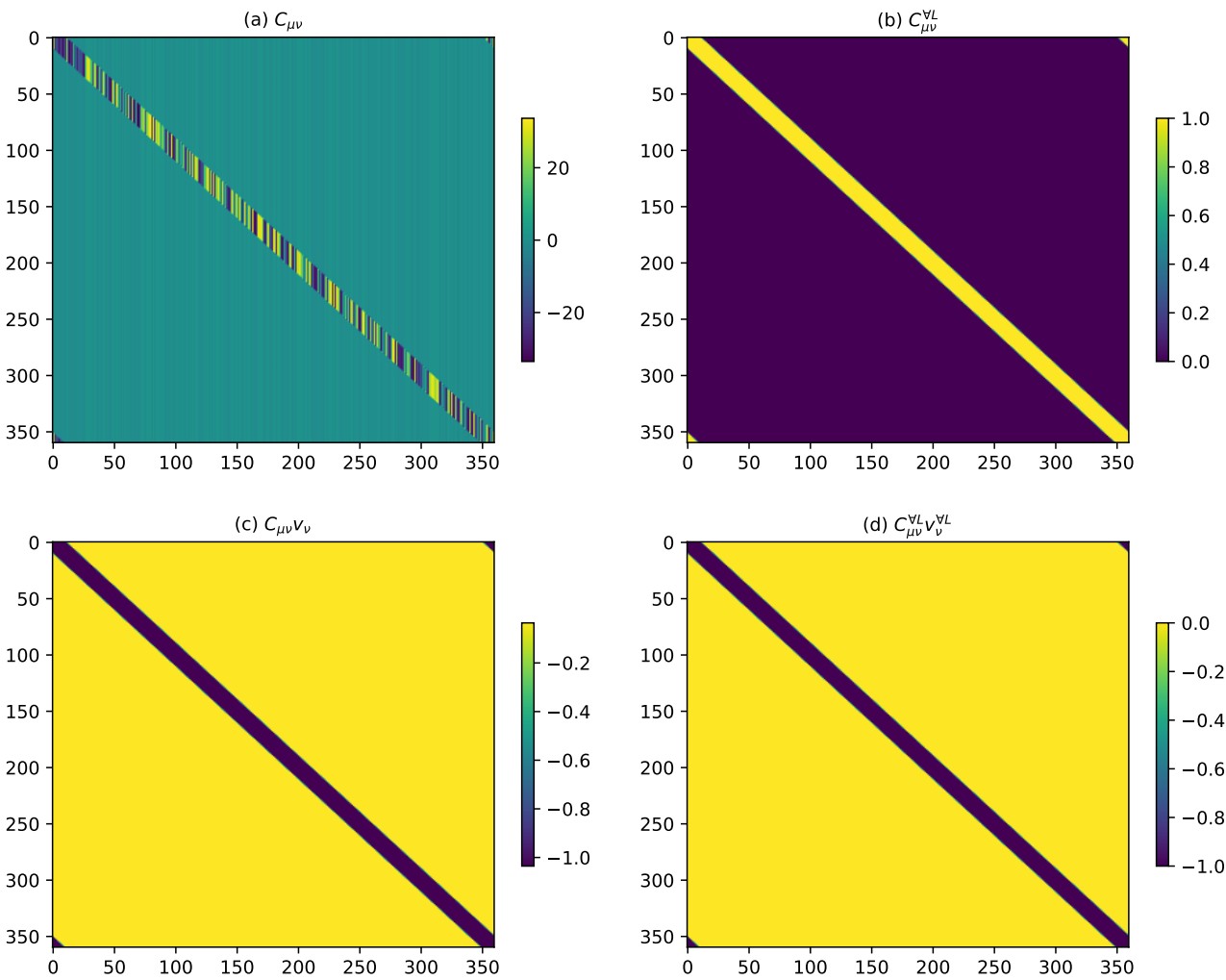

*Figure 4.* Equivalence of learned parameters with sinusoidal embedding in domain embedding base: (a) $C_{\mu\nu}$ learned parameters in domain embedding base, (b) $C^{\forall L}$ devised parameters in domain embedding base, (c) $C_{\mu\nu}W_{\nu 0}$ the interesting matrix in domain embedding base, (d) Transformed $C^{\forall L}_{\mu\nu}W^{\forall L}_{\nu 0}$ using original parameters. The main squared difference between (c) and (d) is $\mathcal{O}(10^{-5})$, demonstrating functional equivalence despite different parameter organizations. Additionally as a side note, comparing (b) with Fig. 2 (b), we observe the advantage of sinusoidal embeddings in terms of their parameter efficiency within the $\mathbf{C} = \mathbf{W}^Q \mathbf{W}^{K\top}$ matrix, particularly when relative positions are more important than absolute positions.

