# OpenReview forum: "A Theoretical Study of (Hyper) Self-Attention through the Lens of Interactions: Representation, Training, Generalization"
_ICML.cc/2025/Conference — ICML 2025 poster_

### Official Review · Reviewer_Ct39 · 2025-02-25

**Overall Recommendation:** 3

**Summary:**

The paper presents a theoretical study of self-attention mechanisms, specifically focusing on their representation, training, and generalization capabilities through the lens of mutual interaction among entities. The authors introduce a novel perspective called "interacting entities," demonstrating that a single layer of linear self-attention can effectively represent, learn, and generalize functions that capture pairwise interactions, even in out-of-distribution scenarios. They also propose two extensions: HyperFeatureAttention, which captures interactions between features, and HyperAttention, which models higher-order interactions among multiple entities.

**Claims And Evidence:**

The claims made in the paper are supported by a combination of theoretical proofs and empirical experiments. The authors provide evidence showing that self-attention can generalize well across different distributions and sequence lengths.

**Essential References Not Discussed:**

I think there are many attention variants in recent years, but this paper does not mention much.

**Experimental Designs Or Analyses:**

The experimental designs are sound and validate the theoretical findings.

**Methods And Evaluation Criteria:**

The proposed methods, including HyperFeatureAttention and HyperAttention, are well-grounded in theoretical constructs and are evaluated against established benchmarks. The evaluation criteria make sense for the tasks at hand, focusing on representation and generalization across different scenarios.

**Other Comments Or Suggestions:**

NA.

**Other Strengths And Weaknesses:**

The paper exhibits strengths in originality and clarity, providing a fresh perspective on self-attention mechanisms while effectively communicating complex ideas. However, some sections could be more concise, and additional examples could enhance the understanding of the proposed extensions.

**Questions For Authors:**

1.	How could the proposed HyperFeatureAttention and HyperAttention be used in MARL settings?
2.	Are there any potential drawbacks or challenges when applying your methods to larger datasets or more complex tasks?

**Relation To Broader Scientific Literature:**

The key contributions of the paper build on existing theories of self-attention and representation learning. The authors relate their findings to prior works on Transformers and attention mechanisms, positioning their contributions within the broader context of machine learning literature.

**Theoretical Claims:**

I did not check the proofs.

---

> ### Author Rebuttal · Authors · 2025-04-01
>
> *RESPONSE 1* (About the references):
> In the introduction we referenced several works on "theoretically understanding attention" but we agree that attention modifications could be discussed more. In the final version of the paper we will mention these approaches, i.e, the papers such as (DEBERTA) https://arxiv.org/pdf/2006.03654; (Improving sampling by modifying the effective diffusion) https://arxiv.org/abs/2410.05258;  (Selective Attention) https://arxiv.org/pdf/2410.02703.
>
> In addition, we will expand the discussion with the "mechanistic interpretability" literature such as (Tranformer Circuits) https://transformer-circuits.pub and previous studies attempted to understand and modify attention and modify attention.
>
> Also, We will briefly mention NA2Q (ICML, 2023) and other relevant methods in the revised version of our paper.
>
> *RESPONSE 2* (Use of HyperFeatureAttention and HyperAttention in MARL Settings):Our work views tokens as interacting entities—a perspective that naturally encompasses agents in multi-agent reinforcement learning (MARL). In MARL, interactions are often more complex than simple pairwise relationships, and our proposed extensions are well suited to address such challenges. In particular, HyperAttention is designed to capture higher-order (e.g., three-way or n-way) interactions, which are common in collaborative or competitive multi-agent environments where an agent’s outcome may depend on the joint behavior of multiple agents. Similarly, HyperFeatureAttention enables the coupling of different feature-level interactions (such as spatial positions, velocities, or task-specific attributes), providing a more expressive representation of agent dynamics. Although our current experiments primarily focus on controlled settings, the theoretical framework and empirical evidence presented in Sections 5 and 6 of our paper indicate that these modules can be naturally extended to MARL scenarios. In future work, we plan to investigate and validate these extensions in MARL benchmarks, which we believe will further demonstrate the versatility and potential impact of our approach in such applications.
>
> *RESPONSE 3* (Potential Drawbacks or Challenges for Larger Datasets and More Complex Tasks): We strongly suggest you to refer RESPONSE 6 of our reply to reviewer xG3k where we summarize the potential drawbacks in terms of complexity of models and how to overcome them.

---

### Official Review · Reviewer_DnSa · 2025-03-07

**Overall Recommendation:** 3

**Summary:**

This paper proposes a theoretical framework viewing self-attention tokens as interacting entities. The key theoretical findings include proving the representational power of single-layer linear self-attention for pairwise interactions, demonstrating convergence of training under mild assumptions, and establishing generalization guarantees to both within-distribution and out-of-distribution settings. Additionally,  the authors introduce two innovative extensions: HyperFeatureAttention, designed to capture couplings between different feature-level interactions, and HyperAttention, which models higher-order (multi-entity) dependencies.

**Claims And Evidence:**

The authors support their claims with rigorous theoretical proofs and empirical validation using a simplified “colliding agents” scenario. However, as the experimental results are confined to synthetic environments, further validation in more complex, real-world scenarios is necessary to fully support the claims.

**Essential References Not Discussed:**

The paper cites many relevant studies, but might lack a detailed discussion of recent advances in Transformer interpretability. A more detailed comparison with current studies on Transformer interpretability would better highlight the contributions and novelties of this paper.

**Experimental Designs Or Analyses:**

The experimental design is sound but might not be adequate. Additional experiments in more complex or empirical scenarios are lacking, which could further enhance confidence in the generalizability of the findings.

**Methods And Evaluation Criteria:**

The theoretical methods employed are appropriate and contribute notably towards improving interpretability and potential performance of self-attention mechanisms. However, the evaluation primarily uses synthetic environments, and extending experiments to more complex real-world scenarios or well-established benchmark datasets could strengthen empirical claims.

**Other Comments Or Suggestions:**

- Lack of clear symbol explanations: For example, Eq. (3) and $f_a^h$ (Line 279 right column).

- Formatting issues: Line 228 contains "let Let"; Line 232-233 uses wrong punctuation.

Overall, while this is an interesting and meaningful study, it feels more like an intermediate-stage work. The authors provide rigorous mathematical foundations, but the lack of empirical validation beyond toy examples limits its impact. Improved readability, more detailed and intuitive theoretical explanations, and broader experimental validation would elevate the paper’s contribution.

**Other Strengths And Weaknesses:**

**Strengths**:
- The authors introduce a novel perspective for understanding the self-attention mechanism.
- The generalization of attention mechanisms to novel HyperAttention modules is innovative and well-motivated.
- The authors provide clear examples to demonstrate the ability of self-attention to capture mutual interactions.

**Weaknesses**:
- Complexity: While HyperAttention may capture higher-order interactions, this comes at the cost of increased computational complexity. The necessity of such complexity is questionable, especially given some research suggesting that higher-order interactions can be approximated using weighted pairwise interactions.
- Limited empirical validation: The paper primarily focuses on theoretical contributions, with experimental results limited to controlled synthetic environments. The authors do not provide empirical evidence demonstrating HyperAttention’s superiority in practical tasks. More diverse empirical demonstrations would enhance the credibility of the findings.
- Presentation: The presentation of theorems and formulations could be enhanced with clearer definitions and intuitive explanations.

**Questions For Authors:**

- Q1: How does the complexity of the HyperAttention scale when capturing very high-order interactions (e.g., four-way or higher)? Would this limit practical applicability? Can HyperFeatureAttention and HyperAttention mechanisms practically scale to large datasets?

- Q2: How sensitive are your theoretical guarantees to changes in embedding dimension (specifically when d < N)? Could empirical experiments be performed to quantify this sensitivity?

**Relation To Broader Scientific Literature:**

This paper addresses gaps such as the interpretability of the attention mechanism and out-of-distribution generalization.

**Theoretical Claims:**

I reviewed the proofs in the Appendices, which are the key contributions of this theoretical paper.  The proofs  are basic sound. However, clarity regarding symbolic definitions and intuitive explanations of mathematical formulations could be further improved.

---

> ### Author Rebuttal · Authors · 2025-04-01
>
> LINK (please open in a private window): https://drive.google.com/file/d/1lJ3HYR6i02jpm3CAxg4cu4J6icQXubJ8/view?usp=share_link
>
> *RESPONSE 1* (On the experiments for "further validation in more complex, real-world scenarios", "more complex or empirical
> scenarios", "well-established benchmark datasets"): For self-attention, our main goal is to provide rigorous theories for the understanding of the existing self attention mechanism. Consequently, our current experiments primarily validate the theoretical framework of linear self-attention, offering strong empirical support as opposed to conducting large-scale experiments on self-attention (which has already done on many literature as we cited). That being said, we have done large-scale experiments such as famous many body problem where we predict trajectories of bodies (please refer to LINK)
>
> *RESPONSE 2* ("clarity regarding symbolic definitions and intuitive explanations of mathematical formulations could be further improved" and "presentation of theorems and formulations ... with clearer definitions and intuitive explanations"): We made the symbolic definitions more distinguishable (for instance we will replace $s^{(n)}(i)$ "index to corresponding element mapping" function with $\mathcal{X}^{(n)}(i)$ since the former version may be confused with $\mathbf{s}^{(n)}$ "count vector") and we further improve the mathematical formulations. Further, in the final version of the paper we will provide clearer definitions and intuitive explanations
>
>
> *RESPONSE 3* (on the references on "Transformer interpretability"): Please refer to RESPONSE 1 of our reply to reviewer Ct39.
>
> *RESPONSE 4* (on the concerns and questions on complexity and practicality of Hyper(Feature)Attention): please refer to the "RESPONSE 6 (Comparison Between Attention Models)" part of our part of our reply to Reviewer xG3k.
>
> *RESPONSE 5* (Regarding empirical evidence for Hyper(Feature) Attention): Please refer to *RESPONSE 3* in our reply to reviewer N655.
>
> *RESPONSE 6* (to Question 2): When we only remove the $d<N$ assumption, we start to get nonzero errors i.e. approximations to the exact solutions to the task (which is the practical real world situation). This is explained in both at the end of Section 3 "*Why d=N?*" part and in Theorem B.4. Shortly, "... Our goal is to understand how self attention captures mutual interactions, and d = N ensures orthogonal domain embeddings, yielding an exact and transparent representation. Compressing to $d < N$ is perpendicular to this focus and can be addressed separately using standard techniques... Starting with $d = N$ allows us to establish theorems that elucidate how self-attention captures pairwise interactions, while reducing to $d < N$ merely introduces a small approximation gap without altering the core theory. For completeness, we provide.. (Theorem~B.4)". That is being said, we runned experiments to quantify this sensitivity on the same colliding agents environment with $d<N$. As a result, we had nonzero errors (train, test, out of distribution). As the embedding dimension approached to the dictionary size the errors approached to zero. Although it is an important insight, that discussion is orthogonal to the theories for this paper,so we left those results for a future work. However, we may add them to the final version of the paper.
>
> *RESPONSE 7*: Lastly, we are thankful for pointing out the typos, formatting issues and symbol explanation issues. Considering the length of the paper, unfortunately we made those small errors which are easy to fix and will be totally fixed in the final version of the paper.

---

### Official Review · Reviewer_Qpd5 · 2025-03-11

**Overall Recommendation:** 3

**Summary:**

The paper studies the self-attention mechanism that is central to many of today's ML models (such as in NLP, computer vision and multi-agent systems). Instead of Transformers (which involve several layers of multi-head attention and other components), the paper examines a simplified model that consists of a unique linear self-attention head (linear = no softmax). Most application examples are toy multi-agent systems. The paper describes some mathematical properties of this simplified self-attention model and verifies them empirically; additionally it proposes two generalisations of self-attention, called HyperFeatureAttention and HyperAttention.

The mathematical properties are that, given some assumptions (claimed to be mild in practice):
* Self-attention can represent any pairwise interaction of tokens (theorem 3.1),
* and does so more efficiently than any fully-connected neural network (theorem 3.2);
* gradient descent converges to zero training error (theorem 4.4),
* which also means perfect fit on a test set from the same distribution (theorem 4.8),
* and perfect fit on sequences that are longer or shorter than in the training set (theorem 4.10).

The two proposed extensions of self-attention are supposed to generalise the pairwise interaction property:
* HyperFeatureAttention can represent interactions between several features of the same two tokens. The authors claim that even two layers of multi-head self-attention cannot do this.
* HyperAttention can represent interactions between more than two tokens at a time.

## update after rebuttal

**Claims And Evidence:**

The theoretical claims are well supported, both by mathematical proofs and by experiments. An exception is the claim that everything generalises to $d<N$ (see below about theorem B.4).

The proposed extensions of self-attention are justified theoretically, but not put into practice. However, it is conceivable that more traditional self-attention architectures can also represent these complex interactions, perhaps in surprising ways. (After all, LLMs use standard self-attention and are very good at modeling language!) An insightful further experiment would be to train and compare toy models with and without Hyper(Feature)Attention.

**Essential References Not Discussed:**

The paper cites Elhage et al. 2021, which shows that the authors are at least partly aware of the "mechanistic interpretability" literature. While it is only tangentially related to this paper, it could be valuable to consult and cite it more.

In particular, the claim that earlier works
"often do not delve into interpreting the roles of the learned parameters beyond attention
patterns"
(l. 73-75 col. 2)
seems a bold statement given papers like https://arxiv.org/abs/2412.11965 or even the Elhage paper itself. (These papers describe the idea of expanding the parameters of attention heads in language models to represent token interactions.)
Another point about the same sentence: to my knowledge attention patterns are not the state of the art in interpreting the behaviour of attention heads  (https://icml.cc/virtual/2024/poster/32735)

More generally, surveys on mechanistic interpretability of language models could be valuable resources: https://aclanthology.org/2020.tacl-1.54, https://arxiv.org/pdf/2405.00208, https://arxiv.org/pdf/2501.16496.

The submission attempts not only to theoretically understand attention, but also to generalise or modify the mechanism.
Previous studies have also attempted this, but none of them is cited here.
Their motivations are various and not directly related to the present ones, but I think it would still make sense to briefly mention them.
I am aware of the following (there are probably many more):
https://arxiv.org/pdf/2006.03654, https://arxiv.org/abs/2410.05258, https://arxiv.org/pdf/2410.02703.

**Ethical Review Concerns:**

No concerns.

**Experimental Designs Or Analyses:**

Looks good to me, but I did not thoroughly check them.

**Methods And Evaluation Criteria:**

The mathematical claims are proven and tested empirically, which is good. For the proposed extensions, see the previous point.

**Other Comments Or Suggestions:**

Typos, grammar and clarity:
* Footnote 1: what is k and L?
* l. 176: "self-self attention"
* l. 228: "let Let,"
* l. 241: "Let ... as" -> "Let ... be"
* l. 242: comma -> full stop
* l. 268: "converges" -> "converges to"
* section 4.1: please state the assumptions before the theorem
* l. 248 (col. 2): "than" -> "then"
* l. 322: "showed" -> "show"
* eq. (3): what is A? I could only guess it when I read appendix E.
* l. 309 (col. 2): "Remark completeness" -> "remark for completeness"
* l. 370 (col. 2): "a circle of radius R executes" -> "a circle of radius R that executes"
* l. 1623: I guess the second H should be an A.

**Other Strengths And Weaknesses:**

The theoretical results are really nice!

However, I doubt that the assumptions are as mild as the authors claim.
I am particularly skeptical of assumption 4.2:
In the case of language, some tokens are very rare, and (depending on their joint distribution) it is quite possible that two of these rare tokens never co-occur. As I understand the definitions, this would lead to a column (?) of zeros, contradicting full rank.

The proposed generalisations of self-attention are also valuable ideas.

But, in my opinion, this does not mean that standard self-attention is as weak as the authors claim.
For example, in my understanding, the "skip-trigram "bugs"" (Elhage et al. 2021, quotation marks original) are essentially a theoretical construction that does not seem to occur in real language models. Real language models clearly "know" that "keep ... in mind" and "keep ... at bay" are correct but "keep ... in bay" is not.
I assume this is because the attention output of "keep" is not the whole output: the attention head usually also attends to other tokens to some extent, such as the current token "in"; and there is a skip connection that enables direct usage of the embedding of "in" (which may encode bigram statistics, compare Elhage et al.'s section on zero-layer transformers).

This questions to what extent HyperAttention is really more useful than more standard mechanisms.
But again, whatever may be the case, it is definitely a valuable idea!

Finally, I feel that the connection between the theoretical results and the proposed architectural modifications is a bit loose. Both parts would be valuable on their own. If they are to appear in the same paper, the authors should better emphasise how they are connected.

**Questions For Authors:**

You have given an example of an interaction that standard two-layer multi-head attention apparently cannot represent, but HyperFeatureAttention can. Conversely, are there interactions that HyperFeatureAttention cannot represent but standard (two-layer multi-head) attention can? This is quite important if we want to know how useful HyperFeatureAttention really is. And whatever the answer, you should talk about it in the paper.

**Relation To Broader Scientific Literature:**

The most relevant literature tries to gain a deeper mathematical understanding of Transformers. For example, the paper cites a work that shows Transformers to be Turing-complete. As a part of understanding Transformers, many works also try to understand their training process, i.e., how the training loss converges. Both threads are continued in the theoretical contributions of this paper.

**Theoretical Claims:**

I checked the following proofs:

* proof of 3.1: no issue
* proof of 3.2: correct
* proof of B.4: The "for every $\varepsilon$" claim is not proven and in fact seems to be wrong. In the proof the $\varepsilon$ is introduced by $\|B_* -F\| \leq \varepsilon$. But given $F$ and $d$, $B_*$ minimises $\|B_*-F\|$ over all rank-$d$ matrices, so $\|B_*-F\|$ is fixed! So the statement $\|B_* -F\| \leq \varepsilon$ breaks down for any $\varepsilon$ smaller than the fixed $\|B_* -F\|$.

    The theorem in its present formulation should therefore be removed from the paper. To save the claim that the $d<N$ case is reasonably similar to $d=N$, the authors should reflect carefully on what the Eckart-Young theorem and/or the Johnson-Lindenstrauss lemma actually imply for the question at hand. It may be possible to replace an "always" statement with a "with high probability" statement.

* proof of 4.4: in the equations on line 1278 I think $1/B$ should be $2/B$, but this does not affect the validity of the proof (as ultimately it is about when these things are zero).

    I did not check the rest of the proof in detail.

---

> ### Author Rebuttal · Authors · 2025-04-01
>
> *RESPONSE 1* (the concerns on practicality of proposed models and what is their use while self-attention works in NLP): We agree that traditional self-attention architectures can also represent these complex interactions "in surprising ways". However, the main distinction is that the proposed models can represent those with fewer parameters. For instance, it is discussed  in Appendix E.1 and Appendix E.2's "a short note on approximate embedding" paragraph. We agree that we need to carry some of those discussions to the main text and make it clear in the main text, which will be done in the revised version. Another point is we do not advocate replacing conventional self-attention with the proposed models. For more details we strongly suggest referring to "RESPONSE 6" part of our reply to reviewer xG3k.
>
> *RESPONSE 2* (insightful experiments): As for the small-scale experiments, we have promising preliminary results, please refer to "RESPONSE 3" part of our reply to reviewer N655.
>
> *RESPONSE 3* (the concerns on proof of Thm B.4): We agree with your concerns and we think that old version was not clear. Therefore, we completely changed the theorem statement and the proof, keeping the essence the same as old version. In the new version, we use low rank approximations to the matrices that would solve the task with zero error if embedding dimension were large enough. We bound the infinity norm between the output of the linear self attention having $d<N$ and the correct labels. The bound is in terms of singular values of the matrices that would solve the task with zero error if embedding dimension were large enough. Unfortunately, it is not allowed to provide whole theorem and proof with anonymous links.
>
> *RESPONSE 4* ("earlier works often do not delve into interpreting the roles of the learned parameters beyond attention patterns") While making that statement we were mostly thinking about the papers not only theoretical but also \emph{mathematical}, with rigorous theories expectations and validations of those experiments. We apologize for the confusion and in final version we will clarify. In more detail, our approach is different from conventional mechanistic interpretability approach that our approach is rigorous/mathematical and our experiments are for validating our predictions. Their approach  is like "phenomenology", and our approach is "fundamental".
>
> *RESPONSE 5* (About the scepticism on assumption 4.2): In the revised version of the paper we will add a section to the appendix that justifies Assumption 4.2. In short, assuming $\mathbf{s}^{(n)} \in \mathbb{R}^{|\mathcal{S}|}$ are sampled from a distrubution such that covariance of $\mathbf{s}^{(n)}$ is positive definite, we will use matrix Bernstein inequality and Weyl's inequality and prove that the statement in the Assumption 4.2 hold with high probability: for some $\gamma >0$, B is the number of data samples, $\mathbb{P}\left(\mathrm{rank}(\mathbf{S}_{{B}\mu}) \le |\mathcal{S}|\right)\le e^{-\gamma B}$
>
> *RESPONSE 6* (real LLM's not falling to skip trigram bug): We agree that LLMs as a whole do not have such bugs. As you mentioned, the skip connections from earlier layers help to approximately get rid of those problems and by adding more layers, the approximations gets better. In our analysis, we focus on self-attention itself (not a large neural network) and compared it with our new modules (small modules not large networks). We strongly suggest looking at the RESPONSE 6 part of our reply to reviewer xG3k (especially the last paragraph of it), which we will add to the paper.
>
> *RESPONSE 7* (On the question about the usefulness of HyperFeatureAttention (HFA)): Both HFA and self-attention have same complexity, so we compared them by parameter count,since even MLPs can represent any function with enough parameters. Our intentially unfair comparison used two-layer multihead self-attention against a single-layer single-head HFA. A fairer comparison (single-layer single-head vs. single-layer single-head) shows HFA captures self-attention. We acknowledge that some interactions require two-layer multihead self-attention, but for more on HFA’s utility, please see “RESPONSE 3” in our reply to reviewer N655.
>
> *RESPONSE 8*(connection between theoretical results and the proposed modifications): Various colliding agent example scenarios in Appendix B.1.3 and "Non-Identical Agents Revisited" in Appendix E.1, we introduce how HFA is a natural construction from the theoretical insights on self-attention. Though seemingly unrelated, they were stepping stones. Thank you; we’ll clarify.
>
> *RESPONSE 9*: For essential references please refer to "RESPONSE 1" to reviewer Ct39.

---

### Official Review · Reviewer_xG3k · 2025-03-13

**Overall Recommendation:** 4

**Summary:**

This paper is separated into two broad sections. The first is a theoretical study of linear self-attention, which explores the expressivity and generalisability of a single linear self-attention layer. The second is more empirical, proposing two new architectures based on self-attention. The performance of these models are demonstrated experimentally.

**Claims And Evidence:**

Following on with the two broad sections of the paper:

The theoretical portion of the work appears very rigorous, with claims well justified and all assumptiosn laid out clearly and justified. While I was unable to robustly verify every proof, due to expertise in this area, I believe what I have checked holds up.

I think the experimental section leaves slightly more to be desired. The theory is tested for one indicative problem setting, with two different types of embedding. I may be missing something, but while there is discussion of performance for the 'One-Hot Embedding' problem, I am unable to find these results in the paper. Figure 1 applies to Sinusoidal Embeddings, but showing the theoretical results to allow us to compare would be useful. I think demonstrating these findings on more than one problem setting would also be good, although I appreciate that I lean more on the empirical side and this is a strongly theoretical paper.

One claim I did not see justified was in footnote 5: that the approach in this paper has greater representational capacity *while maintaining comparable efficiency* - it is later mentioned that HyperAttention in this paper is $\mathcal{O}(L^3)$, and so it would be good to demonstrate that this does not lead to drastic computational increases compared to the low-rank approximation of prior work.

On a slightly separate note, a lot of the appendix is filled with figures which are neither discussed nor referenced in the main text. If these plots provide evidence or information, it would be good to at lesat have an in-text reference for them in the main body of the paper.

**Essential References Not Discussed:**

N/A

**Experimental Designs Or Analyses:**

I think additional experimental settings could be good as a means to improve the paper, but as stated I do not think there is a desparate need for that as a principally theoretically-minded paper. The experiment that they do include would benefit from some additional discussion in the main body of the paper, rather than purely relegating that to the appendix, and I am unable to find the results for one-hot embeddings.

**Methods And Evaluation Criteria:**

I have sort of discussed this above. I think additional experimentation, using more than one problem setting would be good, but I think it is worth bearing in mind that the key contribution of this paper reads more theoretical than empirical and thus the bar for experimental results should, rightly, be lower.

The motivation of the theory is good, and all simplifying decisions and assumptions are well justified.

I personally found the introduction of the methods somewhat hard to follow - for a less theoretically-minded person like myself, a short statement explaining, in words, what the HFA and HA layers actually do and how they differ from SA would be helpful.

**Other Comments Or Suggestions:**

See above.

**Other Strengths And Weaknesses:**

I found most of the paper well written and clear, and feel it is clearly of a high quality. That said, I found a few points that I wanted to raise.

- I found the examples used (particularly the contextualising example about alleles) a little bit counterintuitive/out of nowhere. Finding a way to ground an example in language would be good, given it is the field which has arguably seen the largest breakthroughs as a result of self-attention.
- It would be nice in the main paper to highlight with a bit more discussion the effect of not having $d=N$, as this introduces some level of compression and thus may violate some of the approximations despite being a more practical system.
- I found line 248 ('In short, ...') confusing English.
- I found the HyperFeatureAttention notation a bit confusing - what are the curly brackets for, are they just used as normal parentheses?
- The frequent tense changes in the paragraph starting 'Setup: Colliding Agents on a Cylindrical Grid.' made the text quite hard to follow.

**Questions For Authors:**

- Can you please explain how you can verify that HyperAttention is comparably efficient to prior work in this area, which uses low rank approximations?
- Given this theory, what are the violations that make the expected guarantees not generally true in the real world - as in, why do many systems in practice **not** have zero error and perfect generalisation to new data and completely different lengths?
- Can you please explain what the use of HyperFeatureAttention and HyperAttention are over self attention, given the guarantees already present for self-attention?

**Relation To Broader Scientific Literature:**

The paper seems well researched and placed in scientify literature, although I am not well placed to contextualise this work.

**Theoretical Claims:**

I attempted to verify a number of the proofs, though was unable to check every single one due to the sheer length of the paper and time constraints (@AC, see comment). What I checked seemed to be correct.

---

> ### Author Rebuttal · Authors · 2025-04-01
>
> LINK (OPEN IN PRIVATE WINDOW): https://drive.google.com/file/d/1lJ3HYR6i02jpm3CAxg4cu4J6icQXubJ8/view?usp=share_link
>
> *RESPONSE 1* (the results of one-hot embedding): The results were explained as "negligible error $\Theta(10^{-7})$ on test and out of distribution data". However, we agree and for complenetess we will add the learned parameters to the paper (see LINK).
>
> *RESPONSE 2* (additional experiments for theory validation): We experimented the theories on the genotype phenotype mapping task (explained Sec. 3 Example 2 and Appendix B.2) and got same results $\Theta(10^{-7})$ error on test and OOD data.We may add them to paper.
>
> *RESPONSE 3* (the figures in the appendix "which are neither discussed nor referenced in the main text"): discussed in Sec.7 as "shown in Figures 3 and 4, these matrices are indistinguishable from the theoretical counterparts"
>
> *RESPONSE 5*: Sanford et al. shares our $\mathcal{O}(L^3)$ complexity. The footnote 5 is unnecessary and will be removed.
>
> *RESPONSE 6* (Comparison Between Attention Models): We will add the following discussion and the attached table (LINK) to the paper. Letting embedding dim to be $d$, sequence length to be $L$, we compare those models in terms of number of parameters, computational complexity, and abilities.
>
> - For self-attention, it captures mutual interactions between entities (cf.Thms 3.1,4.4,4.6,4.8).If it has multiple heads, it can capture summation over mutual interactions between features of the entities. It has ${\Theta}(d^2)$ parameters, and its computational complexity is ${\Theta}(L^2)$. In order to reduce the computational complexity to ${\Theta}(L)$, people came up with approximations called "Linear Attention"
> (Katharopolus et al 2020, Wang et al 2020). However, despite the name "linear", the method is used to approximate nonlinear selfattention.
>
> - HyperFeatureAttention captures couplings of mutual interactions between features (cf. Sec.5, AppendixE). If it has multiple heads, it can capture summation over couplings of mutual interactions between features. Same as self-attention, HyperFeatureAttention has ${\Theta}(d^2)$ parameters, and its computational complexity is ${\Theta}(L^2)$. The same Linear Attention approximations can be applied to HyperFeatureAttention, reducing its computational complexity to ${\Theta}(L)$. Seeing that the main goal of the paper is not this approximation for HyperFeatureAttention, we did not show it explicitly.
>
> - As for HyperAttention of order $n$, it captures up to $n^{th}$ order interactions (Refer to Sec.~6 and Appendix~F). If it has multiple heads, it can capture summation over up to $n^{th}$ order interactions between features of the tokens. It has ${\Theta}(d^2)$ parameters, and its computational complexity is ${\Theta}(L^n)$. Using similar Linear Attention approximations, in Appendix~F.3, we explained how to reduce computational complexity of HyperAttention to ${\Theta}(L)$. We will add the requirements on the query and key matrices that if their entries are less than $o(\sqrt[3]{\log L})$ then the infinite norm between the actual outputs and approximate outputs is less than $1/\mathrm{poly}(L)$.
>
> - While we do not advocate replacing conventional self-attention with HyperAttention or HyperFeatureAttention, we propose these mechanisms as complementary enhancements. In a typical multi-head architecture, certain heads may employ standard self-attention while others utilize HyperFeatureAttention (of varying orders) or HyperAttention to capture richer, higher-order interactions. Depending on the computational constraints, the HyperAttentions may leverage the linear approximations described in Appendix~F.3.
>
> *RESPONSE 7* ("the violations that make expected guarantees not generally true in the real world" and  "why dont we have zero test errors and what happens when $d<N$"): When we only have $d<N$ assumption, we start to get nonzero errors i.e. approximations to the exact solutions to the task (which is the practical real world). This is explained in both at the end of Sec. 3 "*Why d=N?*" part and in Thm B.4. Another key assumption in our analysis is the (weak, strong, universal) realizability. Although these conditions may not hold in practical scenarios, implying that a single layer, single-head self-attention mechanism may not fully capture the complexities of real-world tasks, it is important to note that SOTA models typically employ multiple layers of self-attention blocks. Such architectures can approximate the desired functions even when the strict realizability assumptions are violated.Actually, the realizability and d=N assumptions are very similar in essence because when letting d large enough, we simply showed in Appendix B that realizability is satisfied (with several example scenarios such as colliding agents, genotype phenotype mapping, vision task, time series prediction). A thorough investigation of these approximations, lies beyond the scope of this work.

---

> > ### Comment · Reviewer_xG3k · 2025-04-02
> >
> > Dear Authors,
> >
> > Thank you for your response, as well as taking the time to generate additional results.
> >
> > Re: One hot, I might be missing something but I am not sure in the PDF where results showing the accuracy (i.e. $10^{-7}$) are. Let me know if there's something I've not seen.
> >
> > Re: additional experiments, I think including these results would be beneficial. Crucially, though, I think these results need to be *shown* rahter than stated without evidence.
> >
> > Re: Figure references, I apologise - this was an oversight on my behalf.
> >
> > Re: complexity, that is good to know, thank you for clearing it up.
> >
> > Re: Comparison, I think that a breakdown of this type would be really helpful for the flow of the paper, probably attached to a paragraph explaining what the comparison means.
> >
> > Re: violations, I think it is fair to recognise that as a theoretical piece of work this will rely on certain assumptions which may not always hold for real-world problems, but that this is OK. I agree with your comments that investigating these approximations goes beyond the scope of this work, which is already quite dense as-is.
> >
> > I do not plan on increasing or decreasing my score - I have recommended acceptance because I think this is a well written paper and answers some interesting questions. That said, it is also quite beyond my domain of expertise and so I am unable to make any strong recommendation due to a lack of confidence in my review. All of that said, well done and good job!

---

> > > ### Author Response · Authors · 2025-04-06
> > >
> > > Thank you for the rebuttal comment. You can find the result for one-hot at line 408 left column.

---

### Official Review · Reviewer_N655 · 2025-03-21

**Overall Recommendation:** 3

**Summary:**

The paper introduces a broad theoretical perspective to analyze the capabilities of self-attention mechanisms, particularly focusing on the interactions between entities. The paper extends the traditional pairwise self-attention to higher-order interactions and presents two novel mechanisms: HyperFeatureAttention and HyperAttention. These mechanisms capture feature-level and multi-entity dependencies, respectively. The authors provide a comprehensive theoretical analysis demonstrating how self-attention learns and generalizes mutual interactions, with guarantees for out-of-distribution (OOD) generalization. The paper also includes experiments validating these theoretical claims, showing that the learned parameters align with the theoretical predictions.

**Claims And Evidence:**

This paper has given a clear representation and proof.

**Essential References Not Discussed:**

Regarding high-order interaction among multi-agent reinforcement learning, an important work should be considered in the introduction or related works section: NA2Q: Neural Attention Additive Model for Interpretable Multi-Agent Q-Learning, ICML, 2023.

**Experimental Designs Or Analyses:**

Although the theoretical claims are supported by experiments, the experiments mainly focus on simple toy problems (e.g., colliding agents). More real-world and complex experimental setups are necessary to fully assess the robustness and scalability of the proposed methods in practical settings.

**Methods And Evaluation Criteria:**

The introduction of HyperFeatureAttention and HyperAttention extends the self-attention mechanism by capturing higher-order interactions, a potentially useful contribution for complex multi-agent systems and tasks involving feature dependencies. And it demonstrates that the proposed linear self-attention framework converges to optimal solutions under mild assumptions and generalizes well across varying sequence lengths and tasks.

**Other Comments Or Suggestions:**

The paper emphasizes the interpretability of the learned parameters, but in some cases, the learned parameters (especially with sinusoidal embeddings) lack clear, intuitive interpretations. Further exploration on how to make these parameters more interpretable in complex settings would strengthen the practical relevance of the work.

**Other Strengths And Weaknesses:**

While the introduction of HyperFeatureAttention and HyperAttention is conceptually valuable, the increase in complexity might hinder the practical applicability of these extensions for large-scale problems. The computational cost and potential trade-offs between expressiveness and efficiency are not fully addressed.

**Questions For Authors:**

1. The HyperAttention mechanism captures higher-order dependencies, but its computational cost scales cubically. How do you plan to optimize this mechanism for large-scale applications? Are there approximations or heuristics you are considering to make it more efficient in real-world settings?
2. While the experiments validate your theoretical findings on simple tasks like colliding agents, how does your framework perform on more complex and realistic datasets (e.g., NLP tasks, computer vision, multi-agent reinforcement learning)? Can you extend the experiments to such tasks to further validate the scalability and generalization of your methods?
3. While you discuss the interpretability of learned parameters, some experiments (especially those with sinusoidal embeddings) reveal that the learned parameters do not always align with intuitive expectations. Could you provide further insights on how you plan to improve interpretability in more complex settings or for larger models?

**Relation To Broader Scientific Literature:**

This paper has made a broad literature.

**Theoretical Claims:**

The paper provides a clear and unified framework to interpret self-attention mechanisms through the lens of mutual interactions across various domains. It offers new insights into how self-attention works and generalizes, specifically with respect to out-of-distribution shifts and interaction learning

---

> ### Author Rebuttal · Authors · 2025-04-01
>
> LINK (please open in a private window): https://drive.google.com/file/d/1lJ3HYR6i02jpm3CAxg4cu4J6icQXubJ8/view?usp=share_link
>
> *RESPONSE 1* (On Computational Complexity of Models and Response to Question 1): In our appendix, we showed that by leveraging linear self-attention approximations, the computational complexity of HyperAttention with even softmax activation function can be reduced to $\mathcal{O}(L)$. We suggest looking at "RESPONSE 6" part of our reply to reviewer xG3k, where we detailly answered this question and what kind of revisions we will do.
>
> *RESPONSE 2* (to Question2): For self attention, our main goal is to provide rigorous theories for the understanding of the existing self attention mechanism, as opposed to conducting large-scale experiments (which has already been done on many literature as we cited). Our current experiments validate the theoretical framework of linear self-attention.
>
> *RESPONSE 3* (Regarding emprical evidence for Hyper(Feature)Attention): We agree experimental verification of HFA and HA are important. As this work is primarily theoretical, we now present small-scale experiments while working on large-scale versions in a following paper. In our small-scale experiments, we evaluate next-token prediction under two setting. 1) 1-layer 1-head self-attention, HyperFeatureAttention with context window of 1024 but all of the rest of the hyperparemeters are the same as GPT-2 paper (e.g. embedding dimension is 768 etc.)2) 3 layer 3 head self-attention, HyperFeatureAttention experiments with all of the hyperparameters the same as GPT-2 (context window of 1024, embedding dimension of 768 and so on). 3) 1-layer 1-head self-attention, HyperAttention, HyperFeatureAttention with context window of 256 but all of the rest of the hyperparemeters are the same as GPT-2 paper (e.g. embedding dimension is 768 etc.)
> In both experiments the networks we compare have the same number of parameters. Self-attention and HyperFeatureAttention networks have the same computational complexities but HyperAttention has $\mathcal{O}(L^3)$ complexity (we did not apply the approximations reducing complexity yet).
> We deliberately adopted certain design choices that do not favor our proposed models which implicitly favor self attention to ensure a rigorous evaluation. Although these results are preliminary, they consistently indicate that networks incorporating HyperAttention and HyperFeatureAttention achieve better perplexities and cross entropy looses compared to those based solely on standard self-attention.
> For the comparison over the results, please refer to the *LINK*. As for large-scale experiments with comparisons to SOTA, we have deferred extensive, large-scale experimens for the new models to a forthcoming follow-up study because 1) this paper focuses on fundamental theoretical insights and 2) we have possible variations that preserve the essence of Hyper(Feature)Attention models yet may enhance empirical performance, thoroughly exploring these refinements and different possible architectures involves extensive research.
>
> In addition, in the various colliding agent example scenarios in Appendix B.1.3 and "Non-Identical Agents Revisited" in Appendix E.1, with several examples, we step by step induce how HyperFeatureAttention is a natural construction from the theoretical insights on self-attention. Similarly, we explain in Section 6 how HyperAttention is natural construction from our self-attention theories. It is also worth noting that HyperFeatureAttention and HyperAttention are designed to supplement rather than replace existing self-attention mechanisms. For instance, in a multi-head architecture, some heads may employ standard self-attention, while others integrate HyperFeatureAttention (to capture cross-coupled feature interactions) some employ HyperAttention (e.g., order-3 or order-4) to capture higher order interactions -even combination of HyperFeature and Hyper Attention is possible.
>
> *RESPONSE 4* (the interpretability of parameters with sinusoidal embedding and response to question 3): We believe that the confusion was merely due to a narration problem. It was stated "..., the learned parameters do not overlap with the parameters we originally devised, especially for the sinusoidal embedding, seen in Figure 2 -these learned parameters lack easy interpretation ... as discussed in Corollary D.5, this outcome is a natural consequence of the generalization theories ... Therefore we focus on" transformed versions of the parameters ... Further our generalization theories are on those transformed versions. "As shown in figures 3 and 4, these matrices are indistinguishable from the theoretical counterparts. With only ${\Theta}(10^{-4})$ mean square distance between their entries. Thus, in this simple setting, we can fully interpret the meaning of the parameters". We will fix narration.
>
> *RESPONSE 5*: For references look at RESPONSE 1 to reviewer Ct39.

---

### Decision · Program_Chairs · 2025-05-01

**Decision:**

Accept (poster)

**Comment:**

The paper introduces HyperFeatureAttention and HyperAttention which are used to factor in higher-order information into the network layers. AC finds this to be a useful contribution that can be important in certain applications. The delivery of the paper is also clear and makes the contributions apparent. Reviewers unanimously accept the paper. AC complies. The authors did not respond to one of the reviewers' comment. I strongly suggest to address this as well as including all the other feedback in the main paper.